# A stochastic model of hippocampal synaptic plasticity with geometrical readout of enzyme dynamics

Yuri Elias Rodrigues[1,2,3§], Cezar M Tigaret[4], Hélène Marie[1,2§], Cian O'Donnell[5,6§], Romain Veltz[3*§]

[1]Université Côte d'Azur, Nice, France; [2]Institut de Pharmacologie Moléculaire et Cellulaire (IPMC), CNRS, Valbonne, France; [3]Inria Center of University Côte d'Azur (Inria), Sophia Antipolis, France; [4]Neuroscience and Mental Health Research Innovation Institute, Division of Psychological Medicine and Clinical Neurosciences,School of Medicine, Cardiff University, Cardiff, United Kingdom; [5]School of Computing, Engineering, and Intelligent Systems, Magee Campus, Ulster University, Londonderry, United Kingdom; [6]School of Computer Science, Electrical and Electronic Engineering, and Engineering Mathematics, University of Bristol, Bristol, United Kingdom

*For correspondence:
romain.veltz@inria.fr

Present address: §Strathclyde Institute of Pharmacy & Biomedical Sciences, University of Strathclyde, Glasgow, United Kingdom

§Co-senior author

Competing interest: The authors declare that no competing interests exist.

**Abstract** Discovering the rules of synaptic plasticity is an important step for understanding brain learning. Existing plasticity models are either (1) top-down and interpretable, but not flexible enough to account for experimental data, or (2) bottom-up and biologically realistic, but too intricate to interpret and hard to fit to data. To avoid the shortcomings of these approaches, we present a new plasticity rule based on a geometrical readout mechanism that flexibly maps synaptic enzyme dynamics to predict plasticity outcomes. We apply this readout to a multi-timescale model of hippocampal synaptic plasticity induction that includes electrical dynamics, calcium, CaMKII and calcineurin, and accurate representation of intrinsic noise sources. Using a single set of model parameters, we demonstrate the robustness of this plasticity rule by reproducing nine published ex vivo experiments covering various spike-timing and frequency-dependent plasticity induction protocols, animal ages, and experimental conditions. Our model also predicts that in vivo-like spike timing irregularity strongly shapes plasticity outcome. This geometrical readout modelling approach can be readily applied to other excitatory or inhibitory synapses to discover their synaptic plasticity rules.

## Editor's evaluation

Synaptic plasticity is a ubiquitous but also highly complex phenomenon and developing a unifying description has been challenging. This study presents a realistic biophysical model of plasticity induction, with a novel read-out of CaMKII and Calcineurin. It is able to describe a wide range of experimental results and sets a new benchmark for realistic computational models.

## Introduction

To understand how brains learn, we need to identify the rules governing how synapses change their strength in neural circuits. The dominant principle at the basis of current models of synaptic plasticity is the Hebb postulate (*Hebb, 1949*) which states that neurons with correlated electrical activity strengthen their synaptic connections, while neurons active at different times weaken their connections. In particular, spike-timing-dependent plasticity (STDP) models (*Blum and Abbott, 1996*;

*Gerstner et al., 1996*; *Eurich et al., 1999*) were formulated based on experimental observations that precise timing of pre- and post-synaptic spiking determines whether synapses are strengthened or weakened (*Debanne et al., 1996*; *Tsodyks and Markram, 1997*; *Bi and Poo, 1998*; *Markram et al., 2011*). However, experiments also found that plasticity induction depends on the rate and number of stimuli delivered to the synapse (*Dudek and Bear, 1992*; *Sjöström et al., 2001*), and the level of dendritic spine depolarisation (*Artola et al., 1990*; *Magee and Johnston, 1997*; *Sjöström and Häusser, 2006*; *Golding et al., 2002*; *Hardie and Spruston, 2009*). The lack of satisfactory plasticity models based solely on neural spiking prompted researchers to consider simple models based on synapse biochemistry (*Castellani et al., 2001*; *Castellani et al., 2005*). Following a proposed role for postsynaptic calcium ($Ca^{2+}$) signalling in synaptic plasticity (*Lisman, 1989*), previous models assumed that the amplitude of postsynaptic calcium controls long-term alterations in synaptic strength, with moderate levels of calcium causing long-term depression (LTD) and high calcium causing long-term potentiation (LTP) (*Shouval et al., 2002*; *Karmarkar and Buonomano, 2002*). However, experimental data suggests that calcium dynamics are also important (*Yang et al., 1999*; *Mizuno et al., 2001*; *Wang et al., 2005*; *Nevian and Sakmann, 2006*; *Tigaret et al., 2016*). As a result, subsequent phenomenological models of plasticity incorporated slow variables that integrate the fast synaptic input signals, loosely modelling calcium and its downstream effectors (*Abarbanel et al., 2003*; *Rubin et al., 2005*; *Rackham et al., 2010*; *Clopath and Gerstner, 2010*; *Kumar and Mehta, 2011*; *Graupner and Brunel, 2012*; *Honda et al., 2013*; *Standage et al., 2014*; *De Pittà and Brunel, 2016*). Concurrently, more detailed models tried to explicitly describe the molecular pathways integrating the calcium dynamics and its stochastic nature (*Cai et al., 2007*; *Shouval and Kalantzis, 2005*; *Miller et al., 2005*; *Zeng and Holmes, 2010*; *Yeung et al., 2004*). However, even these models do not account for data showing that plasticity is highly sensitive to physiological conditions such as the developmental age of the animal (*Dudek and Bear, 1993*; *Meredith et al., 2003*; *Cao and Harris, 2012*; *Cizeron et al., 2020*), extracellular calcium and magnesium concentrations (*Mulkey and Malenka, 1992*; *Inglebert et al., 2020*) and temperature (*Volgushev et al., 2004*; *Wittenberg and Wang, 2006*; *Klyachko and Stevens, 2006*). This limits the predictive power of this class of plasticity models.

An alternative approach taken by several groups (*Bhalla and Iyengar, 1999*; *Jędrzejewska-Szmek et al., 2017*; *Blackwell et al., 2019*; *Chindemi et al., 2022*; *Zhang et al., 2021*) was to model the complex molecular cascade leading to synaptic weight changes. The main benefit is the direct correspondence between the model's components and biological elements, but at the price of numerous poorly constrained parameters. Additionally, the increased number of nonlinear equations and stochasticity makes fitting to plasticity experiment data difficult (*Mäki-Marttunen et al., 2020*).

Subtle differences between experimental STDP protocols can produce completely different synaptic plasticity outcomes, indicative of finely tuned synaptic behaviour as detailed above. To tackle this problem, we devised a new plasticity rule based on a bottom-up, data-driven approach by building a biologically-grounded model of plasticity induction at a single rat hippocampal CA3–CA1 synapse. We focused on this synapse type because of the abundant published experimental data that can be used to quantitatively constrain the model parameters. Compared to previous models in the literature, we aimed for an intermediate level of detail: enough biophysical components to capture the key dynamical processes underlying plasticity induction, but not the detailed molecular cascade underlying plasticity expression; much of which is poorly quantified for the various experimental conditions we cover in this study.

Our model is centred on dendritic spine electrical dynamics, calcium signalling and immediate downstream molecules, which we then map to synaptic strength change via a conceptually new dynamical, geometric readout mechanism. It assumes that a compartment-based description of calcium-triggered processes is sufficient to reproduce known properties of LTP and LTD induction. Also, neither spatially-resolved elements (*Bartol et al., 2015*; *Griffith et al., 2016*) nor calcium-independent processes are required to predict the observed synaptic change. Crucially, the model also captured intrinsic noise based on the stochastic switching of synaptic receptors and ion channels (*Yuste et al., 1999*; *Ribrault et al., 2011*). We report that, with a single set of parameters, the model could account for published data from spike-timing and frequency-dependent plasticity experiments, and variations in physiological parameters influencing plasticity outcomes. We also tested how the model responded to in vivo-like spike timing jitter and spike failures.

## Results

### A multi-timescale model of synaptic plasticity induction

We built a computational model of plasticity induction at a single CA3-CA1 rat glutamatergic synapse (Figure 1). Our goal was to reproduce results on synaptic plasticity that explored the effects of several experimental parameters: fine timing differences between pre and postsynaptic spiking (Figure 2 and Figure 3); stimulation frequency (Figure 4); animal age (Figure 5); external calcium and magnesium (Figure 6); stochasticity in the firing structure (Figure 7), temperature and experimental conditions variations (Supplemental Information). Where possible, we set parameters to values previously estimated from synaptic physiology and biochemistry experiments, and tuned the remainder within physiologically plausible ranges to reproduce our target plasticity experiments (see *Materials and methods*).

The model components are schematized in *Figure 1* (full details in *Materials and methods*). For glutamate release, we used a two-pool vesicle depletion and recycling system, which accounts for short-term presynaptic depression and facilitation. When glutamate is released from vesicles, it can bind to the postsynaptic α-amino-3-hydroxy-5-methyl-4-isoxazolepropionic acid and N-methyl-D-aspartate receptors (AMPArs and NMDArs, respectively), depolarizing the spine head by ~30 mV (*Kwon et al., 2017*; *Jayant et al., 2017*; *Beaulieu-Laroche and Harnett, 2018*). The dendritic spine membrane depolarization causes the activation of voltage-gated calcium channels (VGCCs) and removes magnesium ($[Mg^{2+}]_o$) block from NMDArs. Backpropagating action potentials (BaP) can also depolarize the spine membrane by up to ~60 mV (*Kwon et al., 2017*; *Jayant et al., 2017*). As an inhibitory component, we modelled a gamma-aminobutyric acid receptor (GABAr) synapse on the dendrite shaft (*Destexhe et al., 1998*). Calcium ions influx through VGCCs and NMDArs can activate SK potassium channels (*Adelman et al., 2012*; *Griffith et al., 2016*), which provide a tightly-coupled local negative feedback limiting spine depolarisation. Upon entering the spine, calcium ions also bind to calmodulin (CaM). Calcium-bound CaM in turn activates two major signalling molecules (*Fujii et al., 2013*): $Ca^{2+}$/calmodulin-dependent protein kinase II (CaMKII) and calcineurin (CaN) phosphatase, also known as PP2B (*Saraf et al., 2018*). We included these two enzymes because of the overwhelming evidence that CaMKII activation is necessary for Schaffer-collateral LTP (*Giese et al., 1998*; *Chang et al., 2017*), while CaN activation is necessary for LTD (*O'Connor et al., 2005*; *Otmakhov et al., 2015*). Later, we show how we map the joint activity of CaMKII and CaN to LTP and LTD. Ligand-gated ion channels (ionotropic receptors) and voltage-gated ion channels have an inherent random behavior, stochastically switching between open, closed and internal states (*Ribrault et al., 2011*). If the number of ion channels is large, then the variability of the total population activity becomes negligible relative to the mean (*O'Donnell and van Rossum, 2014*). However individual hippocampal synapses contain only small numbers of receptors and ion channels, for example they contain ~10 NMDArs and <15 VGCCs (*Takumi et al., 1999*; *Sabatini and Svoboda, 2000*; *Nimchinsky et al., 2004*), making their total activation highly stochastic. Therefore, we modelled AMPAr, NMDAr, VGCCs and GABAr as stochastic processes. Presynaptic vesicle release events were also stochastic: glutamate release was an all-or-none event, and the amplitude of each glutamate pulse was drawn randomly, modelling heterogeneity in vesicle size (*Liu et al., 1999*). The inclusion of stochastic processes to account for an intrinsic noise in synaptic activation (*Deperrois and Graupner, 2020*) contrasts with most previous models in the literature, which either represent all variables as continuous and deterministic or add an external generic noise source (*Bhalla, 2004*; *Antunes and De Schutter, 2012*; *Bartol et al., 2015*).

The synapse model showed nonlinear dynamics across multiple timescales. For illustration, we stimulated the synapse with single simultaneous glutamate and GABA vesicle releases (*Figure 1b*). AMPArs and VGCCs open rapidly but close again within a few milliseconds. The dendritic GABAr closes more slowly, on a timescale of ~10 ms. NMDArs, the major calcium source, closes on timescales of ~50 and ~250 ms for the GluN2A and GluN2B subtypes, respectively.

To show the typical responses of the spine head voltage and $Ca^{2+}$, we stimulated the synapse with a single presynaptic pulse (EPSP) paired 10 ms later with a single BaP (1Pre1Post10; *Figure 1c left*). For this pairing, the arrival of a BaP at the spine immediately after an EPSP, leads to a large $Ca^{2+}$ transient aligned with the BaP due to the NMDArs first being bound by glutamate then unblocked by the BaP depolarisation (*Figure 1c right*).

Single pre or postsynaptic stimulation pulses did not cause depletion of vesicle reserves or substantial activation of the enzymes. To illustrate these slower-timescale processes, we stimulated the synapse with a prolonged protocol: one presynaptic pulse followed by one postsynaptic pulse 10 ms

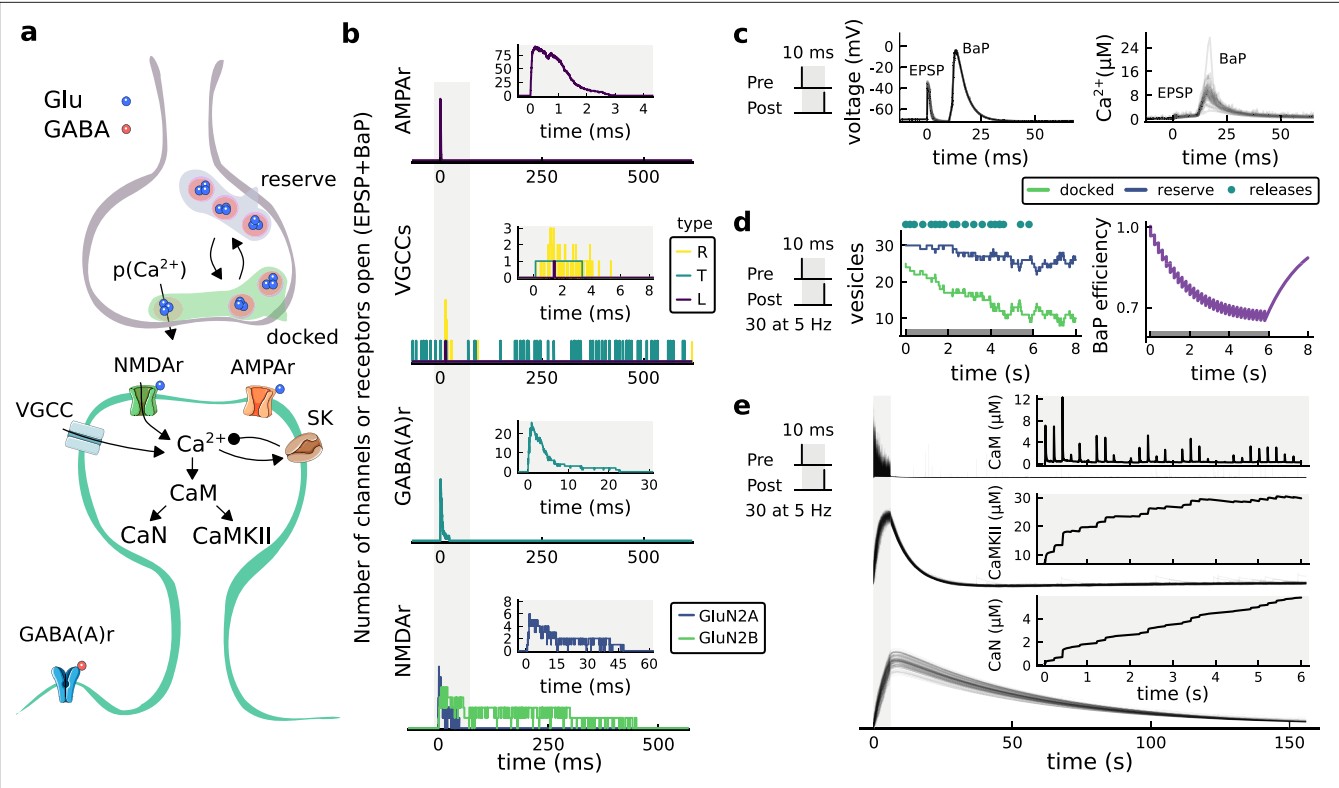

**Figure 1.** | The synapse model, its timescales and mechanisms. (**a**), Model diagram with the synaptic components including pre and postsynaptic compartments and inhibitory transmission (bottom left). AMPAr, NMDAr: AMPA- and NMDA-type glutamate receptors respectively; GABA(A)r: Type A GABA receptors; VGCC: R-, T- and L-type voltage-gated Ca²⁺ channels; SK: SK potassium channels. (**b**), Stochastic dynamics of the different ligand-gated and voltage-gated ion channels in the model. Plots show the total number of open channels as a function of time. The insets show a zoomed time axis highlighting the difference in timescale of the activity among the channels. (**c**), Dendritic spine membrane potential (left) and calcium concentration (right) as function of time for a single causal (1Pre1Post10) stimulus (EPSP: single excitatory postsynaptic potential, '1Pre'; BaP: single back-propagated action potential, '1Post'). (**d**), Left: depletion of vesicle pools (reserve and docked) induced by 30 pairing repetitions delivered at 5 Hz (***Sterratt et al., 2011***), see *Materials and methods*. The same depletion rule is applied to both glutamate- and GABA-containing vesicles. Right: BaP efficiency as function of time. BaP efficiency phenomenologically captures the distance-dependent attenuation of BaP (***Buchanan and Mellor, 2007***; ***Golding et al., 2001***), see *Materials and methods*. (**e**), Concentration of active enzyme for CaM, CaN, and CaMKII, as function of time triggered by 30 repetitions of 1Pre1Post10 pairing stimulations delivered at 5 Hz. The vertical grey bar is the duration of the stimuli, 6 s. The multiple traces in the graphs in panels **c** (right) and e reflect the run-to-run variability due to the inherent stochasticity in the model.

later, repeated 30 times at 5 Hz (***Figure 1d–e***). The number of vesicles in both the docked and reserve pools decreased substantially over the course of the stimulation train (***Figure 1d left***), which in turn caused decreased vesicle release probability. Similarly, by the 30th pulse, the dendritic BaP amplitude had attenuated to ~85% (~70% BaP efficiency; ***Figure 1d right***) of its initial amplitude, modelling the effects of slow dendritic sodium channel inactivation (***Colbert et al., 1997***; ***Golding et al., 2001***). Free CaM concentration rose rapidly in response to calcium transients but also decayed back to baseline on a timescale of ~500 ms (***Figure 1e top***). In contrast, the concentration of active CaMKII and CaN accumulated over a timescale of seconds, reaching a sustained peak during the stimulation train, then decayed back to baseline on a timescale of ~10 and ~120 s respectively, in line with experimental data (***Quintana et al., 2005***; ***Fujii et al., 2013***; ***Chang et al., 2017***; ***Figure 1e***).

The effects of the stochastic variables can be seen in ***Figure 1b–d***. The synaptic receptors and ion channels open and close randomly (***Figure 1b***). Even though spine voltage, calcium, and downstream molecules were modelled as continuous and deterministic, they inherited some randomness from the upstream stochastic variables. As a result, there was substantial trial-to-trial variability in the voltage and calcium responses to identical pre and postsynaptic spike trains (grey traces in ***Figure 1c***). This variability was also passed on to the downstream enzymes CaM, CaMKII and CaN, but was filtered and therefore attenuated by the slow dynamics of CaMKII and CaN. In summary, the model contained

stochastic nonlinear variables acting over five different orders of magnitude of timescale, from ~1 ms to ~1 min, making it sensitive to both fast and slow components of input signals.

## Distinguishing between stimulation protocols using the CaMKII and CaN joint response

It has proven difficult for simple models of synaptic plasticity to capture the underlying rules and explain why some stimulation protocols induce plasticity while others do not. We tested the model's sensitivity by simulating its response to a set of protocols used by *Tigaret et al., 2016* in a recent ex vivo experimental study on adult (P50-55) rat hippocampus with blocked GABAr. We schematized the *Tigaret et al., 2016* protocols in *Figure 2a*. Notably, three leading spike-timing and calcium-dependent plasticity models (*Song et al., 2000*; *Pfister and Gerstner, 2006*; *Graupner and Brunel, 2012*) could not fit well these data (*Figure 2b–d*). Next, we asked if our new model could distinguish between three pairs of protocols (see *Figure 2e–m*). For each of these pairs, one of the protocols experimentally induced LTP or LTD, while the other subtly different protocol caused no change (NC) in synapse strength.

The first pair of protocols differed in intensity. A protocol which caused no plasticity consisted of 1 presynaptic spike followed 10 ms later by one postsynaptic spike repeated at 5 Hz for 1 min (1Pre1Post10, 300 at 5 Hz). The other protocol induced LTP, but differed only in that it included a postsynaptic doublet instead of a single spike (1Pre2Post10, 300 at 5 Hz), implying a slightly stronger initial BaP amplitude. We first attempted to achieve separability by plotting CaMKII or CaN activities independently. As observed in the plots in *Figure 2e*, it was not possible to set a single concentration threshold on either CaMKII or CaN that would discriminate between the protocols. This result was expected, at least for CaMKII, as recent experimental data demonstrates a fast saturation of CaMKII concentration in dendritic spines regardless of stimulation frequency (*Chang et al., 2017*).

To achieve better separability we set out to test a different approach, which was to combine the activity of the two enzymes, by plotting the joint CaMKII and CaN responses against each other on a 2D plane (*Figure 2f*). This innovative geometric plot is based on the mathematical concept of orbits from dynamical systems theory (*Meiss, 2007*). In this plot, the trajectories of two protocols can be seen to overlap for the initial part of the transient and then diverge. To quantify trial-to-trial variability, we also calculated contour maps showing the mean fraction of time the trajectories spent in each part of the plane during the stimulation (*Figure 2g*). Importantly, both the trajectories and contour maps were substantially non-overlapping between the two protocols, implying that they can be separated based on the joint CaN-CaMKII activity. We found that the 1Pre2Post10 protocol leads to a weaker response in both CaMKII and CaN, corresponding to the lower blue traces in *Figure 2f*. The decreased response to the doublet protocol was due to the stronger attenuation of dendritic BaP amplitude over the course of the simulation (*Golding et al., 2001*), leading to reduced calcium influx through NMDArs and VGCCs (data not shown).

Using the second pair of protocols, we explored if this combined enzyme activity analysis could distinguish between subtle differences in protocol sequencing. We stimulated our model with one causal paring protocol (EPSP-BaP) involving a single presynaptic spike followed 50 ms later by a doublet of postsynaptic spikes (1Pre2Post50, 300 at 5 Hz), repeated at 5 Hz for one minute, which caused LTP in *Tigaret et al., 2016*. The other, anticausal, protocol involved the same total number of pre and postsynaptic spikes, but with the pre-post order reversed (2Post1Pre50, 300 at 5 Hz). Experimentally, the anticausal (BaP-EPSP) protocol did not induce plasticity (*Tigaret et al., 2016*). Notably, the only difference was the sequencing of whether the pre or postsynaptic neuron fired first, over a short time gap of 50 ms. Although the time courses of CaMKII and CaN activities were difficult to distinguish (*Figure 2h*), the LTP-inducing protocol caused greater CaN activation, compared to the non LTP-inducing protocol. Indeed, this translated to a horizontal offset in both the trajectory and contour map (*Figure 2i–j*), demonstrating that this pair of protocols can also be separated in the joint CaN-CaMKII plane.

The third pair of protocols differed in both duration and intensity. We thus tested the combined enzyme activity analysis in this configuration. In line with a previous study (*Isaac et al., 2009*), *Tigaret et al., 2016* found that a train of doublets of presynaptic spikes separated by 50 ms repeated at a low frequency of 3 Hz for 5 min (2Pre50, 900 at 3 Hz) induced LTD, while a slightly more intense but shorter duration protocol of presynaptic spike doublets separated by 10 ms repeated at 5 Hz for 1

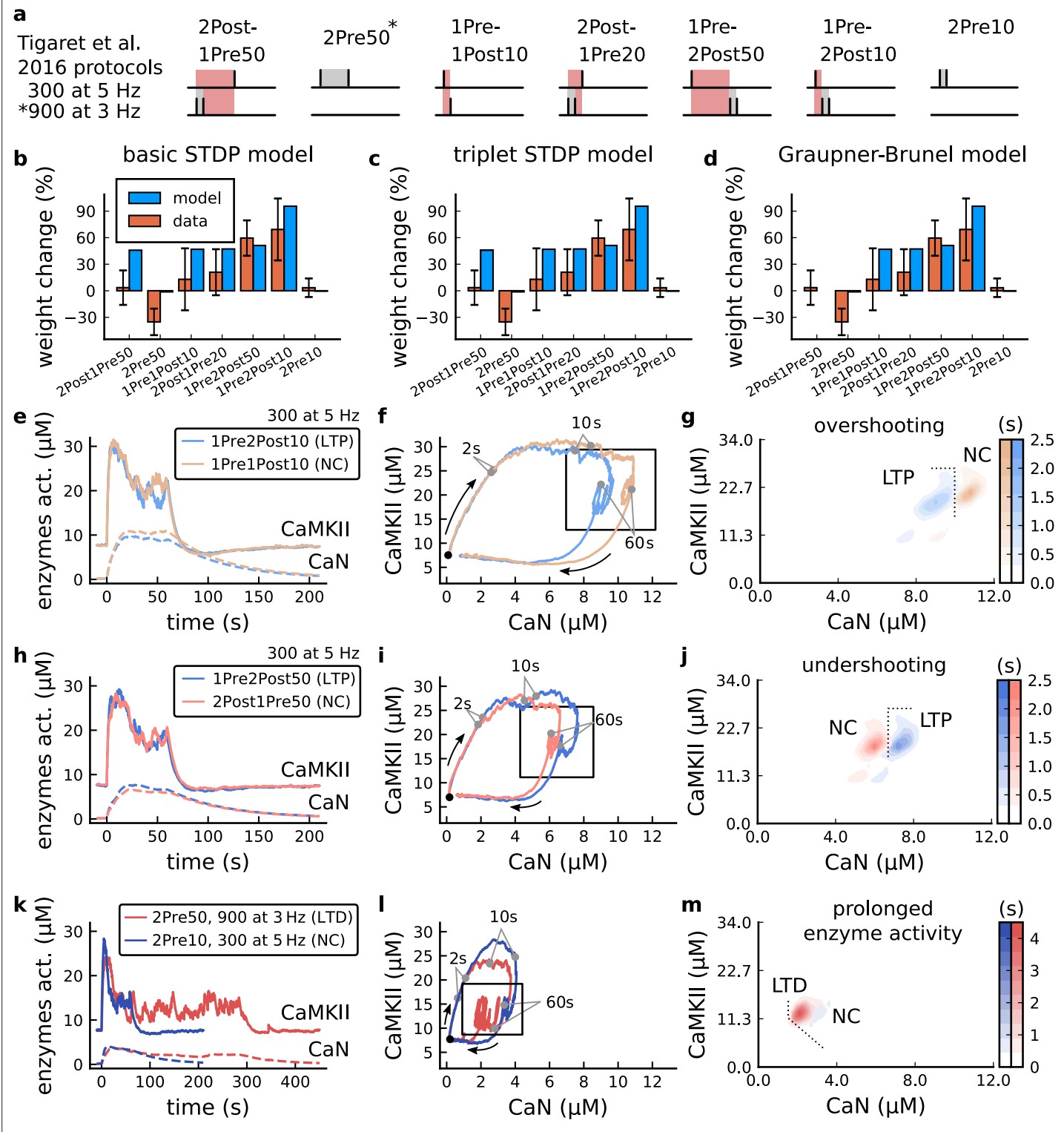

**Figure 2.** | The duration and amplitude of the joint CaN-CaMKII activity differentiates plasticity protocols. (**a**), *Tigaret et al., 2016* protocols, which inspired this model.(**a**) is adapted from Figure 2B from *Tigaret et al., 2016*. (**b–d**), Standard models for predicting plasticity fail to account for *Tigaret et al., 2016* data. Mean weight change for the Tigaret's data (red), error bars denote ±1 s.d. Plasticity protocols indicated by labels on x-axis. Blue bars show mean plasticity predicted for the same protocols by classic STDP model (*Song et al., 2000*) (panel **b**), triplet STDP model (*Pfister and Gerstner, 2006*) (panel **c**), or Graupner-Brunel calcium-based STDP (*Graupner and Brunel, 2012*) model (panel **d**). (**e**), Time-course of active enzyme concentration for CaMKII (solid line) and CaN (dashed line) triggered by two protocols consisting of 300 repetitions at 5 Hz of 1Pre2Post10 or

*Figure 2 continued on next page*

*Figure 2 continued*

1Pre1Post10 stimulus pairings. Protocols start at time 0 s. Experimental data indicates that 1Pre2Post10 and 1Pre1Post10 produce LTP and no change (NC), respectively. (**f**), Trajectories of joint enzymatic activity (CaN-CaMKII) as function of time for the protocols in panel **e**, starting at the initial resting state (filled black circle). The arrows show the direction of the trajectory and filled grey circles indicate the time points at 2, 10, and 60 s after the beginning of the protocol. The region of the CaN-CaMKII plane enclosed in the black square is expanded in panel **g**. (**g**), Mean-time (colorbar) spent by the orbits in the CaN-CaMKII plane region expanded from panel f for each protocol (average of 100 samples). For panels **g, j and m** the heat maps were based on enzyme and 2Post1Pre50 (NC) depicted in the same manner as in panels (**e-g**). (**k-m**), CaN-CaMKII activities for the LTD-inducing protocol 2Pre50 (900 repetitions at 3 Hz) and the NC protocol 2Pre10 (300 repetitions at 5 Hz) depicted in the same manner as in panels **e-g**.

min (2Pre10, 300 at 5 Hz) did not cause plasticity. When we simulated both protocols in the model (*Figure 2k–m*), both caused similar initial responses in CaMKII and CaN. In the shorter protocol, this activation decayed to baseline within 100 s of the end of the stimulation. However the slower and longer-lasting 2Pre50 3 Hz 900 p protocol caused an additional sustained, stochastically fluctuating, plateau of activation of both enzymes (*Figure 2k*). This resulted in the LTD-inducing protocol having a downward and leftward-shifted CaN-CaMKII trajectory and contour plot, relative to the other protocol (*Figure 2l–m*). These results again showed that the joint CaN-CaMKII activity can be used to predict plasticity changes.

## A geometrical readout mapping joint enzymatic activity to plasticity outcomes

The three above examples demonstrated that plotting the combined CaN-CaMKII activities in a 2D plane (geometrical readout which is abstract e.g. not defined within a physical space) allowed us to distinguish between subtly different protocols with correct assignment of plasticity outcome. We found that the simulated CaN-CaMKII trajectories from the two LTP-inducing protocols (*Figure 2e–g* and *Figure 2h–j*) spent a large fraction of time near ~20 μM CaMKII and 7–10 μM CaN. In contrast, protocols that failed to trigger LTP had either lower (*Figure 2h–j, k–m*), or higher CaMKII and CaN activation (1Pre1Post10, *Figure 2e–g*). The LTD-inducing protocol, by comparison, spent a longer period in a region of sustained but lower ~12 μM CaMKII and ~2 μM CaN and activation *Figure 2k–m*. The plots in *Figure 2g, j and m*, show contour maps of histograms of the joint CaMKII-CaN activity, indicating where in the plane the trajectories spent most time. *Figure 2g and j* indicate that this measure can be used to predict plasticity, because the NC and LTP protocol histograms are largely non-overlapping. In *Figure 2g*, the NC protocol response 'overshoots' (mostly due to higher CaN concentration) the LTP protocol response, whereas in *Figure 2j* the NC protocol response 'undershoots' (mostly due to lower CaN concentration) the LTP protocol response. In contrast, when we compared the response histograms for the LTD and NC protocols, we found a greater overlap (*Figure 2m*). This suggested that, in this case, the histogram alone was not sufficient to separate the protocols, and that protocol duration is also important. LTD induction (2Pre50) required a more prolonged activation than NC (2Pre10). We thus took advantage of these joint CaMKII-CaN activity maps to design a minimal readout mechanism connecting combined enzyme activity to LTP, LTD or NC. We reasoned that this readout would need three key properties. First, although the figure suggests that the CaMKII-CaN trajectories corresponding to LTP and LTD could be linearly separable, we will demonstrate later (see *Figure 3—figure supplement 3*) that the readout requires nonlinear boundaries to activate the plasticity inducing component. Second, since LTD requires more prolonged activity than LTP, the readout should be sensitive to the timescale of the input. Third, a mechanism is required to convert the 2D LTP-LTD inducing signals into a synaptic weight change. After iterating through several designs, we satisfied the first property by designing 'plasticity regions': polygons in the CaN-CaMKII plane that would detect when trajectories pass through. We satisfied the second property by using two plasticity inducing components with different time constants which low-pass-filter the plasticity region signals. We satisfied the third property by feeding both the opposing LTP and LTD signals into a stochastic Markov chain which accumulated the total synaptic strength change. Overall, this readout mechanism acts as a parsimonious model of the complex signalling cascade linking CaMKII and CaN activation to expression of synaptic plasticity (*He et al., 2015*). It can be considered as a two-dimensional extension of previous computational studies that applied analogous 1D threshold functions to dendritic spine calcium concentration (*Shouval et al., 2002*; *Karmarkar and Buonomano, 2002*; *Graupner and Brunel, 2012*; *Standage et al., 2014*).

We now elaborate on the readout design process (see also Figure 21 of *Materials and methods*). We first drew non-overlapping polygons of LTP and LTD 'plasticity regions' in the CaN-CaMKII plane (*Figure 3a*). We positioned these regions over the parts of the phase space where the enzyme activities corresponding to the LTP- and LTD-inducing protocols were most different (*Materials and methods*), as shown by trajectories in *Figure 2f, i and l*. When a trajectory enters in one of these plasticity regions, it activates LTD or LTP indicator variables (*Materials and methods*) which encode the joint enzyme activities (trajectories in the phase plots) transitions across the LTP and LTD regions over time (*Figure 3b*). These indicator variables drove transition rates of a plasticity Markov chain used to predict LTP or LTD (*Figure 3c*), see *Materials and methods*. Intuitively, this plasticity Markov chain models the competing processes of insertion/deletion of AMPARs to the synapse, although this is not represented in the model. The LTD transition rates were slower than the LTP transition rates, to reflect studies showing that LTD requires sustained synaptic stimulation (*Yang et al., 1999*; *Mizuno et al., 2001*; *Wang et al., 2005*). The parameters for this plasticity Markov chain (*Materials and methods*) were fit to the plasticity induction outcomes from different protocols (*Appendix 1—table 1*). At the beginning of the simulation, the plasticity Markov chain starts with 100 processes (*Destexhe et al., 1998*) in the NC state, with each variable representing 1% weight change, an abstract measure of synaptic strength that can be either EPSP, EPSC, or field EPSP slope depending on the experiment. Each process can transit stochastically between NC, LTP and LTD states. At the end of the protocol, the plasticity outcome is given by the difference between the number of processes in the LTP and the LTD states (*Materials and methods*).

In *Figure 3d*, we plot the model's responses to seven different plasticity protocols used by *Tigaret et al., 2016* by overlaying example CaMKII-CaN trajectories for each protocol with the LTP and LTD regions. The corresponding region indicators are plotted as function of time in *Figure 3e*, and long-term alterations in the synaptic strength are plotted as function of time in *Figure 3f*. The three protocols that induced LTP in the *Tigaret et al., 2016* experiments spent substantial time in the LTP region, and so triggered potentiation. In contrast, the combined response (CamKII, CaN) to 1Pre1Post10 overshoots both regions, crossing them only briefly on its return to baseline, and so resulted in little weight change. The protocol that induced LTD (2Pre50, purple trace) is five times longer than other protocols, spending sufficient time inside the LTD region (*Figure 3f*). In contrast, two other protocols that spent time in the same LTD region of the CaN-CaMKII plane (2Post1Pre50 and 2Pre10) were too brief to induce LTD. These protocols were also not strong enough to reach the LTP region, so resulted in no net plasticity, again consistent with *Tigaret et al., 2016* experiments.

We observed run-to-run variability in the amplitude of the predicted plasticity, due to the inherent stochasticity in the model. To ensure that stochastic components are necessary for adequate model behaviour, we compared stochastic and deterministic versions of the model with and without discrete presynaptic release and found that adding stochastic components indeed modified the model's behaviour (*Figure 3—figure supplement 1*). Also, we confirmed that VGCCs are necessary for accurate modelling of *Tigaret et al., 2016* data as blocking these channels reproduced the data obtained in VGCC blockers by Tigaret that is no potentiation could be elicited (*Figure 3—figure supplement 2*). Finally, we stress in *Figure 3—figure supplement 3* that the horizontal boundaries (related to CaMKII activity) are indeed necessary.

In *Figure 3g*, we plot the distributions of the simulation outcomes, along with the experimental data, for the protocols in *Tigaret et al., 2016*. We find a very good correspondence between the model and experiments. Of note, data fitting of the experiments in *Tigaret et al., 2016* (*Figure 3g*) was more accurate with our model than the fitting obtained with existing leading spike-or calcium-based STDP models (*Song et al., 2000*; *Pfister and Gerstner, 2006*; *Graupner and Brunel, 2012*), see *Figure 2b–d*.

Experimentally, LTP can be induced by few pulses while LTD usually requires stimulation protocols of longer duration (*Yang et al., 1999*; *Mizuno et al., 2001*; *Wang et al., 2005*). We incorporated this effect into the geometrical readout model by letting LTP have faster transition rates than LTD (*Figure 3c*). *Tigaret et al., 2016* found that 300 repetitions of anticausal post-before-pre pairings did not cause LTD, in contrast to the canonical spike-timing-dependent plasticity curve (*Bi and Poo, 1998*). We hypothesized that LTD might indeed appear with the anticausal protocol (*Appendix 1—table 1*) if stimulation duration was increased. To explore this possibility in our model, we systematically *Alabi and Tsien, 2012* varied the number of paired repetitions from 100 to 1200, and also co-varied the

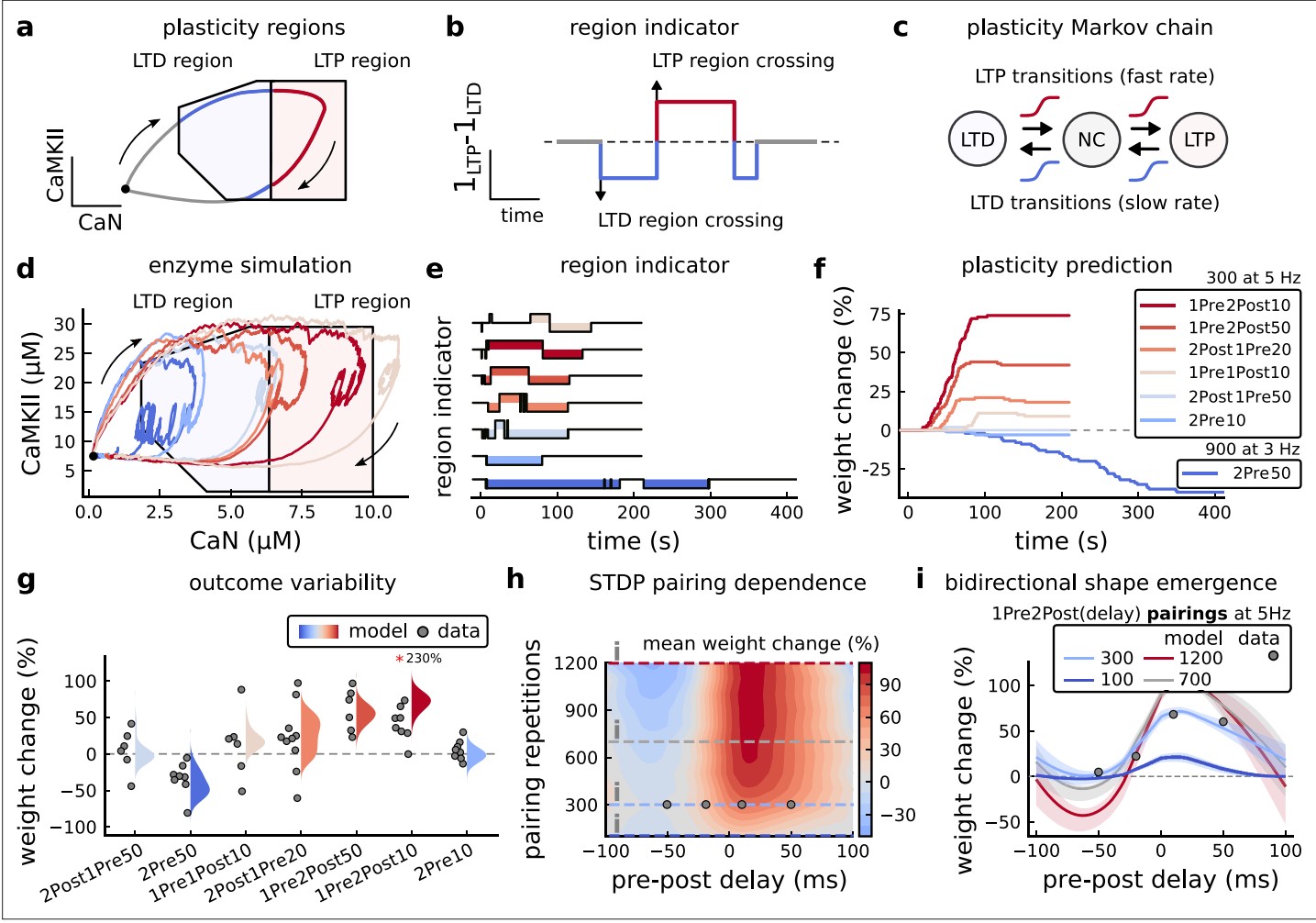

**Figure 3.** Read-out strategy to accurately model *Tigaret et al., 2016* experiment. (**a**) Illustration of the joint CaMKII and CaN activities crossing the plasticity regions. Arrows indicate the flow of time, starting at the filled black circle. (**b**) Region indicator showing when the joint CaN and CaMKII activity crosses the LTD or LTP regions in panel a. For example, the LTP indicator is such that $1_{LTP}(x) = 1$ if $x \in LTP$ and 0 otherwise. Leaving the region activates a mechanism with a slow timescale that keeps track of the accumulated time inside the region. Such mechanism drives the transition rates used to predict plasticity (*Materials and methods*). (**c**), Plasticity Markov chain with three states: LTD, LTP and NC. There are only two transition rates which are functions of the plasticity region indicator (*Materials and methods*). The LTP transition is fast, whereas the LTD transition is slow, meaning that LTD change requires longer time inside the LTD region (panel **a**). The NC state starts with 100 processes. See Figure 23 for more details on the dynamics of the Plasticity Markov Chain. (**d**) Joint CaMKII and CaN activity for all protocols in *Tigaret et al., 2016* (shown in panel **f**). The stimulus ends when the trajectory becomes smooth. Trajectories correspond to those in *Figure 2b, e and h*, at 60 s. (**e**) Region indicator for the protocols in panel **f**. The upper square bumps are caused by the protocol crossing the LTP region, the lower square bumps when the protocol crosses the LTD region (as in panel **d**). (**f**) Synaptic weight (%) as function of time for each protocol. The weight change is defined as the number (out of 100) of states in the LTP state minus the number of states in the LTD state (panel **c**). The trajectories correspond to the median of the simulations in panel **g**. (**g**) Synaptic weight change (%) predicted by the model compared to data (EPSC amplitudes) from *Tigaret et al., 2016* (100 samples for each protocol, also for panel **h and i**). The data (filled grey circles) was provided by *Tigaret et al., 2016* (note an 230% outlier as the red asterisk). (**h**) Predicted mean synaptic weight change (%) as a function of delay (ms) and number of pairing repetitions (pulses) for the protocol 1Pre2Post(delay), where delays are between –100 and 100 ms. LTD is induced by 2Post1Pre50 after at least 500 pulses. The mean weight change along the horizontal dashed line is reported in the STDP curves in panel **i**. (**i**) Synaptic weight change (%) as a function of pre-post delay. Each plot corresponds to a different pairing repetition number (color legend). The solid line shows the mean, and the ribbons are the 2nd and 4th quantiles. The filled grey circles are the data means estimated in *Tigaret et al., 2016*, also shown in panel **g**.

The online version of this article includes the following figure supplement(s) for figure 3:

**Figure supplement 1.** Comparison showing different roles of stochasticity in the model.

**Figure supplement 2.** Effects of blocking VGCCs.

**Figure supplement 3.** Exclusively setting vertical boundaries (no CaMKII selectivity) fails to capture the correct plasticity outcome.

**Figure supplement 4.** Varying *Tigaret et al., 2016* experimental parameters.

pre-post delay from −100 to 100 ms. *Figure 3h* shows a contour plot of the predicted mean synaptic strength change across for the 1Pre2Post(delay) stimulation protocol for different numbers of pairing repetitions. In *Figure 3h and a* LTD window appears after ~500 pairing repetitions for some anti-causal pairings, in line with our expectation. The magnitude of LTP also increases with pulse number, for causal positive pairings. For either 100 or 300 pairing repetitions, only LTP or NC is induced (*Figure 3i*). The model also made other plasticity predictions by varying *Tigaret et al., 2016* experimental conditions (*Figure 3—figure supplement 4*). In summary, our geometrical readout mechanism suggests that the direction and magnitude of the change in synaptic strength can be predicted from the joint CaMKII-CaN activity in the LTP and LTD regions.

## Frequency-dependent plasticity

The stimulation protocols used by *Tigaret et al., 2016* explored how subtle variations in pre and postsynaptic spike timing influenced the direction and magnitude of plasticity (see *Appendix 1—table 1* for experimental differences). In contrast, traditional synaptic plasticity protocols exploring the role of presynaptic stimulation frequency did not measure the timing of co-occurring postsynaptic spikes (*Dudek and Bear, 1992*; *Wang and Wagner, 1999*; *Kealy and Commins, 2010*). These studies found that long-duration low-frequency stimulation (LFS) induces LTD, whereas short-duration high-frequency stimulation induces LTP, with a cross-over point of zero change at intermediate stimulation frequencies. In addition to allowing us to explore frequency-dependent plasticity (FDP), this stimulation paradigm also gave us further constraints to define the LTD polygon region in the model since in *Tigaret et al., 2016*, only one LTD case was available. For FDP, we focused on modelling the experiments from *Dudek and Bear, 1992*, who stimulated Schaffer collateral projections to pyramidal CA1 neurons with 900 pulses in frequencies ranging from 1 to 50 Hz. In addition to presynaptic stimulation patterns, the experimental conditions differed from *Tigaret et al., 2016* in two other aspects: animal age and control of postsynaptic spiking activity (see *Appendix 1—table 1* legend). We incorporated both age-dependence and EPSP-evoked-BaPs in our model (*Materials and methods*). Importantly, the geometrical readout mechanism mapping joint CaMKII-CaN activity to plasticity remained identical for all experiments in this work.

*Figure 4a* shows the joint CaMKII-CaN activity when we stimulated the model with 900 presynaptic spikes at 1, 3, 5, 10, and 50 Hz (*Dudek and Bear, 1992*). Higher stimulation frequencies drove stronger responses in both CaN and CaMKII activities (*Figure 4a*). *Figure 4b and c* show the corresponding plasticity region indicator for the LTP/LTD region threshold crossings and the synaptic strength change. From this set of five protocols, only the 50 Hz stimulation drove a response strong enough to reach the LTP region of the plane (*Figure 4a and d*). Although the remaining four protocols drove responses primarily in the LTD region, only the 3 and 5 Hz stimulations resulted in substantial LTD. The 1 and 10 Hz stimulations resulted in negligible LTD, but for two distinct reasons. Although the 10 Hz protocol's joint CaMKII-CaN activity passed through the LTD region of the plane (*Figure 4a and d*), it was too brief to activate the slow LTD mechanism built into the readout (*Materials and methods*). The 1 Hz stimulation, on the other hand, was prolonged, but its response was too weak to reach the LTD region, crossing the threshold only intermittently (*Figure 4b*, bottom trace). Overall, the model matched well the mean plasticity response found by *Dudek and Bear, 1992*, see *Figure 4e*, following a classic BCM-like curve as function of stimulation frequency (*Abraham et al., 2001*; *Bienenstock et al., 1982*).

We then used the model to explore the stimulation space in more detail by varying the stimulation frequency from 0.5 to 40 Hz, and varying the number of presynaptic pulses from 50 to 1200. *Figure 4f* shows a contour map of the mean synaptic strength change (%) in this 2D frequency–pulse number space. *Dudek and Bear, 1992* experimental conditions, we found that LTD induction required at least ~300 pulses, at frequencies between 1 and 3 Hz. In contrast, LTP could be induced using ~50 pulses at ~20 Hz or greater. The contour map also showed that increasing the number of pulses (vertical axis in *Figure 4f*) increases the magnitude of both LTP and LTD. This was accompanied by a widening of the LTD frequency range, whereas the LTP frequency threshold remained around ~20 Hz, independent of pulse number. This general effect, that increasing pulse number tends to increase the magnitude of plasticity, was also observed in simulation of *Tigaret et al., 2016* (see *Figure 3h*). Ex vivo experiments in *Dudek and Bear, 1992* were done at 35°C. However, lower temperatures are more widely usetd for ex vivo experiments because they extend brain slice viability.

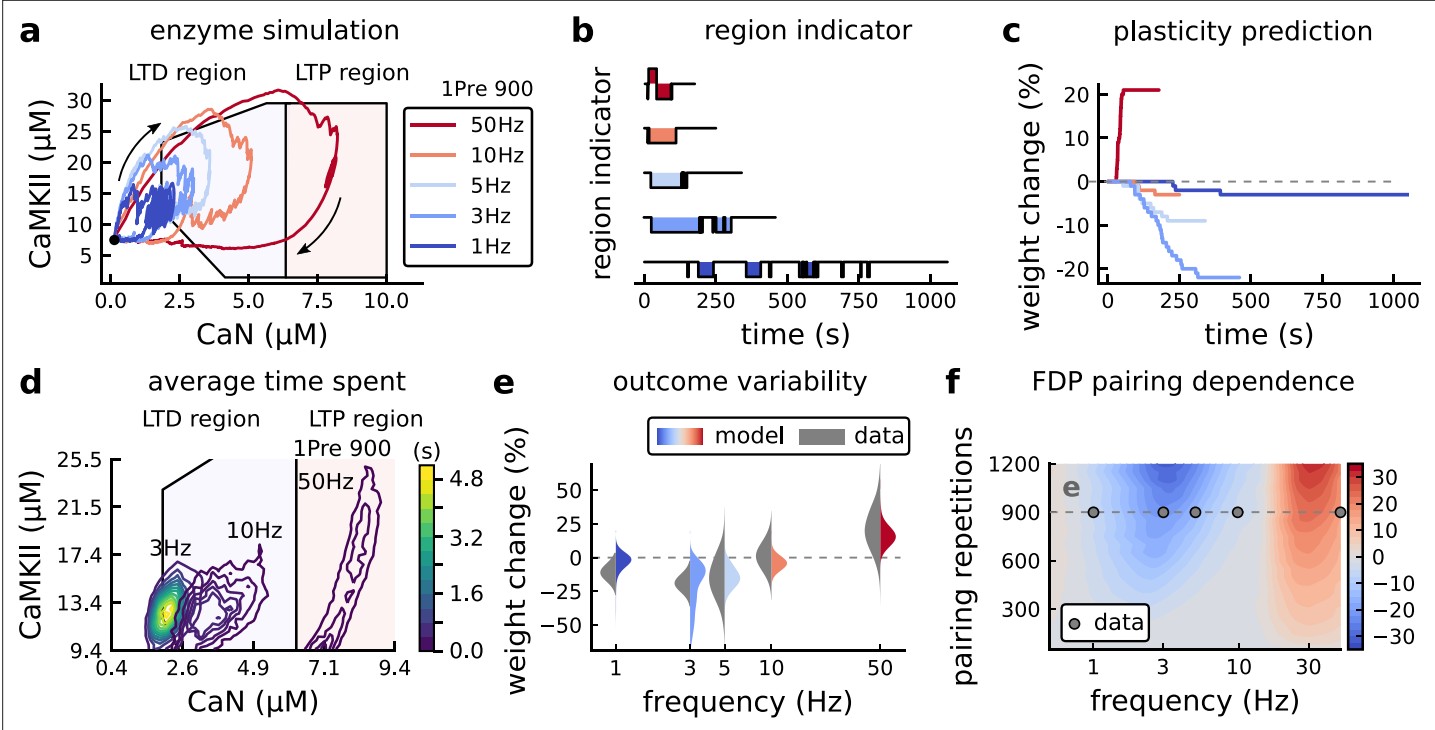

**Figure 4.** Frequency dependent plasticity (FDP), *Dudek and Bear, 1992* dataset. (**a**) Example traces of joint CaMKII-CaN activity for each of *Dudek and Bear, 1992* protocol. (**b**) Region indicator showing when the joint CaMKII-CaN activity crosses the LTD or LTP regions for each protocol in panel **a**. (**c**) Synaptic weight change (%) as a function of time for each protocol, analogous to *Figure 3c*. Trace colours correspond to panel **a**. The trajectories displayed were chosen to match the medians in panel **e**. (**d**) Mean (100 samples) time spent (s) for protocols 1Pre for 900 pairing repetitions at 3, 10, and 50 Hz. (**e**), Comparison between data from *Dudek and Bear, 1992* and our model (1Pre 900 p, 300 sampcomles per frequency, see *Appendix 1— table 1*). Data are represented as normal distributions with the mean and variance of the change in field EPSP slope taken from *Dudek and Bear, 1992*. (**f**), Prediction for the mean weight change (%) when varying the stimulation frequency and pulse number (24x38 × 100 data points, respectively pulse x frequency x samples). The filled grey circles show the *Dudek and Bear, 1992* protocol parameters and the corresponding results are shown in panel **e**. In *Figure 4—figure supplement 1*, we provide additional graphs of frequency dependent plasticity outcomes, including predictions, when varying experimental parameters in *Dudek and Bear, 1992* (external Mg, external Ca, distance from soma, temperature, Poisson spike train during development).

The online version of this article includes the following figure supplement(s) for figure 4:

**Figure supplement 1.** Varying experimental parameters in *Dudek and Bear, 1992* and Poisson spike train during development.

**Figure supplement 2.** The figure shows the weight change (%) for *Dudek and Bear, 1992* protocols (50 Hz, 30 Hz, and 3 Hz, related to *Figure 4* of the main manuscript).

At this point, having fully described the model, we show the importance of the stochasticity of the different components of the model. We simulated three protocols of *Dudek and Bear, 1992* with deterministic equations in *Figure 4—figure supplement 2*. We show that, for the different protocols, if some of the channels are modelled with deterministic equations, the net effect on synapse weight differs from the expected outcome provided by the original model. The relative contributions of each source of noise differed, depending on the plasticity protocol. We can conclude that all noise sources we introduced in our model are important.

## Variations in plasticity induction with developmental age

The rules for induction of LTP and LTD change during development (*Dudek and Bear, 1993*; *Cao and Harris, 2012*), so a given plasticity protocol can produce different outcomes when delivered to synapses from young animals versus mature animals. For example, when *Dudek and Bear, 1993* tested the effects of low-frequency stimulation (1 Hz) on CA3-CA1 synapses from rats of different ages, they found that the magnitude of LTD decreases steeply with age from P7 until becoming minimal in mature animals >P35 (*Figure 5a*, circles). Across the same age range, they found that a theta burst stimulation (TBS) protocol induced progressively greater LTP magnitude with developmental age

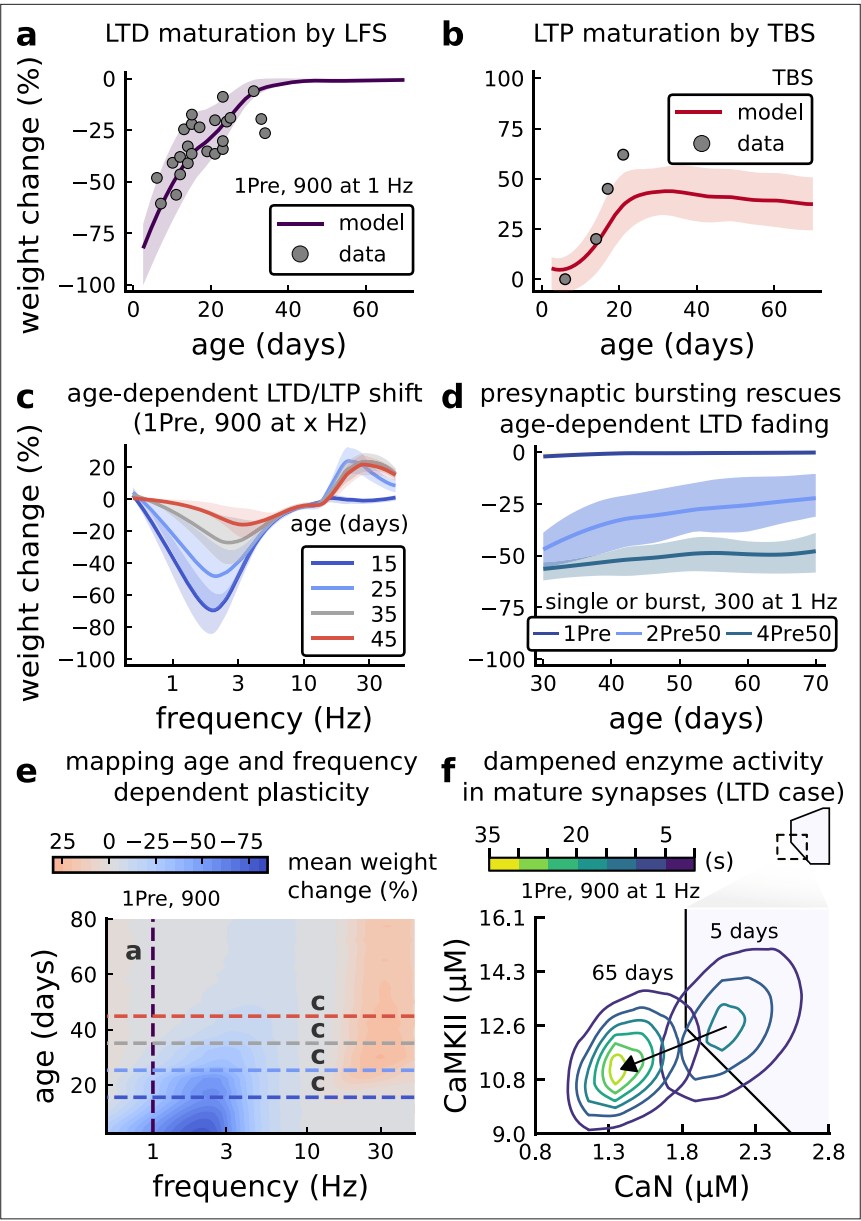

**Figure 5.** Age-dependent plasticity, *Dudek and Bear, 1993* dataset. (**a**), Synaptic weight change for 1Pre, 900 at 1 *Dudek and Bear, 1993*. The solid line is the mean and the ribbons are the 2nd and 4th quantiles predicted by our model (same for panel **b**, **c** and **f**). (**b**), Synaptic weight change for theta burst stimulation (TBS - 4Pre at 100 Hz repeated 10 times at 5 Hz given in 6 epochs at 0.1 Hz, see *Appendix 1—table 1*). (**c**), Synaptic weight change as a function of frequency for different ages. BCM-like curves showing that, during adulthood, the same LTD protocol becomes less efficient. It also shows that high-frequencies are inefficient at inducing LTP before P15. (**d**), Synaptic weight change as a function of age. Proposed protocol using presynaptic bursts to recover LTD at ≥ P35 with less pulses, 300 instead of the original 900 from *Dudek and Bear, 1993*. This effect is more pronounced for young rats. *Figure 5—figure supplement 1* shows a 900 pulses comparison. (**e**), Mean synaptic strength change (%) as a function of frequency and age for 1Pre 900 pulses (32x38 × 100, respectively, for frequency, age and samples). The protocols in *Dudek and Bear, 1993* (panel **a**) are marked with the yellow vertical line. The horizontal lines represent the experimental conditions of panel **c**. Note the P35 was used for *Dudek and Bear, 1992* experiment in *Figure 4f*. (**f**), Mean time spent for the 1Pre 1 Hz 900 pulses protocol showing how the trajectories are left-shifted as rat age increases. In *Figure 5—figure supplement 1*, we provide additional simulations to analyse the synaptic plasticity outcomes, including predictions, of duplets, triplets and quadruplets for FDP, perturbing developmental-mechanisms for *Dudek and Bear, 1993*, and age-related changes in STDP experiments (*Inglebert et al., 2020*; *Tigaret et al., 2016*; *Meredith et al., 2003*).

*Figure 5 continued on next page*

*Figure 5 continued*

The online version of this article includes the following figure supplement(s) for figure 5:

**Figure supplement 1.** Duplets, triplets, and quadruplets for FDP, perturbing developmental-mechanisms for *Dudek and Bear, 1993*, and age-related changes in STDP experiments (*Inglebert et al., 2020*; *Tigaret et al., 2016*; *Meredith et al., 2003*).

---

(*Figure 5b*, circles). Multiple properties of neurons change during development: the NMDAr switches its dominant subunit expression from GluN2B to GluN2A (*Sheng et al., 1994*; *Popescu et al., 2004*; *Iacobucci and Popescu, 2017*), the reversal potential of the receptor (GABAr) switches from depolarising to hyperpolarizing (*Rivera et al., 1999*; *Meredith et al., 2003*; *Rinetti-Vargas et al., 2017*), and the action potential backpropagates more efficiently with age (*Buchanan and Mellor, 2007*). These mechanisms have been proposed to underlie the developmental changes in synaptic plasticity rules because they are key regulators of synaptic calcium signalling (*Meredith et al., 2003*; *Buchanan and Mellor, 2007*). However, their sufficiency and individual contributions to the age-related plasticity changes are unclear and have not been taken into account in any previous model. We incorporated these mechanisms in the model (*Materials and methods*) by parametrizing each of the three components to vary with the animal's postnatal age, to test if they could account for the age-dependent plasticity data.

We found that elaborating the model with age-dependent changes in NMDAr composition, GABAr reversal potential, and BaP efficiency, while keeping the same plasticity readout parameters, was sufficient to account for the developmental changes in LTD and LTP observed by *Dudek and Bear, 1993* (*Figure 5a and b*). We then explored the model's response to protocols of various stimulation frequencies, from 0.5 to 40 Hz, across ages from P5 to P80 (*Figure 5c and e*). *Figure 5c* shows the synaptic strength change as function of stimulation frequency for ages P15, P25, P35, and P45. The magnitude of LTD decreases with age, while the magnitude of LTP increases with age. *Figure 5e* shows a contour plot of the same result, covering the age-frequency space.

The 1 Hz presynaptic stimulation protocol in *Dudek and Bear, 1993* did not induce LTD in adult animals (*Dudek and Bear, 1992*). We found that the joint CaN-CaMKII activity trajectories for this stimulation protocol underwent an age-dependent leftward shift beyond the LTD region (*Figure 5f*). This implies that LTD is not induced in mature animals by this conventional LFS protocol due to insufficient activation of enzymes. In contrast, *Tigaret et al., 2016* and *Isaac et al., 2009* were able to induce LTD in adult rat tissue by combining LFS with presynaptic spike pairs repeated 900 times at 3 Hz. Given these empirical findings and our modelling results, we observe that LTD induction in adult animals requires that the stimulation protocol: (1) causes CaMKII and CaN activity to stay more in the LTD region than the LTP region and (2) is sufficiently long to activate the LTD readout mechanism. With experimental parameters used by *Dudek and Bear, 1993*, this may be as short as 300 pulses when multi-spike presynaptic protocols are used since the joint CaMKII-CaN activity can reach the LTD region more quickly than for single spike protocols. We simulated two such potential protocols as predictions: doublet and quadruplet spike groups delivered 300 times at 1 Hz, with 50 ms between each pair of spikes in the group (*Figure 5d*). The model predicts that both these protocols induce LTD in adults, whereas as shown above, the single pulse protocol did not cause LTD. These simulations suggest that the temporal requirements for inducing LTD may not be as prolonged as previously assumed, since they can be reduced by varying stimulation intensity. See *Figure 5—figure supplement 1* for frequency versus age maps for presynaptic bursts.

*Dudek and Bear, 1993* also performed theta burst stimulation (*Appendix 1—table 1*) at different developmental ages, and found that LTP is not easily induced in young rats (*Cao and Harris, 2012*), as depicted in *Figure 5b*. The model qualitatively matches this trend, and also predicts that TBS induces maximal LTP around P21, before declining further during development (*Figure 5b*, green curve). Similarly, the model predicts that high-frequency stimulation induces LTP only for ages >P15, peaks at P35, then gradually declines at older ages (*Figure 5e*). Note that in *Figure 5b*, we used six epochs instead of four used by *Dudek and Bear, 1993* to increase LTP outcome which is known to washout after one hour for young rats (*Cao and Harris, 2012*).

In contrast to *Dudek and Bear, 1993* findings, other studies have found that LTP can be induced in hippocampus in young animals (<P15) with STDP. For example, *Meredith et al., 2003* found that, at room temperature, 1Pre1Post10 induces LTP in young rats, whereas 1Pre2Post10 induces NC. This

relationship was inverted for adults, with 1Pre1Post inducing no plasticity and 1Pre2Post10 inducing LTP (as captured by our model in *Figure 5—figure supplement 1*).

Together, these results suggest that not only do the requirements for LTP/LTD change with age, but also that these age-dependencies are different for different stimulation patterns. Finally, we explore which mechanisms are responsible for plasticity induction changes across development in the FDP protocol (*Figure 5—figure supplement 1*) by fixing each parameter to young or adult values for the FDP paradigm. Our model analysis suggests that the NMDAr switch (*Iacobucci and Popescu, 2017*) is a dominant factor affecting LTD induction, but the maturation of BaP (*Buchanan and Mellor, 2007*) is the dominant factor affecting LTP induction, with GABAr shift having only a weak influence on LTD induction for *Dudek and Bear, 1993* FDP.

Plasticity requirements during development do not necessarily follow the profile in *Dudek and Bear, 1993* as shown by *Meredith et al., 2003* STDP experiment. Our model suggests that multiple developmental profiles are possible when experimental conditions vary within the same stimulation paradigm. This is illustrated in *Figure 6—figure supplement 2a–c* by varying the age of STDP experiments done in different conditions. We fitted well the data from *Wittenberg and Wang, 2006* by adapting the model with appropriate age and temperature.

## Effects of extracellular calcium and magnesium concentrations on plasticity outcome

The canonical STDP rule (*Bi and Poo, 1998*), measured in cultured neurons with high extracellular calcium ($[Ca^{2+}]_o$) and at room temperature, was recently found not to be reproducible at physiological $[Ca^{2+}]_o$ in CA1 brain slices (*Inglebert et al., 2020*). Instead, by varying the $[Ca^{2+}]_o$ and $[Mg^{2+}]_o$, *Inglebert et al., 2020* found a spectrum of STDP rules with either no plasticity or full-LTD for physiological $[Ca^{2+}]_o$ conditions ($[Ca^{2+}]_o < 1.8$ mM) and a bidirectional rule for high $[Ca^{2+}]_o$ ($[Ca^{2+}]_o > 2.5$ mM), shown in *Figure 6a-c*.

We attempted to reproduce *Inglebert et al., 2020* findings by varying $[Ca^{2+}]_o$ and $[Mg^{2+}]_o$ with the following consequences for the model mechanisms (*Materials and methods*). On the presynaptic side, $[Ca^{2+}]_o$ modulates vesicle release probability. On the postsynaptic side, high $[Ca^{2+}]_o$ reduces NMDAr conductance (*Maki and Popescu, 2014*), whereas $[Mg^{2+}]_o$ affects the NMDAr $Mg^{2+}$ block (*Jahr and Stevens, 1990*). Furthermore, spine calcium influx activates SK channels, which hyperpolarize the membrane and indirectly modulate NMDAr activity (*Ngo-Anh et al., 2005*; *Griffith et al., 2016*).

*Figure 6a–c* compares our model to *Inglebert et al., 2020* STDP data at different $[Ca^{2+}]_o$ and $[Mg^{2+}]_o$. Note that *Inglebert et al., 2020* used 150 pairing repetitions for the anti-causal stimuli and 100 pairing repetitions for the causal stimuli both delivered at 0.3 Hz. At $[Ca^{2+}]_o=1.3$ mM, *Figure 6a* shows that the STDP rule induced weak LTD for brief causal delays. At $[Ca^{2+}]_o = 1.8$ mM, in *Figure 6b*, the model predicted a full-LTD window. At $[Ca^{2+}]_o = 3$ mM, in *Figure 6c*, it predicts a bidirectional rule with a second LTD window for long causal pairings, previously theorized by *Rubin et al., 2005*.

*Figure 6d* illustrates the time spent by the joint CaN-CaMKII activity for 1Pre1Post10 using *Inglebert et al., 2020* experimental conditions. Each density plot corresponds to a specific Ca/Mg ratio as in *Figure 6a–c*. The response under low $[Ca^{2+}]_o$ spent most time inside the LTD region, but high $[Ca^{2+}]_o$ shifts the trajectory to the LTP region. *Figure 6—figure supplement 1a* presents density plots for the anti-causal protocols.

*Inglebert et al., 2020* fixed the Ca/Mg ratio at 1.5, although aCSF formulations in the literature differ (see *Appendix 1—table 1*). *Figure 6—figure supplement 1d* shows that varying the Ca/Mg ratio and $[Ca^{2+}]_o$ for *Inglebert et al., 2020* experiments restrict LTP to Ca/Mg >1.5 and $[Ca^{2+}]_o$ >1.8 mM.

*Figure 6e* shows a map of plasticity as function of pre-post delay and Ca/Mg concentrations and the parameters where LTP is induced for the 1Pre1Post10 protocol. Since plasticity rises steeply at around $[Ca^{2+}]_o = 2.2$ mM (see *Figure 6—figure supplement 1b*), small fluctuations in $[Ca^{2+}]_o$ near this boundary could cause qualitative transitions in plasticity outcomes. For anti-causal pairings, increasing $[Ca^{2+}]_o$ increases the magnitude of LTD (*Figure 6—figure supplement 1b* illustrates this with *Inglebert et al., 2020* data). Our model can identify the transitions between LTD and LTP as a function of the ratio between $[Ca^{2+}]_o$ and $[Mg^{2+}]_o$, see *Figure 6—figure supplement 1*.

*Inglebert et al., 2020* also found that increasing the pairing frequency to 5 or 10 Hz results in a transition from LTD to LTP for 1Pre1Post10 at $[Ca^{2+}]_o = 1.8$ mM (*Figure 6—figure supplement*

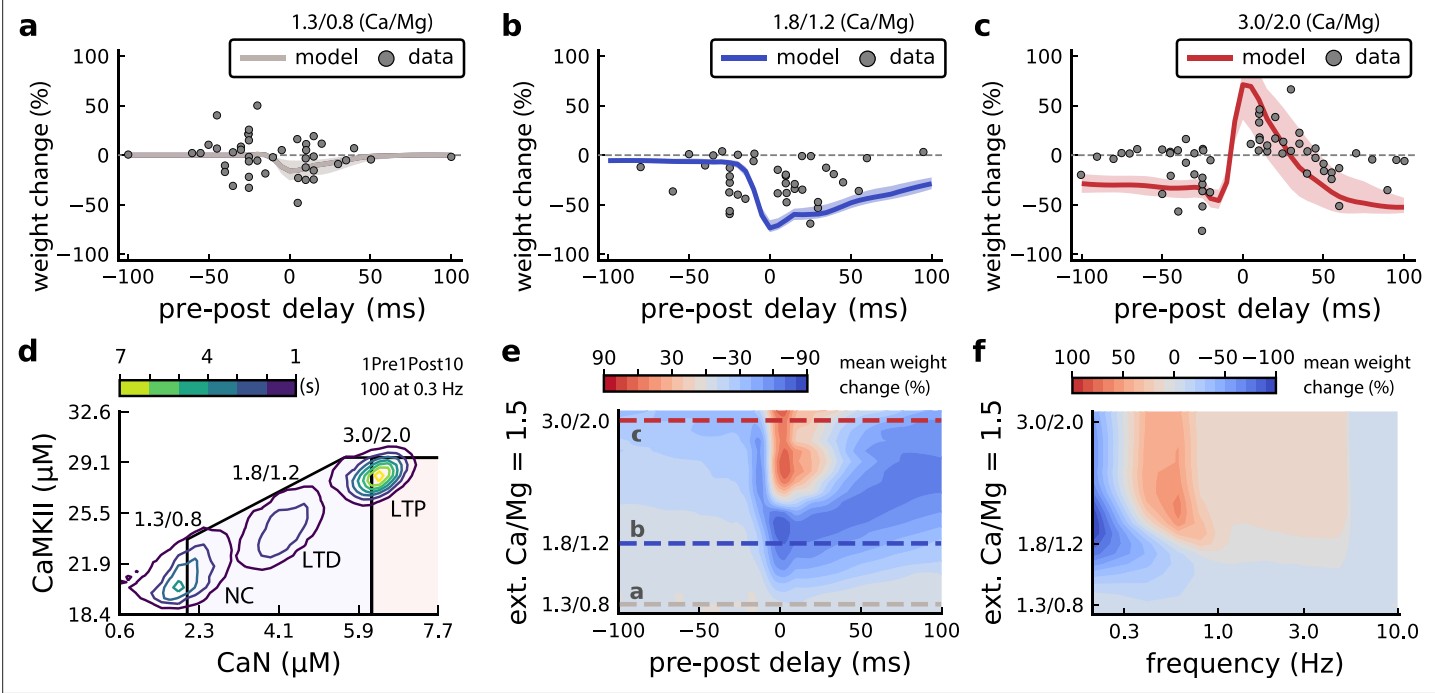

**Figure 6.** Effects of extracellular calcium and magnesium concentrations on plasticity. (**a**), Synaptic weight (%) for a STDP rule with $[Ca^{2+}]_o$=1.3 mM (fixed ratio, Ca/Mg = 1.5). According to the data extracted from *Inglebert et al., 2020*, the number of pairing repetitions for causal/positive (anti-causal/negative) delays is 100 (150), both delivered at 0.3 Hz. The solid line is the mean, and the ribbons are the 2nd and 4th quantiles predicted by our model (all panels use 100 samples). (**b**), Same as a, but for $[Ca^{2+}]_o$ = 1.8 mM (Ca/Mg ratio = 1.5). (**c**), Same as a, but for $[Ca^{2+}]_o$ = 3 mM (Ca/Mg ratio = 1.5). (**d**), Mean time spent for causal pairing, 1Pre1Post10, at different Ca/Mg concentration ratios. The contour plots are associated with the panels **a, b and c. e**, Predicted effects of extracellular Ca/Mg on STDP outcome. Synaptic weight change (%) for causal (1Pre1Post10, 100 at 0.3 Hz) and anticausal (1Post1Pre10, 150 at 0.3 Hz) pairings varying extracellular Ca from 1.0 to 3 mM (Ca/Mg ratio = 1.5). The dashed lines represent the experiments in the panel **a, b and c**. We used 21x22 × 100 data points, respectively calcium x delay x samples. (**f**), Predicted effects of varying frequency and extracellular Ca/Mg for an STDP protocol. Contour plot showing the mean synaptic weight (%) for a single causal pairing protocol (1Pre1Post10, 100 samples) varying frequency from 0.1 to 10 Hz and $[Ca^{2+}]_o$ from 1.0 to 3 mM (Ca/Mg ratio = 1.5). We used 21 x 18 × 100 data points, respectively calcium x frequency x samples.

The online version of this article includes the following figure supplement(s) for figure 6:

**Figure supplement 1.** $[Ca^{2+}]_o$ and $[Mg^{2+}]_o$ related modifications for *Inglebert et al., 2020* experiment.

**Figure supplement 2.** Temperature and age effects.

---

*1c*), similar frequency-STDP behaviour has been reported in the cortex (*Sjöström et al., 2001*). In *Figure 6f*, we varied both the pairing frequencies and $[Ca^{2+}]_o$ and we observe similar transitions to *Inglebert et al., 2020*. However, the model's transition for $[Ca^{2+}]_o$ = 1.8 mM was centred around 0.5 Hz, which was untested by *Inglebert et al., 2020*. The model predicts no plasticity at higher frequencies, unlike the data, that shows scattered LTP and LTD (see *Figure 6—figure supplement 1c*). Another frequency dependent comparison, *Figure 3—figure supplement 4c* and *Figure 6—figure supplement 1h*, show that *Tigaret et al., 2016* burst-STDP and *Inglebert et al., 2020* STDP share a similar transition structure, different from *Dudek and Bear, 1992* FDP.

In contrast to *Inglebert et al., 2020* results, the model predicts that setting low $[Ca^{2+}]_o$ for *Tigaret et al., 2016* burst-STDP abolishes LTP, and does not induce strong LTD (*Figure 3—figure supplement 4d*). For *Dudek and Bear, 1992* experiment, *Figure 4—figure supplement 1a* $[Mg^{2+}]_o$ controls a sliding threshold between LTD and LTP but not $[Ca^{2+}]_o$ (*Figure 4—figure supplement 1b*). For another direct stimulation experiment, *Figure 6—figure supplement 1f* shows that in an Mg-free medium, LTP expression requires fewer pulses (*Mizuno et al., 2001*).

Despite exploring physiological $[Ca^{2+}]_o$ and $[Mg^{2+}]_o$ Inglebert (*Inglebert et al., 2020*) use a non-physiological temperature ($30°C$) which extends T-type VGCC closing times and modifies the CaN-CaMKII baseline (*Figure 6—figure supplement 2f*). In summary, our model predicts that temperature can change STDP rules in a similar fashion to $[Ca^{2+}]_o$ (*Figure 6—figure supplement 1a and b*). Overall,

we confirm that plasticity is highly sensitive to variations in extracellular calcium, magnesium, and temperature (*Figure 3—figure supplement 4a*, *Figure 6—figure supplement 2d–f*).

## In vivo-like spike variability affects plasticity

In the above sections, we used highly regular and stereotypical stimulation protocols to replicate typical ex vivo plasticity experiments. In contrast, neural spiking in hippocampus in vivo is irregular and variable (*Fenton and Muller, 1998*; *Isaac et al., 2009*). Previous studies that asked how natural firing variability affects the rules of plasticity induction used simpler synapse models (*Rackham et al., 2010*; *Graupner et al., 2016*; *Cui et al., 2018*). We explored this question in our synapse model using simulations with three distinct types of additional variability: (1) spike time jitter, (2) failures induced by dropping spikes, (3) independent pre and postsynaptic Poisson spike trains (*Graupner et al., 2016*).

We introduced spike timing jitter by adding zero-mean Gaussian noise (s.d. $\sigma$) to pre and postsynaptic spikes, changing spike pairs inter-stimulus interval (ISI). In *Figure 7a*, we plot the LTP magnitude as function of jitter magnitude (controlled by $\sigma$) for protocols taken from *Tigaret et al., 2016*. With no jitter, $\sigma = 0$, these protocols have different LTP magnitudes (corresponding to *Figure 3*) and become similar once $\sigma$ increases. The three protocols with a postsynaptic spike doublet gave identical plasticity for $\sigma = 50$ ms.

To understand the effects of jittering, we plotted the trajectories of joint CaN-CaMKII activity (*Figure 7c*). 2Post1Pre50 which 'undershoots' the LTP region shifted into the LTP region for jitter $\sigma = 50$ ms. In contrast, 1Pre1Post10 which 'overshoots' (mostly smaller CaN concentration) the LTP region shifted to the opposite direction towards the LTP region.

Why does jitter cause different spike timing protocols to yield similar plasticity magnitudes? Increasing jitter causes a fraction of pairings to invert causality. Therefore, the jittered protocols became a mixture of causal and anticausal pairings (*Figure 7c*). This situation occurs for all paired protocols. So any protocol with the same number spikes will produce a similar outcome if the jitter is large enough. Note that despite noise the mean frequency was conserved at 5 ±13.5 Hz (see *Figure 7e*).

Next, we studied the effect of spike removal. In the previous sections, synaptic release probability was ~60% (for $[Ca^{2+}]_o = 2.5$ mM) or lower, depending on the availability of docked vesicles (*Materials and methods*). However, baseline presynaptic vesicle release probability is heterogeneous across CA3-CA1 synapses, ranging from $\sim 10 - 90\%$ (*Dobrunz et al., 1997*; *Enoki et al., 2009*) and likely lower on average in vivo (*Froemke and Dan, 2002*; *Borst, 2010*). BaPs are also heterogeneous with random attenuation profiles (*Golding et al., 2001*) and spike failures (*Short et al., 2017*). To test the effects of pre and postsynaptic failures on plasticity induction, we performed simulations where we randomly removed spikes, altering the regular attenuation observed in *Tigaret et al., 2016* protocols.

In *Figure 7b*, we plot the plasticity magnitude as function of sparsity (percentage of removed spikes). The sparsity had different specific effects for each protocol. 1Pre2Post10 and 1Pre2Post50 which originally produced substantial LTP were robust to spike removal until ~60% sparsity. In contrast, the plasticity magnitude from both 1Pre1Post10 and 2Post1Pre50 showed a non-monotonic dependence on sparsity, first increasing then decreasing, with maximal LTP at ~40% sparsity.

To understand how sparsity causes this non-monotonic effect on plasticity magnitude, we plotted the histograms of time spent in the CaN-CaMKII plane for 2Post1Pre50 for three levels of sparsity: 0%, 30%, and 80% (*Figure 7d*). For 0% sparsity, the activation spent most time at the border between the LTP and LTD regions, resulting in no change. Increasing sparsity to 30% caused the activation to shift rightward into the LTP region because there was less attenuation of pre and postsynaptic resources. In contrast, at 80% sparsity, the activation moved into the LTD region because there were not enough events to substantially activate CaMKII and CaN. Since LTD is a slow process and the protocol duration is short (60 s), there was no net plasticity. Therefore for this protocol, high and low sparsity caused no plasticity for distinct reasons, whereas intermediate sparsity enabled LTP by balancing resource depletion with enzyme activation.

Next we tested the interaction of jitter and spike removal. *Figure 7f* shows a contour map of weight change as a function of jitter and sparsity for the 2Post1Pre50 protocol, which originally induced no plasticity (*Figure 3*). Increasing spike jitter enlarged the range of sparsity inducing LTP. In summary, these simulations (*Figure 7a, b, f and h*) show that different STDP protocols have different degrees of

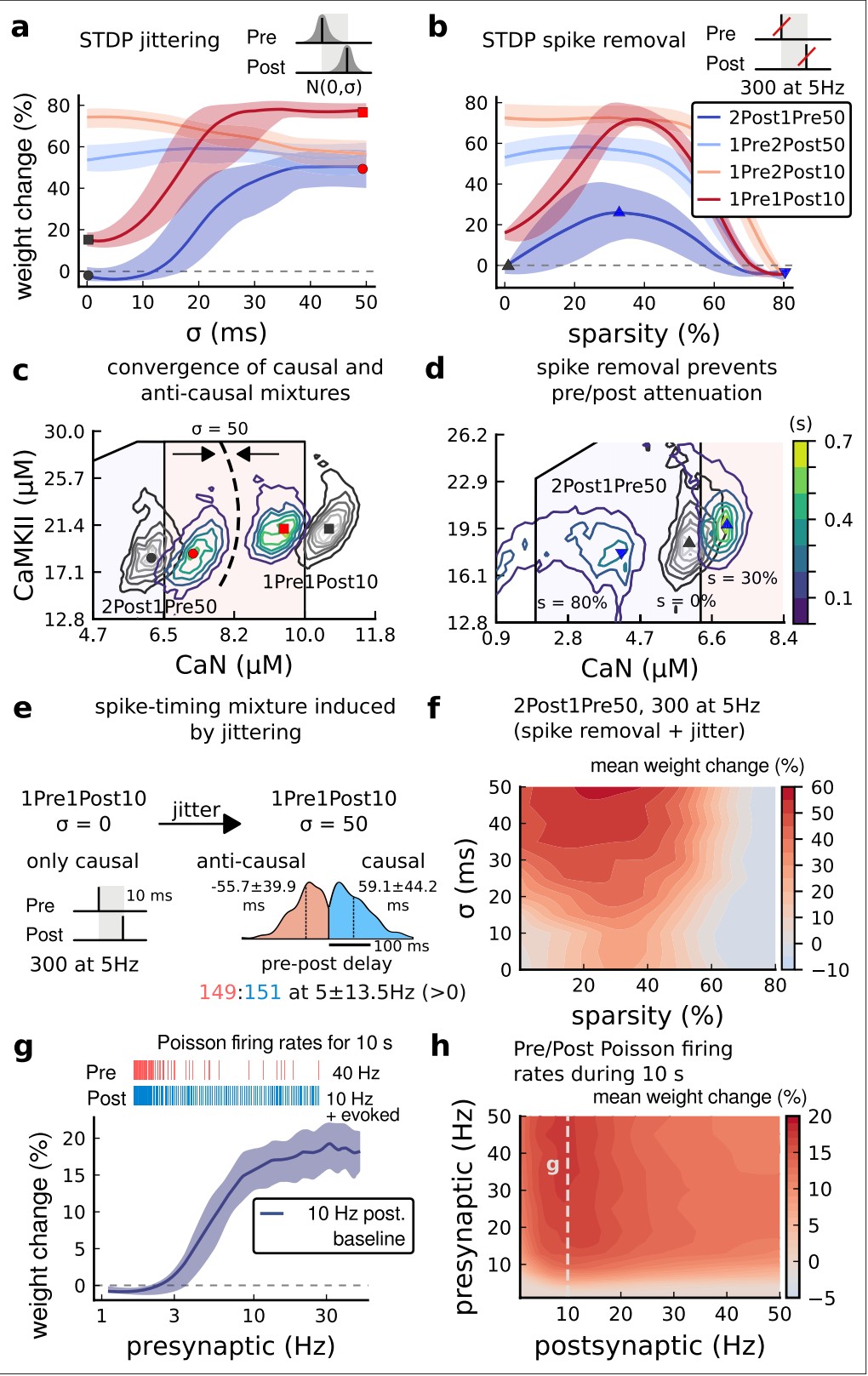

**Figure 7.** Jitter and spike dropping effects on STDP and Poisson spike trains. (**a**) Mean weight (%) for the jittered STDP protocols (protocol color legend shown in **b**). The solid line is the mean, and the ribbons are the 2nd and 4th quantiles predicted by our model using 100 samples (same for panels **a, b** and **g**). (**b**) Mean weight (%) for the same (*Tigaret et al., 2016*) protocols used in panel a subjected to random spike removal (sparsity %). (**c**)

*Figure 7 continued on next page*

*Figure 7 continued*

Effect of jitttering on Mean time (s) spent by joint enzymes trajectories in LTP/LTD regions. Contour plot shows 2Post1Pre50 and 1Pre1Post10 (300 at 5 Hz) without (grey contour plot) and with jittering (coloured contour plot). The circles and squares correspond to the marks in panel **a**. (**d**) Effect of sparsity on Mean time (s) spent by joint enzymes trajectories in LTP/LTD regions. Contour plot in grey showing 0% sparsity for 2Post1Pre50 300 at 5 Hz (see *Figure 2j*). The contour plots show the protocol with spike removal sparsities at 0% (NC), 30% (LTP), and 80% (NC). The triangles correspond to the same marks in panel **a**. (**e**) Distribution of the 50 ms jittering applied to the causal protocol 1Pre1Post10, 300 at 5 Hz in which nearly half of the pairs turned into anticausal. The mean frequency is 5 ±13.5 Hz making it to have a similar firing structure and position in the LTP region. The similar occurs for 2Post1Pre50 (panel **c**). (**f**) Mean weight change (%) combining both jittering (panel a) and sparsity (panel **b**) for 2Post1Pre50, 300 at 5 Hz. (**g**) Mean weight change (%) of pre and postsynaptic Poisson spike train delivered simultaneously for 10 s. The plot shows the plasticity outcome for different presynaptic firing rate (1000/frequency) for a fixed postsynaptic baseline at 10 Hz. The upper raster plot depicts the released vesicles at 40 Hz and the postsynaptic baseline at 10 Hz (including the AP evoked by EPSP). (**h**) Mean weight change (%) varying the rate of pre and postsynaptic Poisson spike train delivered simultaneously for 10 s. The heat map data along the vertical white dashed line is depicted in panel **g**.

sensitivity to noise in the firing structure, suggesting that simple plasticity rules derived from regular ex vivo experiments may not predict plasticity in vivo.

How does random spike timing affect rate-dependent plasticity? We stimulated the model with pre and postsynaptic Poisson spike trains for 10 s, under *Dudek and Bear, 1992* experimental conditions. We systematically varied both the pre and postsynaptic rates (*Figure 7h*). The 10 s stimulation protocols induced only LTP, since LTD requires a prolonged stimulation (*Mizuno et al., 2001*). LTP magnitude monotonically increased with the presynaptic rate (*Figure 7g and h*). In contrast, LTP magnitude varied non-monotonically as a function of postsynaptic rate, initially increasing until a peak at 10 Hz, then decreasing with higher stimulation frequencies. This non-monotonic dependence on post-synaptic rate is inconsistent with classic rate-based models of Hebbian plasticity. From this analysis, we can make the prediction that firing variability can alter the rules of plasticity, in the sense that it is possible to add noise to cause LTP for protocols that did not otherwise induce plasticity. For example, we show that protocols inducing LTP can be hindered by jittering, *e.g.* 1Pre2Post10 in *Figure 7a*. Also, protocols that are just outside the LTP plasticity region may turn into LTP if jitter is applied, *e.g.* 2Post1Pre50 and 1Pre1Post10 in *Figure 7a*. We also investigated how this plasticity dependence on pre- and postsynaptic Poisson firing rates varies with developmental age (*Figure 4— figure supplement 1g–i*). We found that at P5 no plasticity is induced, at P15 a LTP region appears at around 1 Hz postsynaptic rate, and at P20 plasticity becomes similar to the mature age, with a peak in LTP magnitude at 10 Hz postsynaptic rate.

## Discussion

We built a model of a rat CA3-CA1 hippocampal synapse, including key electrical and biochemical components underlying synaptic plasticity induction (*Figure 1*). We developed a novel geometric readout of combined CaN-CaMKII dynamics (*Figures 2–4*) to predict the outcomes from a range of plasticity experiments with heterogeneous conditions: animal developmental age (*Figure 5*), aCSF composition (*Figure 6*), temperature (Supplemental files), and in vivo-like firing variability (*Figure 7*). This readout provides a simple and intuitive window into the dynamics of the synapse during plasticity. Our model is thus based on the joint activity of these two key postsynaptic enzymes at both fast and slow time scales and considers the stochastic dynamics of their activities dictated by the upstream calcium-dependent components at both the pre- and postsynapse. On this basis alone, our model is akin to biological processes where the outcome is jointly determined by several stochastic signalling components and a combination of multiple enzyme activities, that is, are multi-dimensional. The principal assumption underlying the proposed 'geometric readout' mechanism is that all information determining the induction of LTP *vs.* LTD is contained in the time-dependent spine $Ca^{2+}$/calmodulin-bound CaN and CaMKII concentrations, and that no extra elements are required. Further, since both CaN and CaMKII concentrations are uniquely determined by the time course of postsynaptic $Ca^{2+}$ concentration, the model implicitly assumes that the LTP/LTD induction depends solely on spine $Ca^{2+}$ concentration time course, as in previously compared simplified models (see Introduction).

In addition to providing a new model of CA3-CA1 synapse biophysics, the main contribution of this work is the novel readout mechanism mapping synaptic enzymes to plasticity outcomes. This readout was built based on the concept that the full temporal activity of CaN-CaMKII over the minutes-timescale stimulus duration, and not their instantaneous levels, is responsible for changes in synaptic efficacy (*Fujii et al., 2013*). We instantiated this concept by analysing the joint CaN-CaMKII activity in the two-dimensional plane and designing polygonal plasticity readout regions (*Figure 3a*). Here, we used only a two-dimensional readout, but anticipate a straightforward generalisation to higher dimensions. The central discovery is that these trajectories, despite being stochastic, can be separated in the plane as a function of the stimulus (*Figure 3*). This is the basis of our new synaptic plasticity rule. We generalised previous work with plasticity induction based on single threshold and a slow variable (*Badoual et al., 2006*; *Rubin et al., 2005*; *Clopath and Gerstner, 2010*; *Graupner and Brunel, 2012*). In contrast, previous models assume that plasticity is explainable in terms of synaptic calcium or enzyme response to single BAP-EPSP pairings (*Shouval et al., 2002*; *Karmarkar and Buonomano, 2002*). We expect that future studies using high temporal resolution measurements such as those provided by recent FRET tools available for CaMKII (*Chang et al., 2017*; *Chang et al., 2019*) will bring refinements to our model with the possibility to further test our readout predictions.

Let us describe the intuition behind our model more concisely. First, we abstracted away the sophisticated cascade of plasticity expression. Second, the plasticity regions, crossed by the trajectories, are described with a minimal set of parameters. Importantly, their tuning is quite straightforward and done only once, even when the joint activity is stochastic. The tuning of the model is possible thanks to the decoupling of the plasticity process from the spine biophysics which acts as a feedforward input to the plasticity Markov chain and from the distributions of the different trajectories, which are well separated. The separability afforded by the geometrical readout, along with the model flexibility via fitting the plasticity regions, enabled us to reproduce data from nine different experiments using a single fixed set of model parameters. In contrast, we found that classic spike-timing (*Song et al., 2000*; *Pfister and Gerstner, 2006*) or calcium-threshold (*Graupner and Brunel, 2012*) models could not reproduce the range of protocols from *Tigaret et al., 2016* (*Figure 2b–d*). More complicated molecular-cascade models have been shown to account for individual plasticity experiments (*Antunes et al., 2016*; *Jędrzejewska-Szmek et al., 2017*; *Mäki-Marttunen et al., 2020*; *Bhalla, 2017*), but have not been demonstrated to reproduce the wide range of protocols presented here while considering experimental heterogeneity.

For some protocols, the CaMKII-CaN trajectories overshot the plasticity regions (e.g. *Figure 3d*). Although abnormally high and prolonged calcium influx to cells can trigger cell death (*Zhivotovsky and Orrenius, 2011*), the effects of high calcium concentrations at single synapses are poorly understood. Notably, a few studies have reported evidence consistent with an overshoot, where strong synaptic calcium influx does not induce LTP (*Yang et al., 1999*; *Tigaret et al., 2016*; *Pousinha et al., 2017*).

Our model included critical components for plasticity induction at CA3-CA1 synapses: those affecting dendritic spine voltage, calcium signalling, and enzymatic activation. We were able to use our model to make quantitative predictions, because its variables and parameters corresponded to biological components. This property allowed us to incorporate the model components' dependence on developmental age, external Ca/Mg levels, and temperature to replicate datasets across a range of experimental conditions. The model is relatively fast to simulate, taking ~1 min of CPU time to run 1 min of biological time. These practical benefits should enable future studies to make experimental predictions on dendritic integration of multiple synaptic inputs (*Blackwell et al., 2019*; *Oliveira et al., 2012*; *Ebner et al., 2019*) and on the effects of synaptic molecular alterations in pathological conditions. In contrast, abstract models based on spike timing (*Song et al., 2000*; *Pfister and Gerstner, 2006*; *Clopath and Gerstner, 2010*) or simplified calcium dynamics (*Shouval et al., 2002*; *Graupner and Brunel, 2012*) must rely on ad hoc adjustment of parameters with less biological interpretability.

Intrinsic noise is an essential component of the model. How can the synapse reliably express plasticity but be noisy at the same time (*Yuste et al., 1999*; *Ribrault et al., 2011*)? Noise can be reduced either by redundancy or by averaging across time, also called ergodicity (*Sterling and Laughlin, 2015*). However, redundancy requires manufacturing and maintaining more components, and therefore costs energy. We propose that, instead, plasticity induction is robust due to temporal averaging by slow-timescale signalling and adaptation processes. These slow variables display reduced noise

levels by averaging the faster timescale stochastic variables. This may be a reason why CaMKII uses auto-phosphorylation to sustain its activity and slow its decay time (*Chang et al., 2017*; *Chang et al., 2019*). In summary, this suggests that the temporal averaging by slow variables, combined with the separability afforded by the multidimensional readout, allows synapses to tolerate noise while remaining energy-efficient.

A uniqueness of our model is that it simultaneously incorporates biological variables such as electrical components at pre and postsynaptic sites, some with adaptive functions such as attenuation, age and temperature, stochastic noise and fast and slow timescales. Some of these variables have been modelled by other groups, for example stochasticity, BaP attenuation or pre-synaptic plasticity (*Cai et al., 2007*; *Shouval and Kalantzis, 2005*; *Zeng and Holmes, 2010*; *Miller et al., 2005*; *Yeung et al., 2004*; *Shah et al., 2006*; *Deperrois and Graupner, 2020*; *Costa et al., 2015*), but generally independently from each other. To position the uniqueness of our model in this broader context, we also provide a direct comparison of our model with some of the most recent leading models of excitatory synapse plasticity and the experimental work they reproduce (*Appendix 1—table 2* and *Appendix 1—table 3*).

We identified some limitations of the model. First, we modelled only a single postsynaptic spine attached to a two-compartment neuron (soma and dendrite). Second, the model abstracted the complicated process of synaptic plasticity expression. Indeed, even if this replicated the early phase of LTP/LTD expression in the first 30–60 min after induction, we did not take into account slower protein-synthesis-dependent processes, maintenance processes, and synaptic pruning proceed at later timescales (*Bailey et al., 2015*). Third, like most biophysical models, ours contained many parameters. Although we set these to physiologically plausible values and then tuned to match the plasticity data, other combinations of parameters may fit the data equally well (*Marder and Taylor, 2011*; *Mäki-Marttunen et al., 2020*) due to the ubiquitous phenomenon of redundancy in biochemical and neural systems (*Gutenkunst et al., 2007*; *Marder, 2011*). Indeed synapses are quite heterogeneous in receptor and ion channel counts (*Takumi et al., 1999*; *Sabatini and Svoboda, 2000*; *Racca et al., 2000*; *Nimchinsky et al., 2004*), protein abundances (*Shepherd and Harris, 1998*; *Sugiyama et al., 2005*), and spine morphologies (*Bartol et al., 2015*; *Harris and Stevens, 1989*), even within the subpopulation of CA1 pyramidal neuron synapses that we modelled here. Fourth, the activation of clustered synapses could influence the plasticity outcome, and the number of synapses activated during plasticity induction can be difficult to control experimentally. Our model concerns plasticity at a single synapse, which is also important during synaptic cluster activation (*Ujfalussy and Makara, 2020*). We drew from data in *Tigaret et al., 2016* where there is little indication of simultaneous clustered synaptic activation. Furthermore, our simulations are in good agreement with plasticity experiments using local field potential recordings (*Dudek and Bear, 1993*) where the number of activated synapses is uncertain. This indicates that the model proposed here can account for this aspect of synaptic plasticity heterogeneity. Finally, our readout model does not correspond to a specific molecular cascade beyond CaN and CaMKII activations. However, we anticipate that the same mapping could be implemented by simple biochemical reaction networks, with for example, transition rates based on Hill functions for the plasticity boundaries.

Since the model respected the stochasticity of vesicle release (*Rizzoli and Betz, 2005*; *Alabi and Tsien, 2012*), NMDAr (*Nimchinsky et al., 2004*; *Popescu et al., 2004*; *Iacobucci and Popescu, 2017*; *Sinclair et al., 2016*), and VGCC opening (*Magee and Johnston, 1995*; *Sabatini and Svoboda, 2000*; *Iftinca et al., 2006*), the magnitude of plasticity varied from simulation trial to trial. This suggests that the rules of plasticity are inherently stochastic (*Bhalla, 2004*; *Antunes et al., 2016*) and that the variability observed in these experiments (*Inglebert et al., 2020*; *Tigaret et al., 2016*; *Dudek and Bear, 1992*; *Dudek and Bear, 1993*; *Mizuno et al., 2001*; *Meredith et al., 2003*; *Wittenberg and Wang, 2006*) is partly due to stochastic signalling. With our current model, we have been able to reproduce nine experiments with a single set of parameters. By running extensive simulations over the space of protocols beyond those tested experimentally (*Figure 3h and i*; *Figure 4f*; *Figure 5c, e and f*; *Figure 6e and f*), we made testable predictions for plasticity outcomes that should therefore be of a high level of confidence. For example, *Tigaret et al., 2016* did not find LTD when using classic post-before-pre stimulation protocols, but the model predicted that LTD could be induced if the number of pairing repetitions was extended (*Figure 3h and i*). The model also predicts that the lack of LTD induced by FDP in adults can be recovered using doublets, triplets or quadruplets of spikes in the

protocols (*Figure 5d*). We tested the model's sensitivity to spike time jitter and spike failure in the stimulation protocols (*Figure 7*). Our simulations predict that this firing variability can alter the rules of plasticity, in the sense that it is possible to add noise to cause LTP for protocols that did not otherwise induce plasticity.

What do these results imply about the rules of plasticity in vivo? First, we noticed that successful LTP or LTD induction required a balance between two types of slow variables: those that attenuate, such as presynaptic vesicle pools and dendritic BaP, versus those that accumulate, such as slow enzymatic integration (*Cai et al., 2007*; *Mizusaki et al., 2022*; *Deperrois and Graupner, 2020*). This balance is reflected in the inverted-U shaped magnitude of LTP seen as a function of post-synaptic firing rate (*Figure 7h*). Second, although spike timing on millisecond timescales can in certain circumstances affect the direction and magnitude of plasticity (*Figure 3*), in order to drive sufficient activity of synaptic enzymes, these patterns would need to be repeated for several seconds. However, if these repetitions are subject to jitter or failures, as observed in hippocampal spike trains in vivo (*Fenton and Muller, 1998*; *Wierzynski et al., 2009*), then the millisecond-timescale information will be destroyed as it gets averaged out across repetitions by the slow integration processes of CaMKII and CaN (*Figure 7a–d*). The net implication is that millisecond-timescale structure of individual spike pairs is unlikely to play an important role in determining hippocampal synaptic plasticity in vivo (*Froemke and Dan, 2002*; *Sadowski et al., 2016*; *Graupner et al., 2016*).

In summary, we presented a new type of biophysical model for plasticity induction at the rat CA3-CA1 glutamatergic synapse. Although the model itself is specific to this synapse type, the study's insights may generalise to other synapse types, enabling a deeper understanding of the rules of synaptic plasticity and brain learning.

## Materials and methods

### Experimental datasets
The datasets at the basis of our model were obtained directly from the authors *Tigaret et al., 2016* or extracted from graphs in the references in *Appendix 1—table 2* using WebPlotDigitizer v 4.6 software (Rohatgi, A.). The dataset from *Tigaret et al., 2016* is freely available upon request.

### Code availability
All simulations were performed in the Julia programming language (version 1.4.2). This choice was dictated by simplicity and speed (*Perkel, 2019*). The code for the Markov chains is mostly automatically generated from reactions using the Julia package Catalyst.jl (*Loman et al., 2022*), and could be exported to an SBML representation for porting to other languages. The model is available on GitHub at SynapseElife (copy archived at *Veltz, 2023*).

Simulating the synapse model is equivalent to sampling a piecewise deterministic Markov process, and this relies on the thoroughly tested Julia package PiecewiseDeterministicMarkovProcesses.jl. These simulations are event-based, and no approximation is made beyond the ones required to integrate the ordinary differential equations by the LSODA method (Livermore Solver for Ordinary Differential Equations). We ran the parallel simulations in the Nef cluster operated by Inria.

### Notation
We write $\mathbf{1}_A$ for the indicator of a set $A$, meaning that $\mathbf{1}_A(x) = 1$ if $x$ belongs to $A$ and zero otherwise.

### Vesicle release and recycling
The detailed exocytosis model we implemented was motivated by taking into account the following minimal requirements: synaptic failures (which impact STDP protocols), vesicle depletion (for frequency/pulse number repetition dependent protocols), external calcium (motivated by *Inglebert et al., 2020*). This is best modeled by counting the released vesicles, hence our choice of a stochastic model. In biological synapses, an action potential arriving at the presynaptic terminal can trigger the release of a neurotransmitter–filled vesicle, which activates postsynaptic receptors. We derived a vesicle–release Markov chain model based on a deterministic approach described in *Sterratt et al., 2011*.

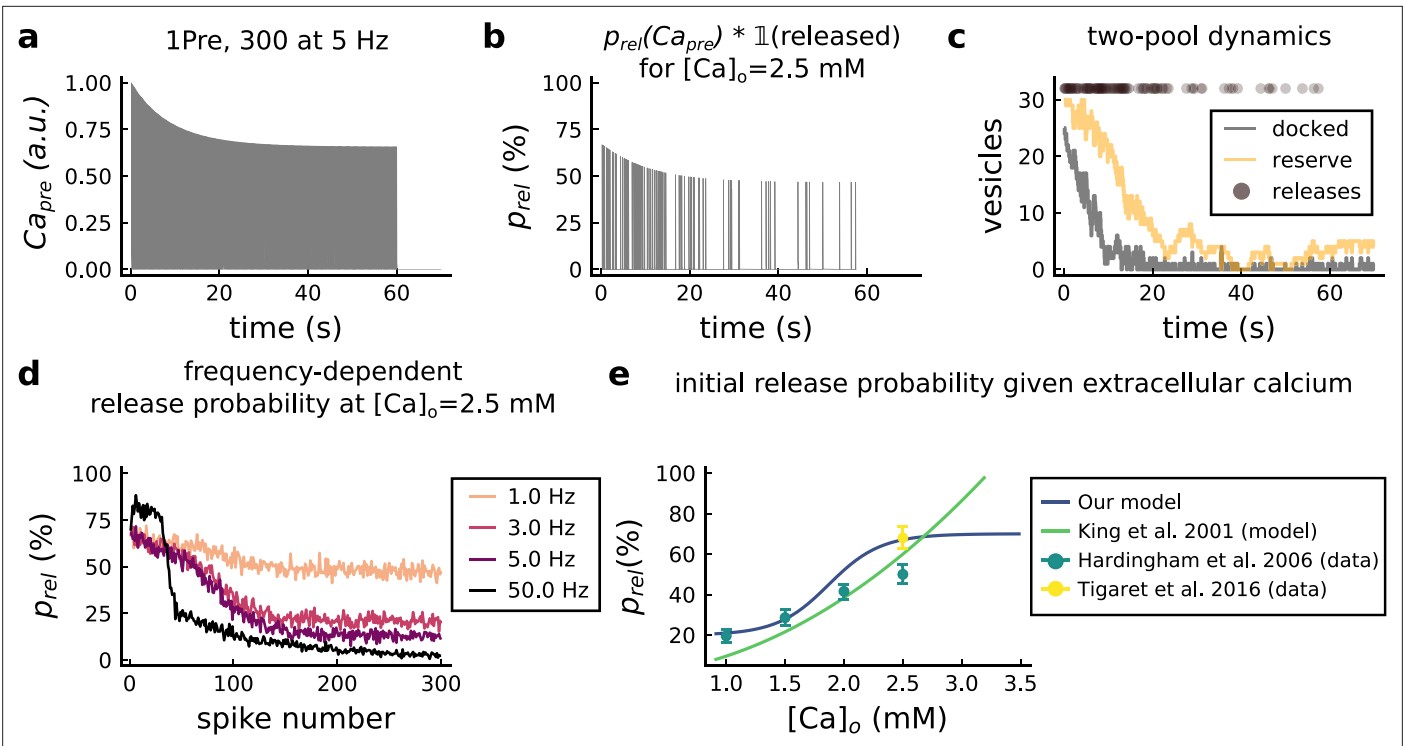

**Figure 8.** Presynaptic release model validation. (**a**) Presynaptic calcium in response to the protocol 1Pre, 300 pulses at 5 Hz displaying adaptation. (**b**) Release probability for the same protocol as panel A but subjected to the docked vesicles availability. (**c**) Number of vesicles in the docked and reserve pools under depletion caused by the stimulation from panel **a**. (**d**) Plot of the mean (300 samples) release probability (%) for different frequencies for the protocol 1Pre 300 pulses at $[Ca^{2+}]_o$ = 2.5 mM. (**e**) Release probability (%) for a single presynaptic spike as a function of $[Ca^{2+}]_o$. Note that **King et al., 2001** model was multiplied by the experimentally measured release probability at $[Ca^{2+}]_o$ = 2 mM since their model has this calcium concentration as the baseline.

Vesicles can be in two states, either belonging to the docked pool (with cardinal $D$) with fast emptying, or to the reserve pool (with cardinal $R$) which replenishes $D$ (**Rizzoli and Betz, 2005**). Initially the docked and reserve pools have $D_0$ and $R_0$ vesicles, respectively. The docked pool loses one vesicle each time a release occurs (**Rudolph et al., 2015**), with transition $D \to D - 1$ (**Figure 8**). The reserve pool replenishes the docked pool with reversible transition $(R, D) \leftrightarrow (R - 1, D + 1)$. Finally, the reserve pool is replenished towards initial value $R_0$ over a timescale $\tau_D^{ref}$ via the transition $(R, D) \to (R + 1, D)$.

In addition to the stochastic dynamics in **Table 1**, each spike triggers a vesicle release $D \to D - 1$ with probability $p_{rel}$:

$$910 p_{rel}([Ca]_{pre}, [Ca^{2+}]_o, D) = \frac{\left([Ca]_{pre}\right)^s}{\left([Ca]_{pre}\right)^s + h([Ca^{2+}]_o)^s} \mathbb{1}_{D>0}, \quad h([Ca^{2+}]_o) = 0.654 + \frac{1.349}{1 + e^{4 \cdot ([Ca^{2+}]_o - 1.708)}}$$

(1)

which is a function of presynaptic calcium $[Ca_{pre}]$ and extracellular calcium concentration, $[Ca^{2+}]_o$, through the threshold $h([Ca^{2+}]_o)$. The function $p_{rel}$ was fitted in **Figure 8e** against data from **Tigaret**

**Table 1.** Stochastic transitions used in the presynaptic vesicle pool dynamics.
Note that the rates depend on the number of vesicles in each pool (**Pyle et al., 2000**).

| Transition | Rate | Initial Condition |
|---|---|---|
| $(R, D) \to (R - 1, D + 1)$ | $(D_0 - D) \cdot R/\tau_D$ | $D(t = 0) = D_0$ |
| $(R, D) \to (R + 1, D - 1)$ | $(R_0 - R) \cdot D/\tau_R$ | $R(t = 0) = R_0$ |
| $(R, D) \longrightarrow (R + 1, D)$ | $(R_0 - R)/\tau_R^{ref}$ | |

**Table 2.** Parameter values for the presynaptic model.
Our model does not implement a larger pool of ~ 180 vesicles (CA3-CA1 hippocampus) sometimes named 'reserve pool' (*Südhof, 2000*) or 'resting pool' (*Alabi and Tsien, 2012*). Furthermore, what we term the 'reserve pool' in our model is sometimes called the 'recycling pool' in other studies, for example *Rizzoli and Betz, 2005*; *Alabi and Tsien, 2012*.

| Name | Value | Reference |
|---|---|---|
| **Vesicle release model (stochastic part)** | | |
| Initial number of vesicles at D | $D_0 = 25$ | 5–20 (*Rizzoli and Betz, 2005*; *Alabi and Tsien, 2012*) |
| Initial number of vesicles at R | $R_0 = 30$ | 17–20 vesicles (*Alabi and Tsien, 2012*) |
| Time constant R→ D (D recycling) | $\tau_D = 5\ s$ | 1 s (*Rizzoli and Betz, 2005*) |
| time constant D → R (R mixing) | $\tau_R = 45\ s$ | 20 s (when depleted) to 5 *min* (hypertonic shock) (*Rizzoli and Betz, 2005*; *Pyle et al., 2000*) |
| Time constant 1→ R (R recycling) | $\tau_R^{ref} = 40\ s$ | 20–30 s (*Rizzoli and Betz, 2005*) |
| Release probability half-activation curve | $h$ | see *Equation 1* |
| Release probability sigmoid slope | $s = 2$ | fixed for all [Ca²⁺]ₒ |
| **Vesicle release model (deterministic part)** | | |
| $Ca_{pre}$ attenuation recovery | $\tau_{pre} = 0.02\ s$ | 50-500 ms with dye (*Maravall et al., 2000*) therefore < 50 to 500 ms without dye |
| Deterministic jump attenuation recovery | $\tau_{rec} = 20\ s$ | ~20 s (*Rizzoli and Betz, 2005*) |
| Deterministic jump attenuation fraction | $\delta_{ca} = 0.0004$ | *Forsythe et al., 1998* |

*et al., 2016*; *Hardingham et al., 2006*. To decide whether a vesicle is released for a presynaptic spike, we use a phenomenological model of $Ca_{pre}$ (see *Figure 8a*) based on a resource-use function (*Tsodyks and Markram, 1997*):

$$\begin{cases} \dfrac{dCa_{pre}}{dt} = -\dfrac{Ca_{pre}}{\tau_{pre}} & Ca_{pre}(0) = 0 \\ \dfrac{dCa_{jump}}{dt} = \dfrac{1 - Ca_{jump}}{\tau_{rec}} - \delta_{Ca} \cdot Ca_{jump} \cdot Ca_{pre} & Ca_{jump}(0) = 1. \end{cases} \tag{2}$$

Upon arrival of the presynaptic spikes, $t \in (t_1, \cdots, t_n)$, we update $Ca_{pre}$ according to the deterministic jump:

$$Ca_{pre} \rightarrow Ca_{pre} + Ca_{jump}.$$

Finally, after $Ca_{pre}$ has been updated, a vesicle may be released with probability $p_{rel}$ (*Figure 8b*).

Parameters for the vesicle release model are given in *Table 2*. We constrained these parameters against the data reported by *Hardingham et al., 2006* and *Tigaret et al., 2016*. Because [Ca²⁺]ₒ modifies the release probability dynamics (*King et al., 2001*), we fixed an initial release probability to 68% for [Ca²⁺]ₒ = 2.5 mM as reported by *Tigaret et al., 2016* (initial value in *Figure 8b and d*). Additionally, *Hardingham et al., 2006* reports a 38% reduction in the initial release probability when changing [Ca²⁺]ₒ from 2.5 mM to 1 mM. Taking these into account, the decreasing sigmoid function in the *Figure 8e* depicts our [Ca²⁺]ₒ-dependent release probability model ($p_{rel}$).

*Figure 8e* shows that our $p_{rel}$ function is in good agreement with a previous analytical model suggesting that $p_{rel}([Ca^{2+}]_o) \propto ([Ca^{2+}]_o)^2\ mM^{-2}$ (*King et al., 2001*). Our model also qualitatively reproduces the decrease of calcium dye fluorescence levels after 20 s of theta trains from *Tigaret et al., 2016*. We interpret their fluorescence measurements as an effect of short-term depression (see *Figure 8b*).

Despite these agreements between our model and previous work, it is a simplified presynaptic model that does not encompass the highly heterogeneous nature of vesicle release. Vesicle release dynamics are known to be sensitive to various experimental conditions such as temperature (*Fernández-Alfonso and Ryan, 2004*), the age for some brain regions (*Rudolph et al., 2015*) or magnesium concentration (*Hardingham et al., 2006*). Although we did not consider this for the presynaptic model, note that we do incorporate such experimental parameters (age, temperature, $[Ca^{2+}]_o$, $[Mg^{2+}]_o$) to our model of NMDARs, which is the main postsynaptic calcium source. Furthermore, since our model of vesicle dynamics is simple, $\tau_{rec}$ in *Equation 2* has two roles: to delay the $p_{rel}$ recovery caused by $Ca_{pre}$ inactivation (enforced by $\delta_{Ca}$ in *Equation 2*) and to prevent vesicle release after HFS-induced depression (*King et al., 2001*; *Rizzoli and Betz, 2005*). Our presynaptic model for $p_{rel}$ is purely phenomenological as, in principle, the $[Ca^{2+}]$ jump parameter $\delta_{Ca}$ should depend on $[Ca^{2+}]_o$, but replacing the model with a more physiological model would not affect the results, since the measured dependence of release probability on $[Ca^{2+}]_o$ is already satisfied by this phenomenological model. Also, multi-vesicular release (MVR) at SC-CA1 synapses was shown to be prominent after manipulations that increase release probability, for example during the facilitation seen with paired-pulse stimulations (*Christie and Jahr, 2006*; *Oertner et al., 2002*). Yet, we chose not to incorporate this mechanism in the pre-synaptic model because we do not hold enough information on how MVR participates to plasticity outcomes of the different protocols we used in this study.

## Model compartments

Our model has three compartments. 1) a spherical dendritic spine linked by the neck to 2) a cylindrical dendrite connected to 3) a spherical soma. The membrane potential of these compartments satisfy the equations below (parameters in *Table 3*). Since the dendrite is a single compartment, the precise spine location is undefined. For more detailed morphological simulations to predict plasticity, see *Ebner et al., 2019*, *Chindemi et al., 2022* and *Jędrzejewska-Szmek et al., 2017*. The distance from the soma to the spine functionally mimics the BaP attenuation as shown in *Golding et al., 2001*, and it is set to 200 μm for all simulations, except in *Figure 3—figure supplement 4e* and *Figure 4—figure supplement 1c*. In these panels, we modified this distance parameter as described in the graph y-axis obtaining results similar to *Ebner et al., 2019*. The different currents in the soma, dendrite and spine are described in *Equations 3–5*.

## Membrane potential and currents

The membrane potential of these compartments satisfy the equations below (parameters in *Table 3*). The different currents are described in the following sections.

$$C_{sp} \cdot \frac{dV_{sp}}{dt} = g_{neck} \cdot (V_{dend} - V_{sp}) + g_L^{sp} \cdot (E_{rev} - V_{sp}) + I_T + I_L + I_R + I_{NMDA} + I_{AMPA} + I_{SK} \tag{3}$$

$$C_{dend} \cdot \frac{dV_{dend}}{dt} = g_{BaP}^{adapt} \cdot (V_{soma} - V_{dend}) + g_{neck} \cdot (V_{sp} - V_{dend}) + g_L^{dend} \cdot (E_{rev} - V_{dend}) + I_{GABA} \tag{4}$$

$$C_{soma} \cdot \frac{dV_{soma}}{dt} = g_{BaP}^{adapt} \cdot (V_{dend} - V_{soma}) + g_L^{soma} \cdot (E_{rev} - V_{soma}) + \lambda_{age} \cdot (I_{BaP} + I_{Na}) + I_K \tag{5}$$

## Action-potential backpropagation (BaP)

### Postsynaptic currents

The postsynaptic currents are generated in the soma, backpropagated to the dendritic spine via a passive dendrite which filters them. The soma generates BaPs using a version of the Na+ and K+ channel models developed by *Migliore et al., 1999*. The related parameters are described in *Table 4* (the voltage unit is mV). We used the following the units: $\alpha_x(V/mV)$ $[ms^{-1}]$ and $\beta_x(V/mV)$ $[ms^{-1}]$.

**Sodium channel**

$$\alpha_m(V_{soma}) = 0.4 \cdot \frac{V_{soma}+30}{1-e^{-\frac{V_{soma}+30}{7.2}}}$$

$$\beta_m(V_{soma}) = 0.124 \cdot \frac{V_{soma}+30}{e^{\frac{V_{soma}+30}{7.2}}-1}$$

$$m_{\inf}(V_{soma}) = \frac{\alpha_m(V_{soma})}{\alpha_m(V_{soma})+\beta_m(V_{soma})}$$

$$m_\tau(V_{soma}) = \frac{1}{\alpha_m(V_{soma})+\beta_m(V_{soma})}$$

$$\alpha_h(V_{soma}) = 0.01 \cdot \frac{V_{soma}+45}{e^{\frac{V_{soma}+45}{1.5}}-1}$$

$$\beta_h(V_{soma}) = 0.03 \cdot \frac{V_{soma}+45}{1-e^{-\frac{V_{soma}+45}{1.5}}}$$

$$\frac{dh}{dt} = \alpha_h(V_{soma}) \cdot (1-h) - \beta_h(V_{soma}) \cdot h$$

$$\frac{dm}{dt} = \frac{m_{\inf}-m}{m_\tau}$$

$$I_{Na} = \gamma_{Na} \cdot m^3 \cdot h \cdot (Erev_{Na} - V_{soma}).$$

**Potassium channel**

$$\alpha_n(V_{soma}) = e^{-0.11 \cdot (V_{soma}-13)}$$

$$\beta_n(V_{soma}) = e^{-0.08 \cdot (V_{soma}-13)}$$

$$n_{\inf}(V_{soma}) = \frac{1}{1+\alpha_n(V_{soma})}$$

$$n_\tau(V_{soma}) = max\left(50 \cdot \frac{\beta_n(V_{soma})}{1+\alpha_n(V_{soma})}; 2\right)$$

$$\frac{dn}{dt} = \frac{n_{\inf}-n}{n_\tau}$$

$$I_K = \gamma_K \cdot n \cdot (Erev_K - V_{soma})$$

To trigger a BaP, an external current $I_{BaP}$ is injected in the soma at times $t \in \{t_1, ..., t_n\}$ (postsynaptic input times) for a chosen duration $\delta_{inj}$ with amplitude $I_{amp}$ ($nA$). Taking $H$ as the Heaviside function, this is expressed as:

$$I_{BaP} = \sum_{i=1}^{n} H(t_i) \cdot (1 - H(t_i + \delta_{inj})) \cdot I_{amp}.$$

The currents underlying the BaP in the soma are filtered in a distance-dependent manner by the dendrite before reaching the dendritic spine. Biologically, BaP adaptation is caused by the inactivation of sodium channels and variations in sodium and potassium channel expression along the dendrite (*Jung et al., 1997*; *Golding et al., 2001*). We used a phenomenological model, implementing distant-dependent BaP amplitude attenuation by modifying the axial resistance $g_{BaP}^{adapt}$ (see *Equation 4* and *Equation 5*) between the dendrite and the soma as follows (*Figure 9c* top):

$$g_{BaP}^{adapt} = \lambda \cdot g_{diff} \cdot \phi_{dist}(d_{soma}), \qquad \phi_{dist}(d_{soma}) = 0.1 + \frac{1.4}{1 + e^{0.02 \cdot (d_{soma} - 230.3\mu m)}} \qquad (6)$$

where $d_{soma}$ is the distance of the spine from the soma and where the factor is dynamically regulated based on a resource-use equation from *Tsodyks and Markram, 1997* with a dampening factor $\lambda_{aux}$ changing the size of the attenuation step $\delta_{decay}$:

$$\frac{d\lambda}{dt} = \frac{1-\lambda}{\tau_{rec}} - \delta_{decay} \cdot \lambda_{aux}^{-1} \cdot \lambda \cdot I_{BaP}(t)$$

$$\frac{d\lambda_{aux}}{dt} = \frac{1-\lambda_{aux}}{\tau_{rec}} - \delta_{aux} \cdot \lambda_{aux} \cdot I_{BaP}(t).$$

The BaP attenuation model is based on *Golding et al., 2001* data for strongly attenuating neurons. Therefore, the second type of attenuation (weakly attenuating) in neurons is not considered (dichotomy in *Figure 9a*). *Figure 9a* compares Golding's data to our model and illustrates the effect of BaP attenuation in the upper panels of *Figure 9a, b*.

*Table 4* shows the BaP attenuation parameters. The plasticity outcomes as function of the dendritic spine distance from the soma are shown in *Figure 3—figure supplement 4e* and *Figure 4—figure supplement 1c*.

## Age-dependent BaP adaptation

Neuronal bursting properties are altered during development through development through the maturation and expression of potassium and sodium channels (*Gymnopoulos et al., 2014*), which change the interaction of hyperpolarizing and depolarizing currents (see *Figure 9b*; *Grewe et al.,*

**Table 3.** Parameters for the neuron electrical properties.

\* The membrane leak conductance in the spine is small since the spine membrane resistance is so high that is considered infinite ($> 10^6 M\Omega$) (**Koch and Zador, 1993**). The current thus mostly leaks axially through the neck cytoplasm. The dendrite leak conductance is also small in order to control the distance-dependent attenuation by the axial resistance term $g_{BaP}^{adapt}$ in **Equation 4** and **Equation 5**. The table provides the parameters associated with these equations which were adjusted (see comparison with Reference value) to fit the dynamics seen in **Golding et al., 2001** and **Buchanan and Mellor, 2007** experiments as in **Figure 9a and b**.

| Name | Value | Reference |
|---|---|---|
| **Passive cable** | | |
| Leak reversal potential | $E_{leak} = -70 \ mV$ | $69 \ mV$ (**Spigelman et al., 1996**) |
| Membrane leak conductance per area (for spine and passive dendrite) | $g_{leak} = 4 \cdot 10^{-6} \ nS/\mu m^2$ | * see table legend (**Koch and Zador, 1993**) |
| Membrane leak conductance per area (only soma) | $g_{soma} = 5.31 \cdot 10^{-3} \ nS/\mu m^2$ | $3 \cdot 10^{-4}$ to $1.3 \cdot 10^{-3} nS/\mu m^2$ (**Fernandez and White, 2010**) |
| Membrane capacitance per area | $C_m = 6 \cdot 10^{-3} \ pF/\mu m^2$ | $1 \cdot 10^{-2} \ pF/\mu m^2$ (**Hines and Carnevale, 1997**) |
| Axial resistivity of cytoplasm | $R_a = 1 \cdot 10^{-2} \ G\Omega \mu m$ | $2 \cdot 10^{-3} \ G\Omega \mu m$ (**Golding et al., 2001**) |
| **Dendrite** | | |
| Dendrite diameter | $D_{dend} = 2 \ \mu m$ | same as **Yi et al., 2017** |
| Dendrite length | $L_{dend} = 1400 \ \mu m$ | apical dendrites, 1200–1600 $\mu m$ **López Mendoza et al., 2018** |
| Dendrite surface area | $A_{dend} = 8.79 \cdot 10^3 \ \mu m^2$ | $\pi \cdot D_{dend} \cdot L_{dend}$ |
| Dendrite volume | $Vol_{dend} = 4.4 \cdot 10^3 \mu m^3$ | $\pi \cdot (D_{dend}/2)^2 \cdot L_{dend}$ |
| Dendritic membrane capacitance | $C_{dend} = 52.77 \ pF$ | $C_m \cdot A_{dend}$ |
| Dendrite leak conductance | $g_L^{dend} = 3.51 \cdot 10^{-2} \ nS$ | $g_{leak} \cdot A_{dend}$ |
| Dendrite axial conductance | $g_{diff} = 50 \ nS$ | $R_a \cdot A_{dend}$ |
| **Soma** | | |
| Soma diameter | $D_{soma} = 30 \ \mu m$ | 21 $\mu m$ (**Stuart et al., 2016**) page 3 |
| Soma area (sphere) | $A_{soma} = 2.82 \cdot 10^3 \ \mu m^2$ | $(4\pi/3) \cdot (D_{soma}/2)^3;$ $2.12 \cdot 10^3 \ \mu m^2$ (**Zhuravleva et al., 1997**) |
| Soma membrane capacitance | $C_{soma} = 16.96 \ pF$ | $C_m \cdot A_{soma}$ |
| Soma leaking conductance | $g_L^{soma} = 15 \ nS$ | $g_{soma} \cdot A_{soma}$ (**Fernandez and White, 2010**) |
| **Dendritic spine** | | |
| Spine head volume | $Vol_{sp} = 0.03 \ \mu m^3$ | **Bartol et al., 2015** |
| Spine head surface | $A_{sp} = 4.66 \cdot 10^{-1} \ \mu m^2$ | $4\pi \cdot (3Vol_{sp}/4\pi)^{2/3}$ |
| Spine membrane capacitance | $C_{sp} = 2.8 \cdot 10^{-3} \ pF$ | $C_m \cdot A_{sp}$ |
| Spine head leak conductance | $g_L^{sp} = 1.86 \cdot 10^{-6} \ nS$ | $g_{leak} \cdot A_{sp}$ |

*Table 3 continued on next page*

*Table 3 continued*

| Name | Value | Reference |
|---|---|---|
| **Dendritic spine neck** | | |
| Spine neck diameter | $D_{neck} = 0.1\ \mu m$ | 0.05–0.6 $\mu m$ *Harris et al., 1992* |
| Neck length | $L_{neck} = 0.2\ \mu m$ | $0.7 \pm 0.6\ \mu m$ (*Adrian et al., 2017*) |
| Neck cross-sectional area | $CS_{neck} = 7.85 \cdot 10^{-3}\ \mu m^2$ | $\pi \cdot (D_{neck}/2)^2$ |
| | | $CS_{neck}/(L_{neck} \cdot R_a)$ |
| Neck resistance | $g_{neck} = 3.92\ nS \approx 255.1\ M\Omega$ | 50 to 550 $M\Omega$ ($275 \pm 27\ M\Omega$) (*Popovic et al., 2015*) |

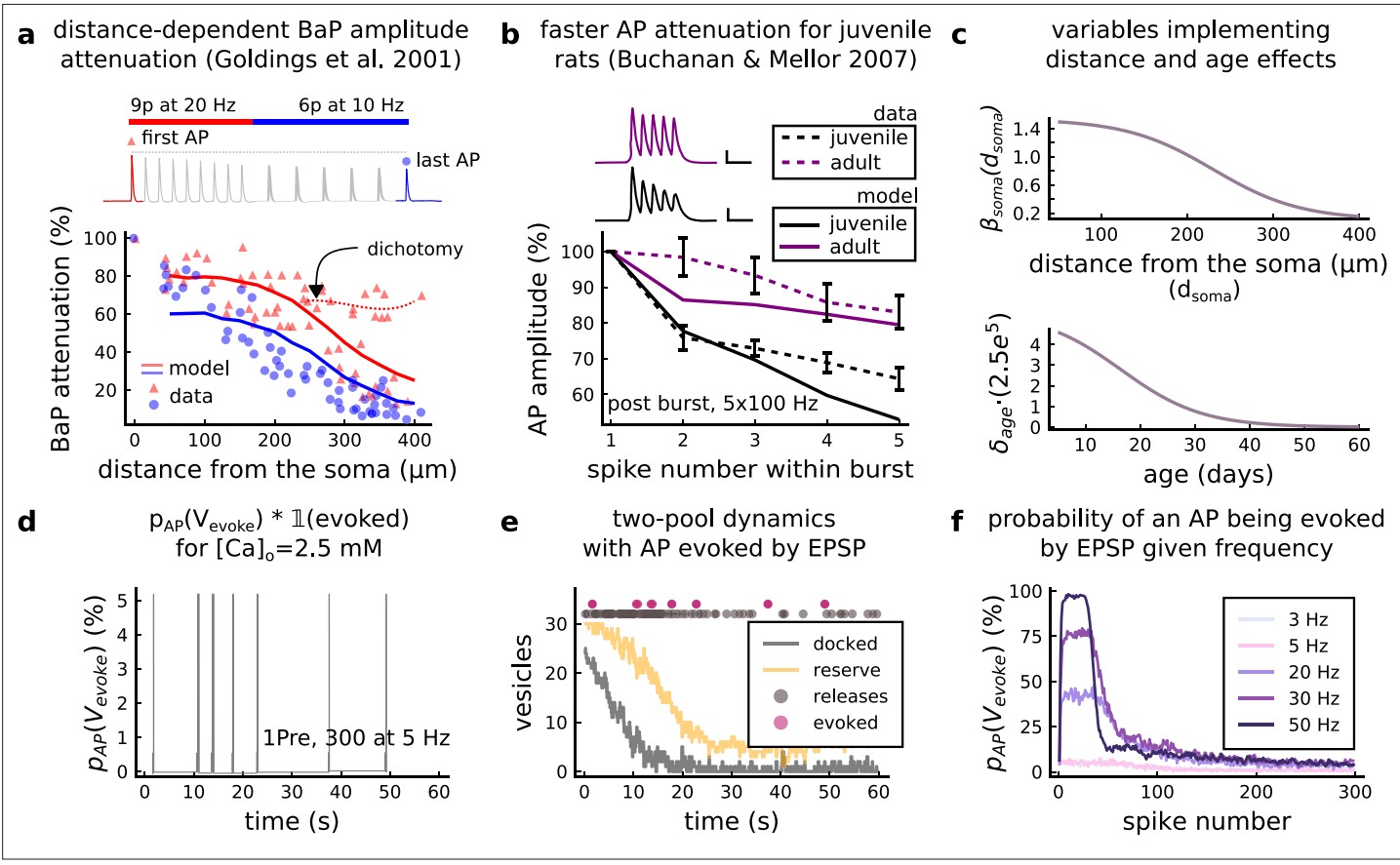

**Figure 9.** AP Evoked by EPSP. (**a**) Model and data comparison for the distance-dependent BaP amplitude attenuation measured in the dendrite and varying the distance from the soma. The stimulation in panel a is set to reproduce the same stimulation as *Golding et al., 2001*. Golding described two classes of neurons: those that are strongly attenuated and those that are weakly attenuated (dichotomy mark represented by the dashed line). However, in this work we consider only strongly attenuated neurons. (**b**) Attenuation of somatic action potential from *Buchanan and Mellor, 2007* and model in response to five postsynaptic spikes delivered at 100 Hz. The value showed for the model is the spine voltage with distance from the soma set to zero (scale 25 ms, 20 mV). (**c**) Top panel shows the $\lambda_{soma}$ used in *Equation 6* to modify the axial conductance between the soma and dendrite. Bottom panel shows the age-dependent changes in the step of the resource-use equation (*Equation 7*) that accelerates the BaP attenuation and decreases the sodium currents in *Equation 5*. (**d**) Probability of evoking an AP multiplied by the successfully evoked AP, $p_{AP}(V_{evoked}) \cdot \mathbb{1}(evoked)$, for the protocol 1Pre, 300 at 5 Hz (2.5 mM Ca). (**e**) Two-pool dynamics with the same stimulation from panel **D** showing the vesicle release, the reserve and docked pools, and the evoked AP. (**f**) Probability of evoking an AP for the protocol 1Pre 300 pulses at different frequencies (3 and 5 Hz have the same probability).

**Table 4.** The Na $+$and K$+$conductances intentionally do not match the reference because models with passive dendrite need higher current input to initiate action potentials (*Levine and Woody, 1978*).

Therefore, we set it to achieve the desired amplitude on the dendrite and the dendritic spine according to the predictions of *Golding et al., 2001* and *Kwon et al., 2017*.

| Name | Value | Reference |
|---|---|---|
| **Soma parameters for Na $+$and K$+$ channel** | | |
| Sodium conductance | $\gamma_{Na} = 8 \cdot 10^2 \, nS$ | generic value, see legend commentary |
| Potassium conductance | $\gamma_K = 40 \, nS$ | generic value, see legend commentary |
| Reversal potential sodium | $Erev_{Na} = 50 \, mV$ | *Migliore et al., 1999* |
| Reversal potential potassium | $Erev_K = -90 \, mV$ | *Migliore et al., 1999* |
| **BaP attenuation parameters** | | |
| Attenuation step factor (age) | $\delta_{age}$ | see *Equation 7* and *Figure 9b and c bottom Buchanan and Mellor, 2007*; *Golding et al., 2001* |
| Attenuation step factor | $\delta_{decay} = 1.727 \cdot 10^{-5}$ | adjusted to fit *Buchanan and Mellor, 2007*; *Golding et al., 2001* |
| Auxiliary attenuation step factor | $\delta_{aux} = 2.304 \cdot 10^{-5}$ | adjusted to fit *Buchanan and Mellor, 2007*; *Golding et al., 2001* |
| Recovery time for the attenuation factor | $\tau_{rec} = 2 \, s$ | adjusted to fit *Buchanan and Mellor, 2007*; *Golding et al., 2001* |
| Recovery time for the age attenuation factor | $\tau_{rec}^{age} = 0.5 \, s$ | adjusted to fit *Buchanan and Mellor, 2007*; *Golding et al., 2001* |
| **AP evoked by EPSP** | | |
| Decay time for $V_{evoke}$ | $\tau_V = 0.04 \, s$ | *Hines and Carnevale, 1997* |
| Delay AP evoked by EPSP | $\delta_{delay-AP} = 0.015 \, s$ | *Fricker and Miles, 2000* |

*2010*; *Jung et al., 1997*). We fitted the data of the age dependent somatic attenuation profiles from *Buchanan and Mellor, 2007* (*Figure 9b*) with our model by including an age-dependent BaP amplitude attenuation factor. We define the attenuation factor $\lambda_{age}$ (*Figure 9c bottom*), as follows.

$$\frac{d\lambda_{aux}}{dt} = \frac{1 - I_{age}}{\tau_{rec}^{age}} - \delta_{age} \cdot \lambda_{age} \cdot I_{BaP}(t), \qquad \delta_{rec}^{age} = \frac{1.391 \cdot 10^{-4}}{1 + e^{0.135 \cdot (age - 16.482 \, days)}}. \qquad (7)$$

In *Equation 5*, the age effects are introduced by multiplying the sodium $I_{Na}$ and the external $I_{BaP}$ currents by the attenuation factor $\lambda_{age}$.

## AP evoked by EPSP

Biologically, a presynaptic stimulation triggers a BaP if sufficient depolarization is caused by the EPSPs reaching the soma (*Stuart et al., 2016*). To model this effect for some LFP recordings protocols, we included an option to choose whether an EPSP can evoke an AP using an event generator resembling the previous presynaptic release probability model $p_{rel}$, as in *Equation 1*. Like $p_{rel}$, the BaPs evoked by EPSPs are calculated offline, before the postsynaptic simulation. We use a variable $V_{evoke}$ which is incremented by 1 at each presynaptic time $t \in (t_1, ..., t_n)$ and has exponential decay:

$$\begin{cases} \dfrac{dV_{evoke}}{dt} = -\dfrac{V_{evoke}}{\tau_v} & V_{evoke}(0) = 0 \\ V_{evoke} \longrightarrow V_{evoke} + 1. \end{cases} \qquad (8)$$

Since the BaPs evoked by EPSPs are triggered by the afferent synapses and are limited by their respective docked pools ($D$), we use the previous $p_{rel}$ to define the probability of an AP to occur. We test the ratio of successful releases from 25 synapses to decide if a BaP is evoked by an EPSP, setting a test threshold of 80%. Therefore, we express the probability of evoking an AP, $p_{AP}(V_{evoke})$, with the following test:

$$\frac{\sum^{25} \mathbf{1}(rand < p_{rel}(V_{evoked}, [Ca^{2+}]_o, D))}{25} > 80\% \, .$$

In real neurons, the EPSP summation dynamics in the soma and dendrites depend on the complex neuron morphology (*Etherington et al., 2010*; *Ebner et al., 2019*) which was not implemented by our model. Instead our 'AP evoked by EPSP test' is a simplified way to produce BaPs, similar to an integrate-and-fire model (*Sterratt et al., 2011*).

Previous work (*Mayr and Partzsch, 2010*) suggests that low-frequency stimulation, as used in *Dudek and Bear, 1992* ([Ca²⁺]ₒ = 2.5 mM), can evoke BaPs with a ~5% probability. Our model accounts for this, but also allows the probability of eliciting an AP to increase with stimulation frequency (*Etherington et al., 2010*). This is captured by $V_{evoke}$ as shown in *Figure 9f*. *Figure 9d, e* show how a 5 Hz stimulation can evoke APs. The delay between the EPSP and the evoked AP is set to $\delta_{delay-AP} = 15ms$, similar to the EPSP-spike latency reported for CA1 pyramidal neurons (*Fricker and Miles, 2000*).

## AMPAr

### Markov chain

The AMPArs are modeled as a Markov chain (*Figure 10*) described by *Robert and Howe, 2003* and *Coombs et al., 2017* and adapted to temperature changes according to *Postlethwaite et al., 2007*. Here, we introduce the additional parameters $\rho_f^{AMPA}, \rho_b^{AMPA}$ to cover AMPAr temperature-sensitive kinetics (*Postlethwaite et al., 2007*). The corresponding parameters are given in *Table 5*.

The AMPAr conductance is given as the sum of the occupancies of the three subconductance states $O2$, $O3$ and $O4$ of the Markov chain in *Figure 10*. The AMPAr current is then:

$$I_{AMPA} = (Erev_{AMPA} - V_{sp}) \cdot (\gamma_{A2} \cdot O2 + \gamma_{A3} \cdot O3 + \gamma_{A4} \cdot O4).$$

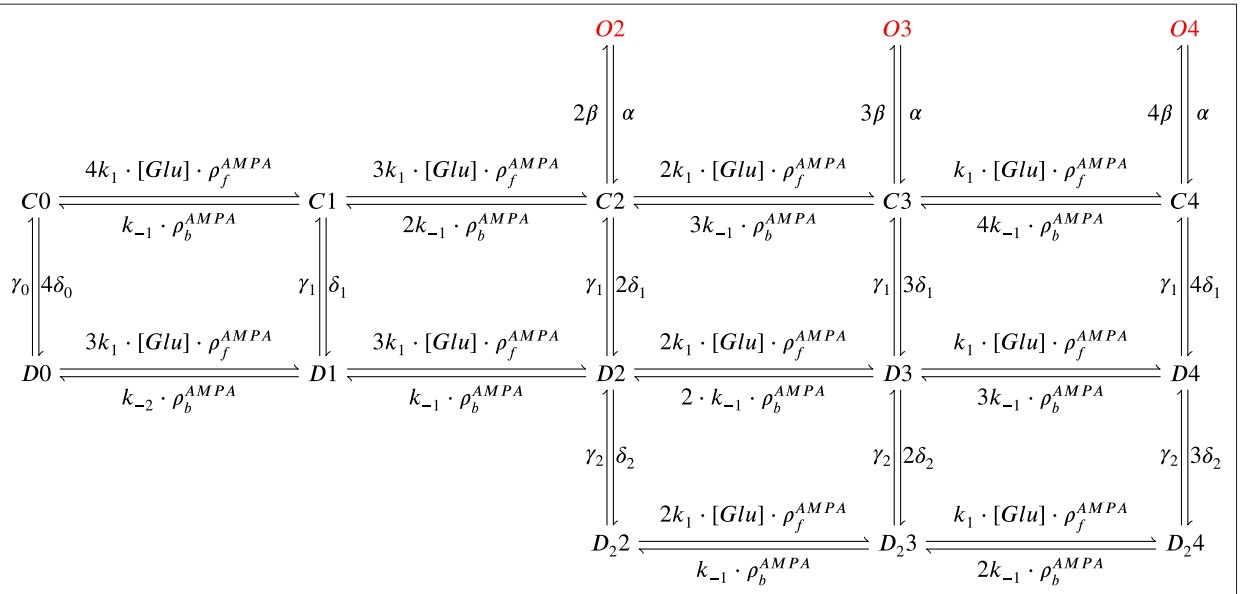

**Figure 10.** AMPAr Markov chain with three sub-conductance states and two desensitisation levels. It includes parameters $\rho_f^{AMPA}$, $\rho_b^{AMPA}$ (binding and unbinding of glutamate) which depend on temperature. Open states are $O2$, $O3$, and ; closed states are $C0$, $C1$, $C2$, $C3$, and $C4$; desensitisation states are $D0$, $D1$, $D2$, $D3$, and $D4$; deep desensitisation states are $D_22$, $D_23$, and $D_24$.

**Table 5.** Parameter values for the AMPAr Markov chain and glutamate release affecting NMDAr, AMPAr.

Properties of GABA release are the same as those for glutamate.

| Name | Value | Reference |
|---|---|---|
| **Glutamate parameters** | | |
| Duration of glutamate in the cleft | $glu_{width} = 0.001\ s$ | *Spruston et al., 1995* |
| Concentration of glutamate in the cleft | $glu_{amp} = 10^3\ \mu M$ | *Spruston et al., 1995* |
| Glutamate variability (gamma distribution$\Gamma$) | $glu_{cv} = \Gamma(1/0.5^2, 0.5^2)$ | *Liu et al., 1999* |
| Glutamate signal | $Glu$ | $glu_{cv} \cdot glu_{amp}$ for AMPAr, NMDAr and copied to GABA neurotransmitter |
| **AMPAr parameters** | | |
| Number of AMPArs | $N_{AMPA} = 120$ | *Bartol et al., 2015* |
| Reversal potential | $Erev_{AMPA} = 0\ mV$ | *Bartol et al., 2015* |
| Subconductance O2 | $\gamma_{A2} = 0.0155\ nS$ | $0.0163\ nS$ (*Coombs et al., 2017*) |
| Subconductance O3 | $\gamma_{A3} = 0.026\ nS$ | $0.0287\ nS$ (*Coombs et al., 2017*) |
| Subconductance O4 | $\gamma_{A4} = 0.0365\ nS$ | $0.0378\ nS$ (*Coombs et al., 2017*) |
| glu binding | $k_1 = 16\ \mu M^{-1}s^{-1}$ | *Robert and Howe, 2003* |
| glu unbinding 1 | $k_{-1} = 7400\ s^{-1}$ | *Robert and Howe, 2003* |
| glu unbinding 2 | $k_{-2} = 0.41\ s^{-1}$ | *Robert and Howe, 2003* |
| Closing | $\alpha = 2600\ s^{-1}$ | *Robert and Howe, 2003* |
| Opening | $\beta = 9600\ s^{-1}$ | *Robert and Howe, 2003* |
| Desensitisation 1 | $\delta_1 = 1500\ s^{-1}$ | *Robert and Howe, 2003* |
| Desensitisation 2 | $\delta_2 = 170\ s^{-1}$ | *Robert and Howe, 2003* |
| Desensitisation 3 | $\delta_0 = 0.003\ s^{-1}$ | *Robert and Howe, 2003* |
| Re-desensitisation 1 | $\gamma_1 = 9.1\ s^{-1}$ | *Robert and Howe, 2003* |
| Re-desensitisation 2 | $\gamma_2 = 42\ s^{-1}$ | *Robert and Howe, 2003* |
| Re-desensitisation 3 | $\gamma_0 = 0.83\ s^{-1}$ | *Robert and Howe, 2003* |

The adaptation of the Markov chain from *Robert and Howe, 2003* is made by changing the forward $\rho_f^{AMPA}$ and backward $\rho_b^{AMPA}$ rates in a temperature-dependent manner matching the decay time reported by *Postlethwaite et al., 2007*:

$$\rho_f^{AMPA} = \frac{10.273}{1+e^{-0.473\cdot(T-31.724^{\circ}C)}}, \qquad \rho_b^{AMPA} = \frac{5.134}{1+e^{-0.367\cdot(T-28.976^{\circ}C)}}.$$

The effects of temperature change on AMPAr dynamics are presented in *Figure 11*, which also shows that the desensitisation is not altered by temperature changes (*Figure 11b and c*). The recovery time from desensitisation is the same as at room temperature (*Robert and Howe, 2003*). Desensitisation measurements are required to account for a temperature-dependent change in the rates of the 'vertical' transitions in *Figure 10*, see *Postlethwaite et al., 2007*. This could be relevant for presynaptic bursts as indicated in *Figure 11b–c*.

## Postsynaptic Ca²⁺ influx

The effects of experimental conditions on the calcium dynamics are due to receptors, ion channels and enzymes. A leaky term models the calcium resting concentration in the *Equation 9*. The calcium fluxes from NMDAr and VGCCs (T, R, L types) are given in *Equation 10*. The diffusion term through the spine neck is expressed in *Equation 11*. Finally, the buffer, the optional dye and the enzymatic reactions are given in *Equation 12* (parameter values given at the *Table 6*):

$$\frac{dCa}{dt} = \frac{Ca_\infty - Ca}{\tau_{Ca}} + \tag{9}$$

$$\frac{Ca_{NMDA} + I_T + I_R + I_L}{2 \cdot F \cdot A_{sp}} + \tag{10}$$

$$\frac{max(Ca_\infty, Ca/3) - Ca}{\tau_{CaDiff}} - \tag{11}$$

$$\frac{dBuff_{Ca}}{dt} - \frac{dDye}{dt} + enzymes. \tag{12}$$

Despite the driving force to the resting concentration, $Ca_\infty = 50\ nM$, the tonic opening of T-type channels causes calcium to fluctuate making its mean baseline value dependent on temperature, extracellular calcium, and voltage. The effects of this tonic opening in various experimental conditions are shown in *Figure 6—figure supplement 2f*. To avoid modelling dendritic calcium sources, we use a dampening term as one-third of the calcium level since calcium imaging comparing dendrite and spine fluorescence have shown this trend (*Segal and Korkotian, 2014*). *Equation 11* implements the diffusion of calcium from the spine to the dendrite through the neck. The time constant for the diffusion coefficient $\tau_{CaDiff}$, is estimated as described in *Holcman et al., 2005*. The calcium buffer and the optional dye are described as a two-state reaction system (*Sabatini et al., 2002*):

$$\begin{aligned} \frac{dBuff_{Ca}}{dt} &= k_{on}^{Buff} \cdot (Buff_{con} - Buff_{Ca}) \cdot Ca - k_{off}^{Buff} \cdot Buff_{Ca} \\ \frac{dDye}{dt} &= k_{on}^{Fluo5} \cdot (Fluo5f_{con} - Dye) \cdot Ca - k_{off}^{Fluo5} \cdot Dye. \end{aligned} \tag{13}$$

*Tigaret et al., 2016* experiments used the synthetic calcium-indicator dye Fluo-5f, which is well-modelled by a single Ca²⁺-dye binding reaction (*Maravall et al., 2000*; *Bartol et al., 2015*). Although we include a detailed model of Calmodulin, which is a major endogenous calcium buffer, the other types are poorly quantified experimentally. Instead, we used a parsimonious generic buffer model that represents an aggregate of these largely unknown endogenous buffers. Future iterations of the model could include more detailed versions of these endogenous buffers, for example calbindin (*Bartol et al., 2015*).

Unlike other calcium-based plasticity models (*Graupner and Brunel, 2012*) using the dye fluorescence decay as an approximation to calcium decay, our model is based on receptor and ion channel kinetics. Additionally, our model can simulate the dye kinetics as a buffer using (*Equation 13*) when appropriate. *Figure 12* highlights differences between calcium and dye dynamics which is affected by the laser-induced temperature increase (*Wells et al., 2007*; *Deng et al., 2014*). We estimated the calcium reversal potential for the calcium fluxes using the Goldman–Hodgkin–Katz (GHK) flux equation described in *Hille, 1978*. The calcium ion permeability, $P_{Ca}$, was used as a free parameter adjusting a single EPSP to produce a calcium amplitude of ~3 μM (*Chang et al., 2017*). This free scaling is needed to compensate for the fact that that GHK equation is derived for a model that assumes ionic currents pass through the membrane as a distributed and continuous flux, rather that the ion channels we modelled as having discrete conductance levels. Although this adaptation implies that we are using the magnitude of the GHK flux in a phenomenological way, it nevertheless captures the nonlinear dependence of relative calcium flux on extracellular calcium concentration.

$$\begin{aligned} \phi(V_{sp}, T) &= z_{Ca} \cdot V_{sp} \cdot F/(R \cdot (T + 273.15K)) \\ \Phi_{Ca}(V_{sp}, [Ca^{2+}]_i) &= -P_{Ca} \cdot z_{Ca} \cdot F \cdot \phi(V_{sp}, T) \cdot \frac{[Ca^{2+}]_i - [Ca^{2+}]_o \cdot e^{-\phi}}{1 - e^{-\phi}}, \end{aligned} \tag{14}$$

where $\Phi_{Ca}(V_{sp}, [Ca^{2+}]_i)$ is used to determine the calcium influx through NMDAr and VGCC in the *Equation 15*, *Equation 16*, *Equation 17* and *Equation 18* using the spine membrane voltage and

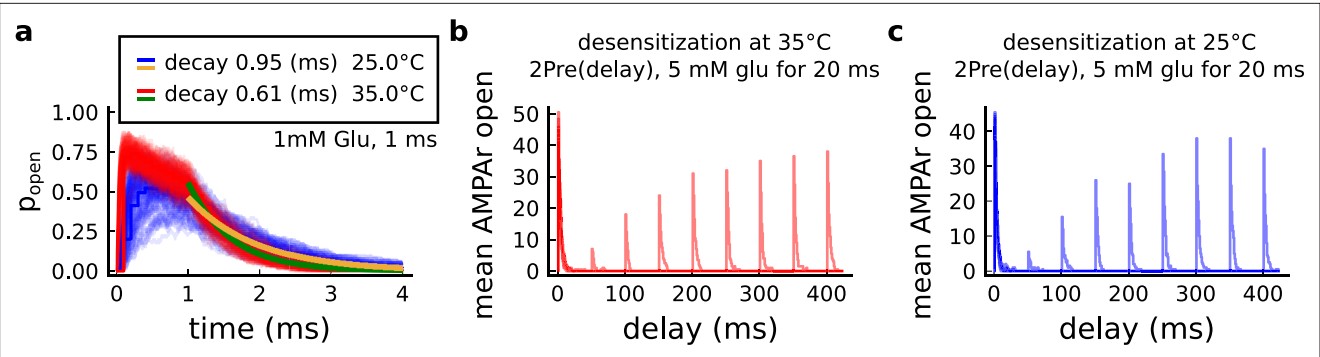

**Figure 11.** Effect of temperature on AMPArs. (**a**) Probability of AMPAr opening ($\frac{O2+O3+O4}{N_{AMPA}}$) and the decay time at different temperatures, in response to 1 mM glutamate applied for 1 ms (standard pulse). **Postlethwaite et al., 2007** data (our model) suggests that AMPAr decay time at 35°C is $\sim 0.5\ ms$ ($\sim 0.6\ ms$) and at 25°C is $\sim 0.65\ ms$ ($\sim 0.95\ ms$). This shows a closer match towards more physiological temperatures. (**b**) Desensitisation profile of AMPAr at 35°C showing how many AMPAr are open in response to repeated glutamate saturating pulses (5 mM Glu for 20 ms) separated by an interval (x-axis). (**c**) Same as in panel b but for 25°C.

**Table 6.** Postsynaptic calcium dynamics parameters.

Note that the buffer concentration, calcium diffusion coefficient, calcium diffusion time constant and calcium permeability were considered free parameters to adjust the calcium dynamics.

| Name | Value | Reference |
|---|---|---|
| **Buffer and dye** | | |
| Association buffer constant | $k_{on}^{Buff} = 247\ \mu M^{-1} s^{-1}$ | *Bartol et al., 2015* |
| Dissociation buffer constant | $k_{off}^{Buff} = 524\ s^{-1}$ | *Bartol et al., 2015* |
| Generic buffer concentration | $Buff_{con} = 62\ \mu M$ | $76.7\ \mu M$ (*Bartol et al., 2015*) |
| **Calcium dynamics** | | |
| Calcium baseline concentration | $Ca_\infty = 0.05\ \mu M$ | $0.037 \pm 0.005$ to $0.054 \pm 0.005\ \mu M$ (*Maravall et al., 2000*) |
| Calcium decay time | $\tau_{Ca} = 10^{-2}\ s$ | 0.05–0.5 s with dye (*Maravall et al., 2000*) therefore < 0.05–0.5 s undyed (unbuffered) |
| Calcium diffusion coefficient | $D_{Ca} = 333.8\ \mu m^2 s^{-1}$ | 200 to 400 $\mu m^2 s^{-1}$ (*Bartol et al., 2015*; *Holcman et al., 2005*) |
| Calcium diffusion time constant | $\tau_{CaDiff} = \frac{Vol_{sp}}{2D_{Ca}^2 \cdot D_{neck}} + \frac{L_{neck}^2}{2D_{Ca}} = 5 \cdot 10^{-4}\ s$ | 8 ms for a $Vol_{sp} = 0.7\ \mu m^3$ (*Holcman et al., 2005*) |
| **GHK equation** | | |
| Temperature | $T = 35°C$ | converted to Kelvin in the *Equation 14* given the protocol |
| Faraday constant | $F = 96.485\ C\ mol^{-1}$ | *Hille, 1978* |
| Gas constant | $R = 8.314\ J\ K^{-1}\ mol^{-1}$ | *Hille, 1978* |
| Calcium permeability | $P_{Ca} = 45\ \mu m\ s^{-1}$ | adjusted to produce 3µM Calcium in response to a Glu release supplementary file from *Chang et al., 2017* |
| Calcium ion valence | $z_{Ca} = 2$ | *Hille, 1978* |

calcium internal concentration $[Ca^{2+}]_i$. Note that for simplicity the calcium external concentration $[Ca^{2+}]_o$ was kept fixed during the simulation and only altered by experimental conditions given by the aCSF composition.

## NMDAr - GluN2A and GluN2B

### Markov chain

In hippocampus, NMDArs are principally heteromers composed of the obligatory subunit GluN1 and either the GluN2A or GluN2B subunits. These N2 subunits determine the kinetics of these receptors, with the GluN1/GLUN2B heteromers displaying slow kinetics (~250 ms) and the GluN1/GluN2A heteromers displaying faster kinetics (~50 ms). We modeled both NMDA subtypes. The NMDAr containing GluN2A is modeled with the following Markov chain (*Popescu et al., 2004*):

$$A_0 \underset{k_{-a}\cdot\rho_b^{NMDA}}{\overset{k_a\cdot[Glu]\cdot\rho_f^{NMDA}}{\rightleftharpoons}} A_1 \underset{k_{-b}\cdot\rho_b^{NMDA}}{\overset{k_b\cdot[Glu]\cdot\rho_f^{NMDA}}{\rightleftharpoons}} A_2 \underset{k_{-c}\cdot\rho_b^{NMDA}}{\overset{k_c\cdot\rho_f^{NMDA}}{\rightleftharpoons}} A_3 \underset{k_{-d}\cdot\rho_b^{NMDA}}{\overset{k_d\cdot\rho_f^{NMDA}}{\rightleftharpoons}} A_4 \underset{k_{-e}\cdot\rho_b^{NMDA}}{\overset{k_e\cdot\rho_f^{NMDA}}{\rightleftharpoons}} A_{O1} \underset{k_{-f}\cdot\rho_b^{NMDA}}{\overset{k_f\cdot\rho_f^{NMDA}}{\rightleftharpoons}} A_{O2}$$

where we have introduced the additional parameters $\rho_f^{NMDA}, \rho_b^{NMDA}$ to account for temperature dependence (below).

The NMDAr containing GluN2B is modeled with a Markov chain based on the above GluN2A scheme. We decreased the rates by ~75% in order to match the GluN2B decay at 25°C as published in *Iacobucci and Popescu, 2018*.

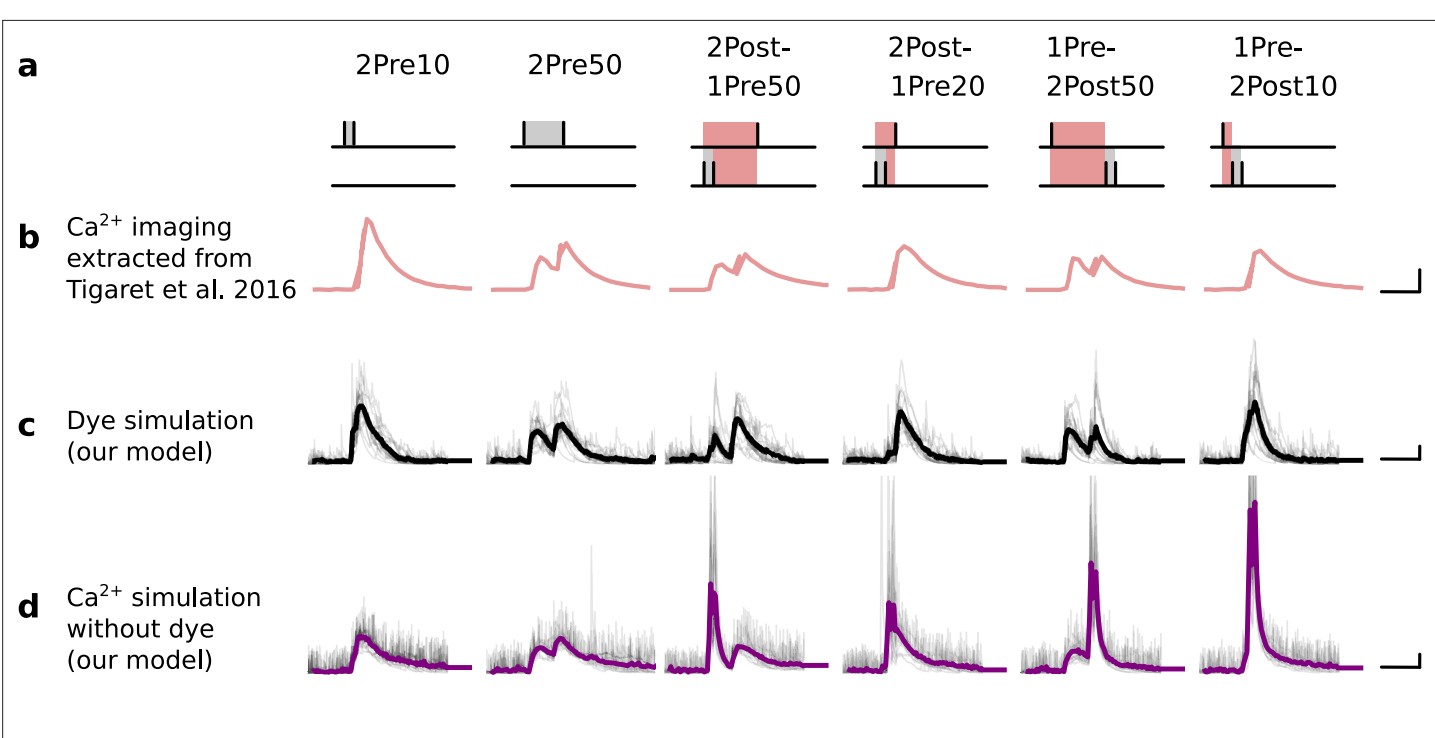

**Figure 12.** Differences between dye measurements and simulated calcium. (**a**), Pre and postsynaptic stimuli as used in *Tigaret et al., 2016*. (**b**), Calcium imaging curves (fluorescence ΔF/A) elicited using the respective stimulation protocols above with Fluo5 200 µM (extracted from *Tigaret et al., 2016*). Scale 100 ms, 0.05 ΔF/A. (**c**), Dye simulation using the model. The dye is implemented by increasing temperature to mimic laser effect on channel kinetics and decreases the interaction between NMDAr and voltage elicited by BaP. Temperature effects over NMDAr are shown in *Korinek et al., 2010*. Also, the effects of temperature on calcium-sensitive probes shown in *Oliveira et al., 2012* (baseline only, likely related to T-type channels). Other examples of laser heating of neuronal tissue are given in *Deng et al., 2014*. Such a dye curve fitting was obtained by increasing temperature by 10°C to mimic laser-induced heating (*Wells et al., 2007*; *Deng et al., 2014*). We achieved a better fit by decreasing the amplitude of the BaP that reaches the dendrite. Additionally, for fitting purposes, we assumed that a temperature increase lead to a decrease in BaP amplitude. Scale 0.6 µM dye, 100 ms. (**d**), Calcium simulation without dye. Scale 0.85 µM $Ca^{2+}$, 100 ms.

**Table 7.** NMDAr parameters.

The existing model of NMDAr (GluN2A) was adapted to obtain the NMDAr (GluN2B) model. The decay time of NMDAr (GluN2B) was fitted to match decay time in *Iacobucci and Popescu, 2018* and the temperature dependence uses the EPSP decay time from *Korinek et al., 2010*.

| Name | Value | Reference |
|---|---|---|
| NMDAr (GluN2A) | | |
| Glutamate binding | $k_a = 34\ \mu M^{-1}s^{-1}$ | *Popescu et al., 2004* |
| Glutamate binding | $k_b = 17\ \mu M^{-1}s^{-1}$ | *Popescu et al., 2004* |
| Forward rate | $k_c = 127\ s^{-1}$ | *Popescu et al., 2004* |
| Forward rate | $k_d = 580\ s^{-1}$ | *Popescu et al., 2004* |
| Opening rate | $k_e = 2508\ s^{-1}$ | *Popescu et al., 2004* |
| Opening rate | $k_f = 3449\ s^{-1}$ | *Popescu et al., 2004* |
| Closing rate | $k_{-f} = 662\ s^{-1}$ | *Popescu et al., 2004* |
| Closing rate | $k_{-e} = 2167\ s^{-1}$ | *Popescu et al., 2004* |
| Backward rate | $k_{-d} = 2610\ s^{-1}$ | *Popescu et al., 2004* |
| Backward rate | $k_{-c} = 161\ s^{-1}$ | *Popescu et al., 2004* |
| Glutamate unbinding | $k_{-b} = 120\ s^{-1}$ | *Popescu et al., 2004* |
| Glutamate unbinding | $k_{-a} = 60\ s^{-1}$ | *Popescu et al., 2004* |
| NMDAr (GluN2B) | | |
| Glutamate binding | $s_b = 0.25k_b$ | adapted from GluN2A (*Popescu et al., 2004*; *Iacobucci and Popescu, 2018*) |
| Glutamate binding | $s_c = 0.25k_c$ | adapted from GluN2A (*Popescu et al., 2004*; *Iacobucci and Popescu, 2018*) |
| Forward rate | $s_c = 0.25k_c$ | adapted from GluN2A (*Popescu et al., 2004*; *Iacobucci and Popescu, 2018*) |
| Forward rate | $s_d = 0.25k_d$ | adapted from GluN2A (*Popescu et al., 2004*; *Iacobucci and Popescu, 2018*) |
| Opening rate | $s_e = 0.25k_e$ | adapted from GluN2A (*Popescu et al., 2004*; *Iacobucci and Popescu, 2018*) |
| Opening rate | $s_f = 0.25k_f$ | adapted from GluN2A (*Popescu et al., 2004*; *Iacobucci and Popescu, 2018*) |
| Closing rate | $s_{-f} = 0.23k_{-f}$ | adapted from GluN2A (*Popescu et al., 2004*; *Iacobucci and Popescu, 2018*) |
| Closing rate | $s_{-e} = 0.23k_{-e}$ | adapted from GluN2A (*Popescu et al., 2004*; *Iacobucci and Popescu, 2018*) |
| Backward rate | $s_{-d} = 0.23k_{-d}$ | adapted from GluN2A (*Popescu et al., 2004*; *Iacobucci and Popescu, 2018*) |
| Backward rate | $s_{-c} = 0.23k_{-c}$ | adapted from GluN2A (*Popescu et al., 2004*; *Iacobucci and Popescu, 2018*) |
| Glutamate unbinding | $s_{-b} = 0.23k_{-b}$ | adapted from GluN2A (*Popescu et al., 2004*; *Iacobucci and Popescu, 2018*) |
| Glutamate unbinding | $s_{-a} = 0.23k_{-a}$ | adapted from GluN2A (*Popescu et al., 2004*; *Iacobucci and Popescu, 2018*) |
| Other parameters | | |

*Table 7 continued on next page*

*Table 7 continued*

| Name | Value | Reference |
|---|---|---|
| Total number of NMDAr | $N_{NMDA} = 15$ | 5–30 (*Spruston et al., 1995*; *Bartol et al., 2015*; *Nimchinsky et al., 2004*) |
| Distribution of GluN2A and GluN2B | defined by $r_{age}$ | fitted from *Sinclair et al., 2016*, see *Figure 13b and e* |
| NMDAr conductance depending on calcium | $\gamma_{NMDA}$ | fitted from *Maki and Popescu, 2014*, see *Figure 13c* |
| NMDAr reversal potential | $\text{Erev}_{NMDA} = 0\ mV$ | *Destexhe et al., 1994* |
| Fraction of calcium carried by NMDAr | $f_{Ca} = 0.1$ | *Griffith et al., 2016* |

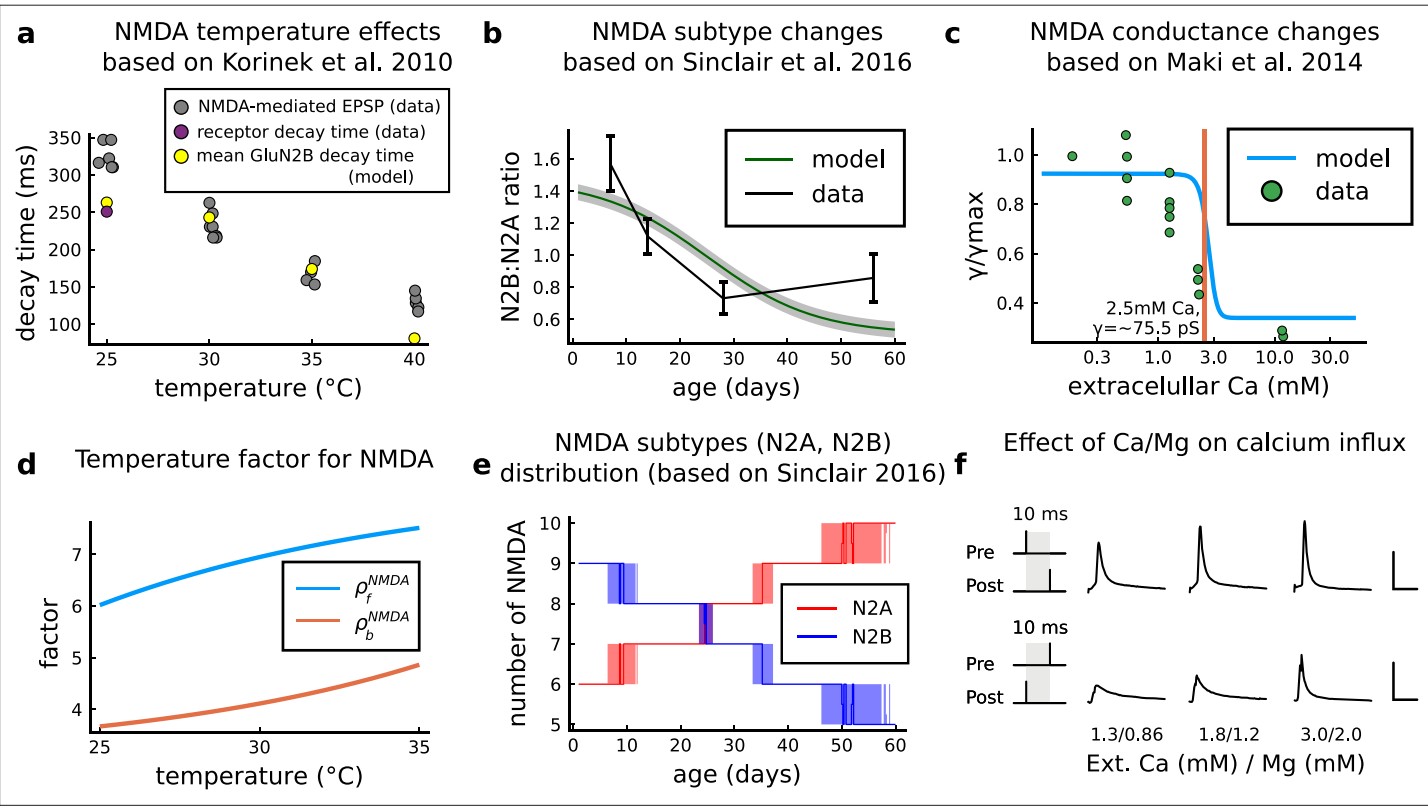

**Figure 13.** NMDAr changes caused by age, temperature and extracellular and magnesium concentrations in the aCSF. (**a**) Decay time of the NMDAr-mediated EPSP recorded from neocortical layer II/III pyramidal neurons (grey) (*Korinek et al., 2010*) compared to the decay time from the GluN2B channel estimated by our model (yellow) and data from Iacobussi's single receptor recording (purple) (*Iacobucci and Popescu, 2018*). (**b**), Comparison of our model of the GluN2B:GluN2A ratio and the GluN2B:GluN2A ratio from the mouse CA1 excitatory neurons. (**c**), Comparison of our model of NMDAr conductance change as a function of extracellular calcium, against data (*Maki and Popescu, 2014*). (**d**), Forward and backwards temperature factors implemented to approximate NMDAr subtypes decay times at room temperature (*Iacobucci and Popescu, 2018*) and temperature changes observed in *Korinek et al., 2010*. (**e**), NMDAr subtype number in our model as a function of animal age. We added noise to have a smoother transition between different ages. (**f**), Calcium concentration changes for causal and anticausal protocols in response to different aCSF calcium and magnesium compositions with fixed Ca/Mg ratio (1.5). Scale bars 50 ms and 5 μM.

$$B_0 \underset{s_{-a}\cdot\rho_b^{NMDA}}{\overset{s_a\cdot[Glu]\cdot\rho_f^{NMDA}}{\rightleftharpoons}} B_1 \underset{s_{-b}\cdot\rho_b^{NMDA}}{\overset{s_b\cdot[Glu]\cdot\rho_f^{NMDA}}{\rightleftharpoons}} B_2 \underset{s_{-c}\cdot\rho_b^{NMDA}}{\overset{s_c\cdot\rho_f^{NMDA}}{\rightleftharpoons}} B_3 \underset{s_{-d}\cdot\rho_b^{NMDA}}{\overset{s_d\cdot\rho_f^{NMDA}}{\rightleftharpoons}} B_4 \underset{s_{-e}\cdot\rho_b^{NMDA}}{\overset{s_e\cdot\rho_f^{NMDA}}{\rightleftharpoons}} B_{O1} \underset{s_{-f}\cdot\rho_b^{NMDA}}{\overset{s_f\cdot\rho_f^{NMDA}}{\rightleftharpoons}} B_{O2}$$

The different rates are given in *Table 7*.

## NMDAr and age switch

The age-dependent expression ratio of the subtypes GluN2A and GluN2B ($r_{age}$) was obtained from experimental data of mouse hippocampus (*Sinclair et al., 2016*). We added noise to this ratio causing ~1 NMDAr subunit to flip towards GluN2A or GluN2B (see *Figure 13e*). The population of 15 total NMDArs is divided in the two subtypes according to the ratio plotted in *Figure 13b*, as a function of age. The ratio to define the number NMDAr subtypes as function of age reads:

$$r_{age} = 0.507 + \frac{0.964}{1 + e^{0.099\cdot(age-25.102\ days)}} + \mathcal{N}(0, 0.05)$$
$$N_{GluN2B} = round\left(\frac{N_{NMDA}\cdot r_{age}}{r_{age}+1}\right)$$
$$N_{GluN2A} = round\left(\frac{N_{NMDA}}{r_{age}+1}\right).$$

The round term in the two previous equations ensures that we have an integer value for the NMDAr subtypes, making the stair shaped curve seen in *Figure 13e*.

## NMDAr and temperature

We adjusted the GluN2A and GluN2B forward and backward rates to follow the temperature effects on NMDAr-mediated EPSPs (*Korinek et al., 2010*), see *Figure 13a and d*. Because GluN2B dominates the NMDAr-mediated EPSP, we fit the GluN2B decay time to data on the NMDAr-mediated EPSP,w as function of temperature as reported by *Korinek et al., 2010* using logistic functions $\rho_f^{NMDA}$ and $\rho_b^{NMDA}$. The decay time comparison is shown in *Figure 13a*. Then, we applied the same temperature factor $\rho_f^{NMDA}$ and $\rho_b^{NMDA}$ for GluN2A. The decay times of GluN2A and GluN2B are similar to those reported by *Iacobucci and Popescu, 2018*. The forward and backward factors are described as follows:

$$\rho_f^{NMDA} = -1230.680 + \frac{1239.067}{1 + e^{-0.099\cdot(T+37.631°C)}}, \qquad \rho_b^{NMDA} = 3.036 + \frac{1621.616}{1 + e^{-0.106\cdot(T-98.999°C)}}.$$

## NMDAr current and Ca²⁺-dependent conductance

NMDAr conductance is modulated by external calcium and is modelled according to the next equations using NMDAr subconductances $A_{O1}$ and $A_{O2}$ (GluN2A), and $B_{O1}$ and $B_{O2}$ (GluN2B).

$$\gamma_{NMDA} = 33.949pS + \frac{58.388}{1 + e^{4\cdot([Ca^{2+}]_o - 2.701\ mM)}}pS$$
$$B(V_{sp}, [Mg]_o) = \frac{1}{1 + \frac{[Mg]_o}{3.57mM}\cdot e^{-0.062\cdot V_{sp}/mV}}$$
$$NMDA = (B_{O1} + B_{O2} + A_{O1} + A_{O2})\cdot B(V_{sp}, [Mg]_o)\cdot\gamma_{NMDA}$$
$$I_{NMDA} = (Erev_{NMDA} - V_{sp})\cdot NMDA$$

We modified the conductance $\gamma_{NMDA}$ as a function of extracellular calcium from that reported by *Maki and Popescu, 2014*. The reported NMDAr conductance at $[Ca^{2+}]_o$ = 1.8 mM is $53 \pm 5pS$. Here, we used the higher conductance 91.3 $pS$ for NMDAr (for both subtypes) at $[Ca^{2+}]_o$ = 1.8 mM to compensate for the small number of NMDArs reported by *Nimchinsky et al., 2004*. Hence, we adjusted the *Maki and Popescu, 2014* data to take into account this constraint: this caused a rightward-shift in the NMDA-conductance curve (*Figure 13c*). The calcium influx $Ca_{NMDA}$ is modulated by the GHK factor, *Equation 14*, as a function of the internal and external calcium concentrations and the spine voltage:

$$Ca_{NMDA} = f_{Ca}\cdot\Phi_{Ca}\cdot NMDA. \tag{15}$$

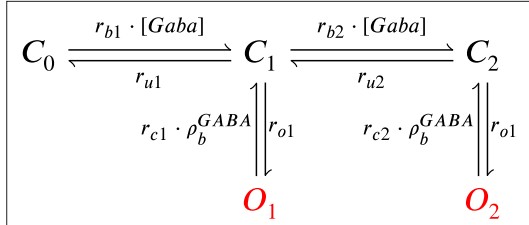

**Figure 14.** GABAr Markov chain model. Closed states ($C_0$, $C_1$ and $C_2$) open in response to GABAr and can go either close again or open ($O_1$ and $O_2$).

The combined effect of extracellular Magnesium (*Jahr and Stevens, 1990*) and Calcium concentration are displayed in *Figure 13f*.

## GABA(A) receptor

Since the precise delay of GABA release relative to glutamate is not known, we assumed GABA and glutamate release are synchronized for simplicity (see *Table 5*). We used the GABA(A) receptor Markov chain (*Figure 14*) presented in *Busch and Sakmann, 1990*; *Destexhe et al., 1998* and we estimated temperature adaptations using the measurements reported by *Otis and Mody, 1992*. *Table 8* presents the GABAr model parameters.

### GABA(A)r and temperature

Because the amplitude of GABA(A) current is altered by the GABAr reversal potential shift during development (*Rinetti-Vargas et al., 2017*), we applied temperature changes only to the closing rates using a logistic function for $\rho_b^{GABA}$, estimated by fitting to the measurements from *Otis and Mody, 1992* (data comparison in the *Figure 15b and e*).

$$\rho_b^{GABA} = 1.470 - \frac{-1.279}{1 + e^{0.191 \cdot (T - 32.167^\circ C)}}.$$

### GABA(A)r current and age switch

The GABA(A)r-driven current changes during development (*Meredith et al., 2003*) passing from depolarizing (excitatory) to hyperpolarizing (inhibitory) (*Chamma et al., 2012*). The reversal potential of chloride ions permeating GABA(A)r shifts from above the membrane resting potential (inward driving force - excitatory) to below the membrane resting potential (outward driving force - inhibitory; *Rinetti-Vargas et al., 2017*). This effect mediated is associated with the KCC2 pump (K Cl co-transporter) which becomes efficient in extruding chloride ions during maturation (*Rinetti-Vargas et al.,*

**Table 8.** GABAr parameters.
The GABAr number and conductance were modified to fit GABAr currents as in *Figure 15b and e*.

| Name | Value | Reference |
|---|---|---|
| **GABA(A) receptor** | | |
| Number of GABAr | $N_{GABA} = 34$ | 30 *Edwards et al., 1990* |
| Chloride reversal potential | see age-dependent equation | fitted from *Rinetti-Vargas et al., 2017* |
| GABAr conductance | $\gamma_{GABA} = 0.036\ nS$ | 0.027 $nS$ (*Macdonald et al., 1989*) |
| Binding | $r_{b1} = 20\ \mu M^{-1}\ s^{-1}$ | *Busch and Sakmann, 1990* |
| Unbinding | $r_{u1} = 4.6 \cdot 10^3\ s^{-1}$ | *Busch and Sakmann, 1990* |
| Binding | $r_{b2} = 10\ \mu M^{-1} s^{-1}$ | *Busch and Sakmann, 1990* |
| Unbinding | $r_{u2} = 9.2 \cdot 10^3\ s^{-1}$ | *Busch and Sakmann, 1990* |
| Opening rate | $r_{ro1} = 3.3 \cdot 10^3\ s^{-1}$ | *Busch and Sakmann, 1990* |
| Opening rate | $r_{ro2} = 10.6 \cdot 10^3\ s^{-1}$ | *Busch and Sakmann, 1990* |
| Closing rate | $r_{c2} = 400\ s^{-1}$ | temperature changes to fit *Otis and Mody, 1992*; *Busch and Sakmann, 1990* |
| Closing rate | $r_{c2} = 9.8 \cdot 10^3\ s^{-1}$ | temperature changes to fit *Otis and Mody, 1992*; *Busch and Sakmann, 1990* |

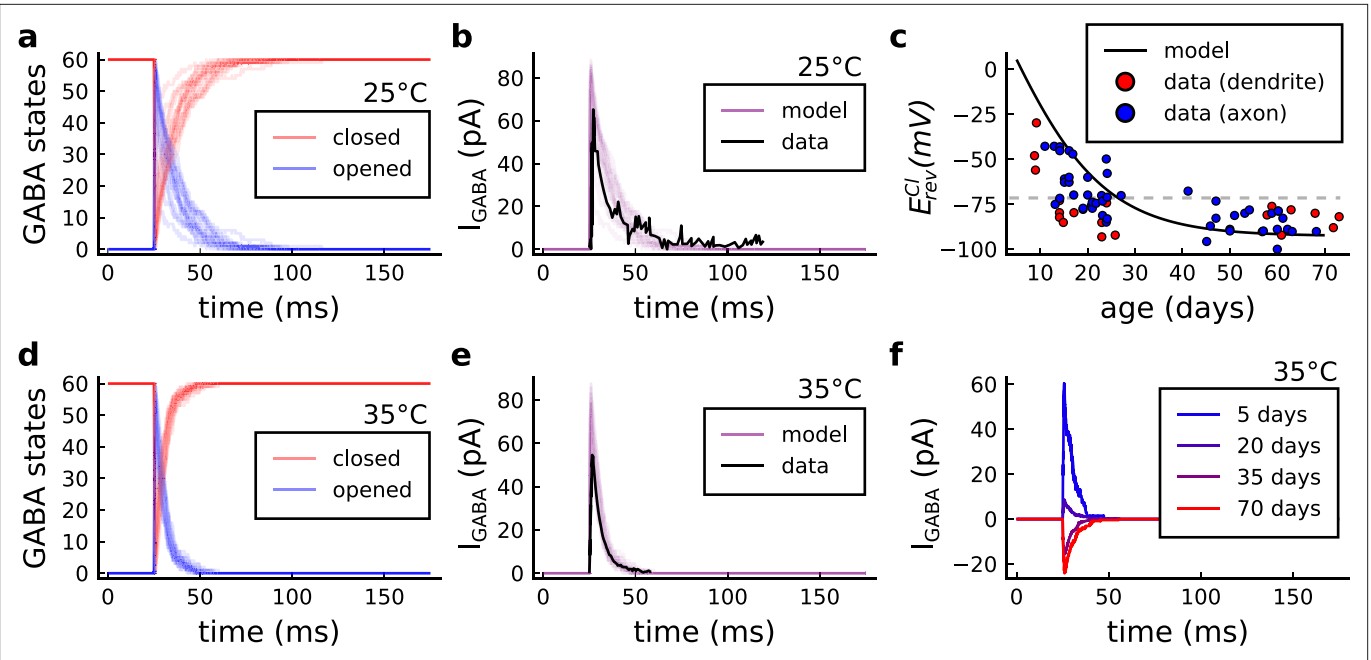

**Figure 15.** GABA(A)r current, kinetics and chloride reversal potential. (**a**) States of GABA(A)r Markov chain at 25°C in response to a presynaptic stimulation. Opened = $O_1 + O_2$, closed = $C_0 + C_1 + C_2$. (**b**) Model and data comparison (*Otis and Mody, 1992*) for GABA(A)r current at 25°C. Even though data were recorded from P70 at 25°C and P15 at 35°C, we normalize the amplitude to invert the polarity and compare the decay time. This is done since the noise around P15 can either make GABAr excitatory or inhibitory as shown by $E_{cl}$ data in panel c. (**c**) Chloride reversal potential ($E_{rev}^{Cl}$) fitted to *Rinetti-Vargas et al., 2017* data. Note that we used both profiles from axon and dendrite age-depended $E_{rev}^{Cl}$ changes since exclusive dendrite data is scarce. (**d**) States of simulated GABA(A)r Markov chain at 35°C in response to a presynaptic stimulation. (**e**) Model and data comparison (*Otis and Mody, 1992*) for GABA(A)r current at 25°C (same normalization as in panel b). (**f**) Change in the polarization of GABA(A)r currents given the age driven by the $E_{rev}^{Cl}$.

*2017*). To account for the GABA(A)r age-dependent shift, we fit a function for the chloride reversal potential ($E_{rev}^{Cl}$) to the data published by *Rinetti-Vargas et al., 2017* (*Figure 15c*):

$$E_{rev}^{Cl} = -92.649 + \frac{243.515}{1 + e^{0.091 \cdot (age - 0.691 \ days)}}$$
$$I_{GABA} = (O_1 + O_2) \cdot (E_{rev}^{Cl} - V_{dend}) \cdot \gamma_{GABA}.$$

## VGCC - T, R, and L type
### Markov chain
A stochastic VGCC model was devised using the channel gating measurements from rat CA1 (2–8 weeks) pyramidal neurons by *Magee and Johnston, 1995* at room temperature. Our model has three different VGCC subtypes described by the Markov chains in *Figure 16*: the T-type (low-voltage), the R-type (medium-to-high-voltage) and the L-type (high-voltage).

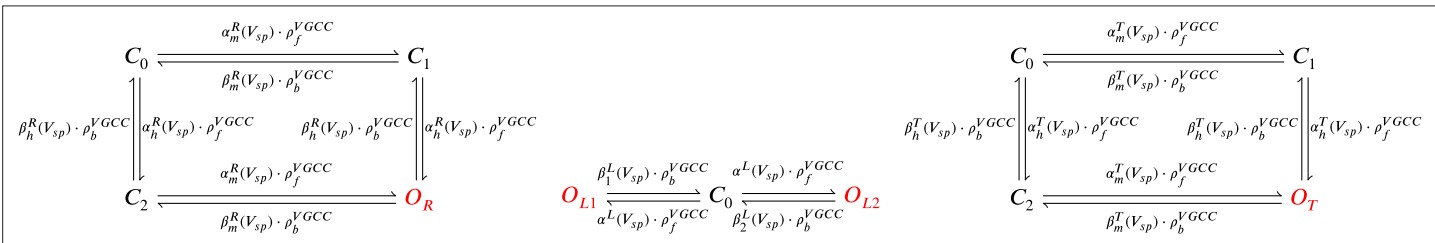

**Figure 16.** From left to right, R-, L-, and T-type VGCCs Markov chain adapted from *Magee and Johnston, 1995*. The R- (left scheme) and T- type (right scheme) have a single open state (red colour), respectively, $O_r$ and $O_T$. The L-type VGCC (middle) has two open states, $O_{L1}$ and $O_{L2}$.

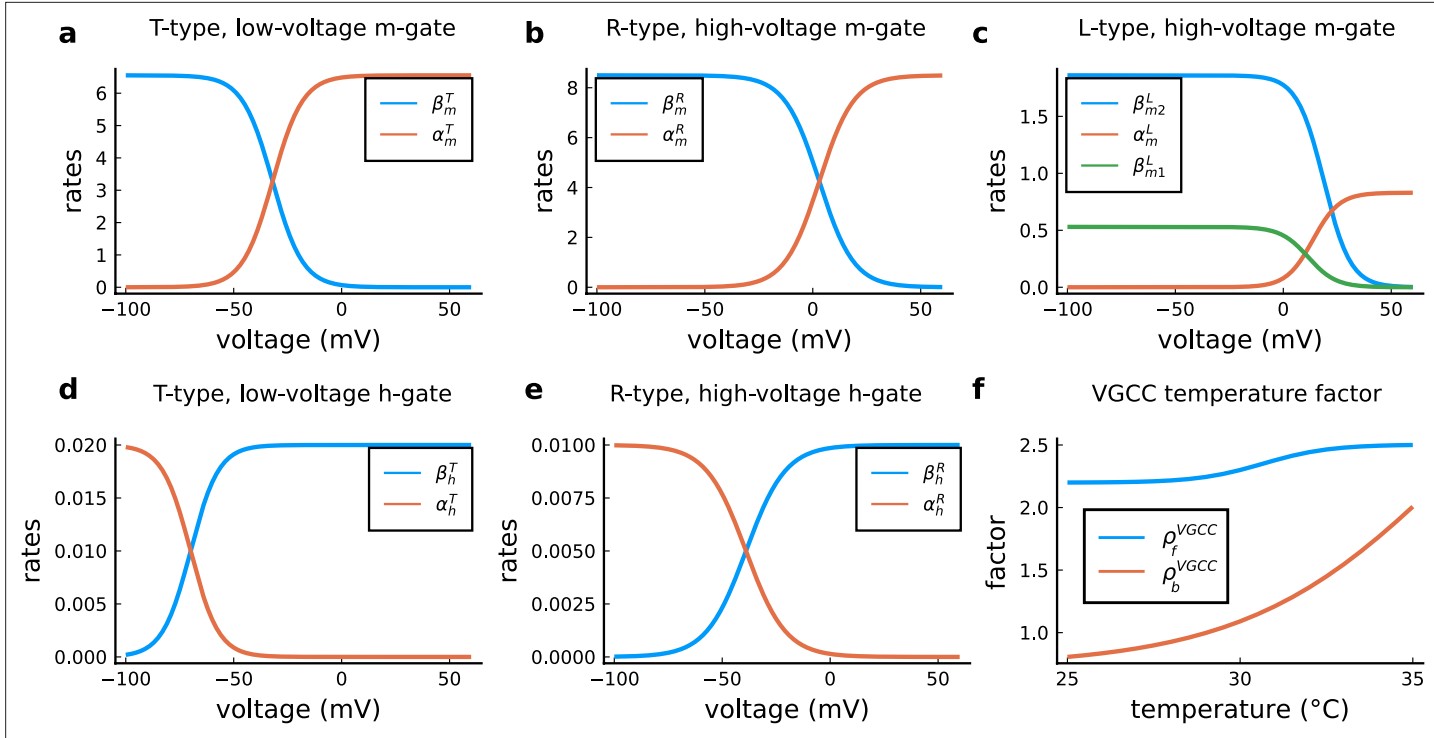

**Figure 17.** VGCC rates and temperature factors. (**a**), Activation ($\alpha_m(V_{sp})$) and deactivation rates ($\beta_m(V_{sp})$) for the T-type m-gate. (**b**), Activation ($\alpha_m(V_{sp})$) and deactivation rates ($\beta_m$) for the R-type m-gate. (**c**), Activation ($\alpha_m(V_{sp})$) and both deactivation rates ($\beta_2^L(V_{sp})$ and $\beta_2^1(V_{sp})$) for the L-type VGCC. (**d**), Activation ($\alpha_h(V_{sp})$) and deactivation rates ($\beta_h(V_{sp})$) for the T-type h-gate. (**e**), Activation ($\alpha_h(V_{sp})$) and deactivation rates ($\beta_h(V_{sp})$) for the R-type h-gate. (**f**), Temperature factor applied to all the rates, forward change ($\rho_f^{VGCC}$) for the $\alpha$ rates and backward change ($\rho_b^{VGCC}$) for the $\beta$ rates.

The VGCC Markov chain models were derived from voltage activation and inactivation profiles reported in *Magee and Johnston, 1995*. The T- (*Figure 17a and d*) and R-type (*Figure 17b and e*) models are composed of independent activation (m) and inactivation (h) gating variables, while the L-type (*Figure 17c*) model has one closed state but two open states, to capture the two timescales of channel closing kinetics reported by *Magee and Johnston, 1995*. The VGCC model equations are given below. We used the following the units: $\alpha_x(V/mV)$ $[ms^{-1}]$ and $\beta_x(V/mV)$ $[ms^{-1}]$.

| R-type h-gate rates | L-type rates |
|---|---|

$$\tau_h^{R\star} = 100$$

$$h_{inf}^{R\star}(V_{sp}) = \frac{1}{1+e^{\frac{V_{sp}+39}{9.2}}}$$

$$\alpha_h^R(V_{sp}) = \frac{h_{inf}^R}{\tau_h^R}$$

$$\beta_h^R(V_{sp}) = \frac{1-h_{inf}^R}{\tau_h^R}$$

$$\alpha^L(V_{sp}) = \frac{0.83}{1+e^{\frac{13.7-V_{sp}}{6.1}}}$$

$$\beta_1^L(V_{sp}) = \frac{0.53}{1+e^{\frac{V_{sp}-11.5}{6.4}}}$$

$$\beta_2^L(V_{sp}) = \frac{1.86}{1+e^{\frac{V_{sp}-18.8}{6.17}}}$$

*Continued*

| T-type h-gate rates |
|---|

$$\tau_h^{T\star} = 50$$

$$h_{inf}^{T\star}(V_{sp}) = \frac{1}{1+e^{\frac{V_{sp}+70}{6.5}}}$$

$$\alpha_h^T(V_{sp}) = \frac{h_{inf}^T}{\tau_h^T}$$

$$\beta_h^T(V_{sp}) = \frac{1-h_{inf}^T}{\tau_h^T}$$

| R-type m-gate rates | T-type m-gate rates |
|---|---|

$$\beta_m^{R\star} = 40 \qquad\qquad \beta_m^{T\star} = 1$$

$$m_{inf}^{R\star} = \frac{1}{1+e^{\frac{3-10}{8}}} \qquad\qquad m_{inf}^{T\star} = \frac{1}{1+e^{\frac{-32+20}{7}}}$$

$$\alpha_m^{R\star}r = \beta_m^{R\star} \cdot \frac{m_{inf}^{R\star}}{1-m_{inf}^{R\star}} \qquad\qquad \alpha_m^{T\star}r = \beta_m^{T\star} \cdot \frac{m_{inf}^{T\star}}{1-m_{inf}^{T\star}}$$

$$\tau_m^R = \frac{1}{\alpha_m^{R\star}+\beta_m^{R\star}} \qquad\qquad \tau_m^T = \frac{1}{\alpha_m^{T\star}+\beta_m^{T\star}}$$

$$m_{inf}^R = \frac{1}{1+e^{\frac{3-V_{sp}}{8}}} \qquad\qquad m_{inf}^T = \frac{1}{1+e^{\frac{-32-V_{sp}}{7}}}$$

$$\alpha_m^R(V_{sp}) = \frac{m_{inf}^R}{\tau_m^R} \qquad\qquad \alpha_m^T(V_{sp}) = \frac{m_{inf}^T}{\tau_m^T}$$

$$\beta_m^R(V_{sp}) = \frac{1-m_{inf}^R}{\tau_m^R} \qquad\qquad \beta_m^T(V_{sp}) = \frac{1-m_{inf}^T}{\tau_m^T}$$

## VGCC and temperature

We used the same temperature factor for every VGCC subtype, respectively $\rho_f^{VGCC}$ and $\rho_b^{VGCC}$ (see *Figure 17f*), as follows:

$$\rho_f^{VGCC} = 2.503 - \frac{0.304}{1+e^{1.048\cdot(T-30.668^\circ C)}}, \qquad \rho_b^{VGCC} = 0.729 + \frac{3.225}{1+e^{-0.330\cdot(T-36.279^\circ C)}}.$$

The VGCC subtypes have different sensitivities to temperature, with temperature factors for decay times ranging from 2 (*Iftinca et al., 2006*) to 50-fold (*Peloquin et al., 2008*). T-type isoforms further complicate the issue: different isoforms can have temperature factors that either speed up or slow down the kinetics. For instance, when passing from room to physiological temperatures, the Ca3.3 isoform has a closing rate ~50% faster (*Iftinca et al., 2006*), but the Ca3.1 isoform becomes ~15% slower. For simplicity in our model, the same temperature factor was adopted for all VGCC subtypes.

## VGCC currents

The VGCC currents are estimated using the GHK (*Equation 14*), as follows:

$$I_T \quad = \gamma_T \cdot \Phi_{Ca} \cdot O_T \tag{16}$$

$$I_R \quad = \gamma_R \cdot \Phi_{Ca} \cdot O_R \tag{17}$$

**Table 9.** VGCC parameters.

The number of VGCC was set to 3 to reproduce the calcium dynamics measured with a dye as in *Figure 12* (*Tigaret et al., 2016*).

| Name | Value | Reference |
|---|---|---|
| VGCC | | |
| VGCC T-type conductance | $\gamma_{CaT} = 12 \cdot 10^3 \; nS$ | same as *Magee and Johnston, 1995* |
| VGCC R-type conductance | $\gamma_{CaR} = 17 \cdot 10^3 \; nS$ | same as *Magee and Johnston, 1995* |
| VGCC L-type conductance | $\gamma_{CaL} = 27 \cdot 10^3 \; nS$ | same as *Magee and Johnston, 1995* |
| number of VGCCs | 3 for each subtype | 1–20 *Higley and Sabatini, 2012* |

$$I_L \quad = \gamma_L \cdot \Phi_{Ca} \cdot (O_{L1} + O_{L2}) \tag{18}$$

*Table 9* presents the parameters to model the VGCC channels. VGCC rates and temperature factors are shown in *Figure 17*.

## SK channel

The small potassium (SK) channel produces hyperpolarizing currents which are enhanced in the presence of intracellular calcium elevations. We included SK channels to incorporate a key negative feedback loop between spine calcium and voltage due to the tight coupling that exists between SK channels and NMDArs (*Adelman et al., 2012*; *Griffith et al., 2016*). Although there are a few publications on the single channel recording of SK channels (*Hirschberg et al., 1998*; *Hirschberg et al., 1999*) and at least one stochastic model of SK channel (*Stanley et al., 2011*), we chose to model SK channels deterministically. In tests, we found that this assumption had only a negligible impact on the outcomes of plasticity protocols (data not shown). Although SK channels can additionally be regulated by metabotropic glutamate receptors and muscarinic receptors (*Tigaret et al., 2016*), we did not include these regulatory steps in the model. The SK channel current was based on the description from *Griffith et al., 2016* as follows:

$$\frac{dm_{sk}}{dt} = \frac{r(Ca) \cdot \rho_f^{SK} - m_{sk}}{\tau_{SK}/\rho_b^{SK}}$$
$$r(Ca) = \frac{Ca^\sigma}{Ca^\sigma + h_{SK}^\sigma}$$
$$I_{SK} = \gamma_{SK} \cdot (E_{rev}^{SK} - V_{sp}) \cdot m_{sk} \cdot N_{SK}$$

There is little information on how temperature effects SK channel function, but *van Herck et al., 2018* suggests a left-ward shift in the SK half-activation when changing from 37°C ($h_{SK} = 0.38 \pm 0.02 \; \mu M$) to 25°C ($h_{SK} = 0.23 \pm 0.01 \; \mu M$); that is a 65% decrease. Thus, to mimic temperature dependence of SK,

**Table 10.** SK channel parameters.

| Name | Value | Reference |
|---|---|---|
| SK channel | | |
| Number of SK channels | $N_{SK} = 15$ | *Lin et al., 2008* |
| SK conductance | $\gamma_{SK} = 10^4 \; nS$ | *Maylie et al., 2004* |
| SK reversal potential | $E_{rev}^{SK} = -90$mV | *Griffith et al., 2016* |
| SK half-activation | $h_{SK} = 0.333 \; \mu M$ | *Griffith et al., 2016* |
| SK half-activation slope | $\sigma = 6$ | 4 *Griffith et al., 2016* |
| SK time constant | $\tau_{SK} = 0.0063 \; s$ | *Griffith et al., 2016* |

we decrease the decay time of the SK hyperpolarizing current by a factor of two when passing from physiological to room temperature.

$$\rho_b^{SK} = 149.37 - \frac{147.61}{1 + e^{0.093 \cdot (T - 98.85°C)}}, \qquad \rho_f^{SK} = 0.005 + \frac{2.205}{1 + e^{-0.334 \cdot (T - 25.59°C)}}$$

*Table 10* presents the parameters to model the SK channel.

## Enzymes - CaM, CaN, and CaMKII

To model enzyme dynamics, we adapted a monomeric CaM-CaMKII Markov chain from *Chang et al., 2019* which was built on the model by *Pepke et al., 2010*. Our adaptation incorporates a simplified CaN reaction which only binds to fully saturated CaM, i.e. CaM bound to four calcium ions on its N and C terminals (see Markov chain in *Figure 18*). A consequence of the Pepke coarse-grained model is that calcium binds and unbinds simultaneously from the CaM terminals (N,C). We assumed a lack of dephosphorylation reaction between CaMKII and CaN since *Otmakhov et al., 2015* experimentally suggested that no known phosphatase affects CaMKII decay time which is probably caused only by CaM untrapping (*Otmakhov et al., 2015*). This was previously theorized in the Michalski's model *Michalski, 2013*, and it is reflected in Chang data (*Chang et al., 2019*; *Chang et al., 2017*). The structure of the corresponding Markov chain is shown in *Figure 18*.

*Chang et al., 2019* data provides a high-temporal resolution fluorescence measurements for CaMKII in dendritic spines of rat CA1 pyramidal neurons and advances the description of CaMKII self-phosphorylation (at room temperature). We modified Chang's model of CaMKII unbinding rates $k_2, k_3, k_4, k_5$ to fit CaMKII dynamics at room/physiological temperature as shown by *Chang et al., 2017* supplemental files. Previous modelling of CaMKII *Chang et al., 2019*; *Pepke et al., 2010* used a stereotyped waveform with no adaptation to model calcium. Our contribution to CaMKII modelling was to use calcium dynamics sensitive to the experimental conditions to reproduce CaMKII data, therefore, allowing us to capture physiological temperature measurements from *Chang et al., 2017*. Note that the CaMKII dynamic has two time scales and we capture only the fastest timescale which ends after stimulation ceases (at 60 s). The slowest dynamic occurs at the end of the stimulus, close to the maximum (*Figure 19a*). This may be caused by the transient volume increase in the dendritic spine as measured by *Chang et al., 2017*. *Table 11* shows the concentration of the enzymes and *Table 12* shows the parameters to model enzymes reactions in shown in *Figure 18*.

We provide an example of equation describing the binding reaction associated to the state *CaM*0. Note that these equations are automatically generated by the code which implements *Table 12*.

$$\frac{CaM0}{dt} = \quad -\frac{k_f^{2C}}{2} \cdot CaM0 \cdot Ca^2 + k_b^{2C} \cdot CaM2C$$
$$-\frac{k_f^{2N}}{2} \cdot CaM0 \cdot Ca^2 + k_b^{2N} \cdot CaM2N$$
$$-k_f^{CaM0} \cdot CaM0 \cdot mKCaM + k_b^{CaM0} \cdot KCaM0 + k2 \cdot PCaM0.$$

The CaN concentration was chosen as the total concentration used in a previous model (*Stefan et al., 2008*) (1.6 μM) scaled by a factor of 12 due to a higher CaN concentration in dendritic spines (*Goto et al., 1986*; *Baumgärtel and Mansuy, 2012*) and taking into account the discrepancy between different CaN concentration studies (*Kuno et al., 1992*; *Goto et al., 1986*). *Kuno et al., 1992* proposes 9.6 μg/mg (7.0+2.6 μg/mg for Aα and Aβ isoforms) for the catalytic subunit A of CaN (CaNA) in the hippocampus, while *Goto et al., 1986* proposes 1.45 μg/mg (presumably for both isoforms). There is therefore a lack of consensus on CaN concentration in neurons, which seems to range between 1 and 10 μg/mg. However, models of CaN in spines (*Stefan et al., 2008*) use low values of CaN concentration (e.g. 1.6 μM), without adjusting for the fact that these values were estimated from measurements from the entire neuropil. There is little information on CaN concentration in spines, but *Kuno et al., 1992* note that the concentration of CaN is 50% to 84% higher in synaptosomes than in neuronal nuclei. With this information in mind, we set CaN spine concentration 20 μM in our model. CaN was entirely activated through CaM for the following reason: CaNA is activated by calcium-CaM in a highly cooperative manner (Hill coefficient 2.8–3), whereas the activation of CaN by calcium (via CaNB) is at most 10% of that achieved with CaM (*Stemmer and Klee, 1994*). In other words, CaNA affinity for CaM is 16 nM to 26 pM (*Creamer, 2020*), while CaNB affinity for calcium ranges from 15 μM to 24 nM

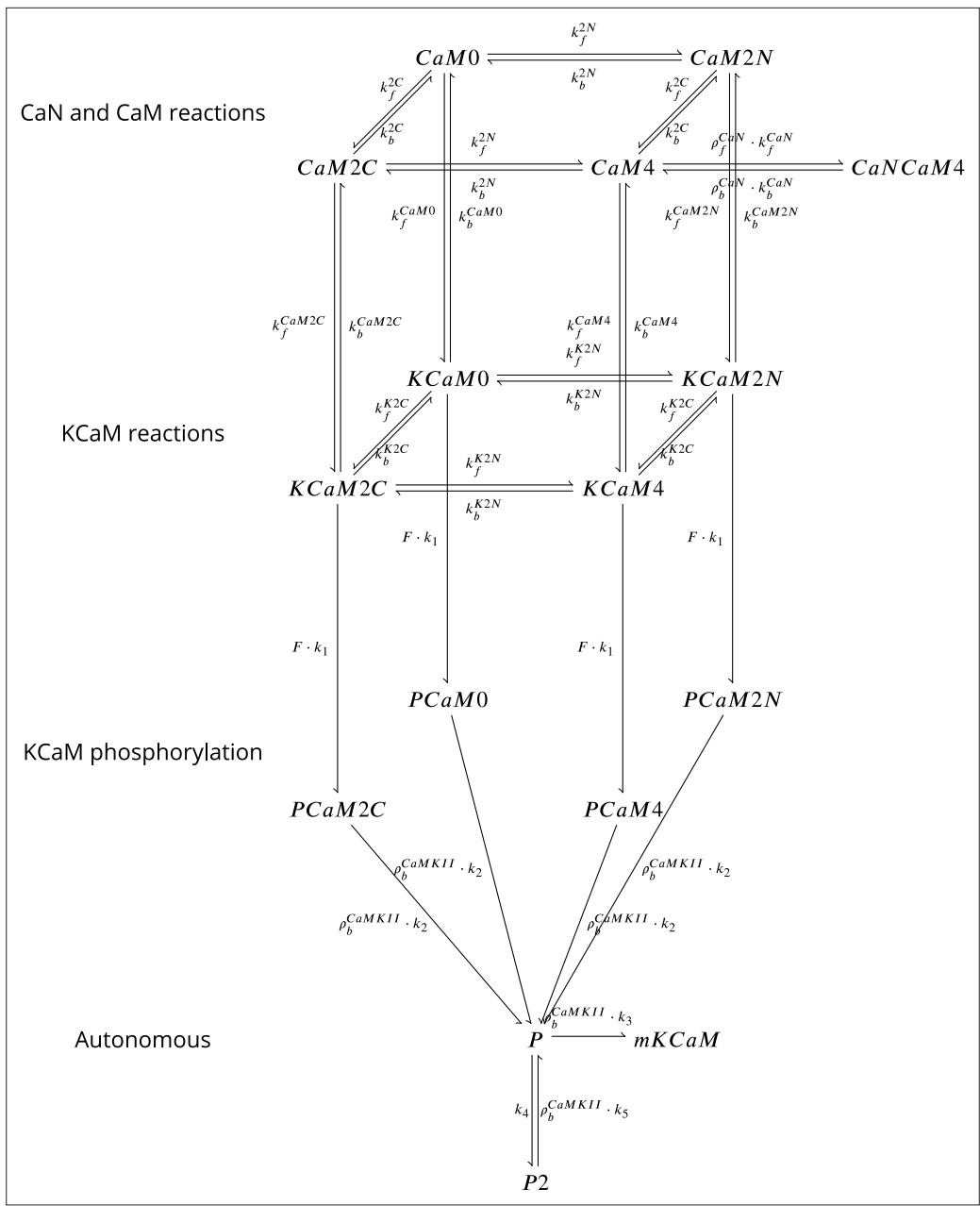

**Figure 18.** Coarse-grained model of CaM, CaMKII, and CaN adapted from *Chang et al., 2019* and *Pepke et al., 2010*. *Figure 18* is adapted from Figure 5 from *Pepke et al., 2010*. Reaction from the CaM-Ca reactions (first layer) are attributed to 2Ca release and binding from different CaM saturation states CaM2C (2Ca bound to terminal C), CaM2N (2Ca bound to terminal N), CaM0 (no calcium bound), CaM4(Ca bound to both C and N terminal). Note that CaN is allowed to bind only to fully saturated CaM. Activated CaN is represented by the state CaNCaM4. Reactions between the first (CaM-Ca reactions) and the second layer (KCaM-Ca reactions) represent the binding of free/monomeric CaMKII (mKCaM) (*Pepke et al., 2010*) to different saturation levels of CaM. Reactions within the layer KCaM-Ca represent the binding of calcium to Calmodulin bound to CaMKII (KCaM0, KCaM2C, KCaM2N, KCaM4). Transition of layer KCaM-Ca reactions to layer KCaM-phosphorylation represents CaMKII bound to CaM that became phosphorylated (PCaM states) (*Pepke et al., 2010*; *Chang et al., 2017*; *Chang et al., 2019*). PCaM can become self-phosphorylated (Autonomous layer with P and P2) and release CaM. Once the KCaM deactivates from autonomous states, it returns to free monomeric CaMKII (mKCaM). The CaMKII activity in this work represent the states (KCaM +PCaM + P + P2). See *Chang et al., 2019* for further explanation on this system. CaNCaM4 represents the CaN activity. For graphical reasons, we could not show the complete list of reactions as given in *Table 12*.

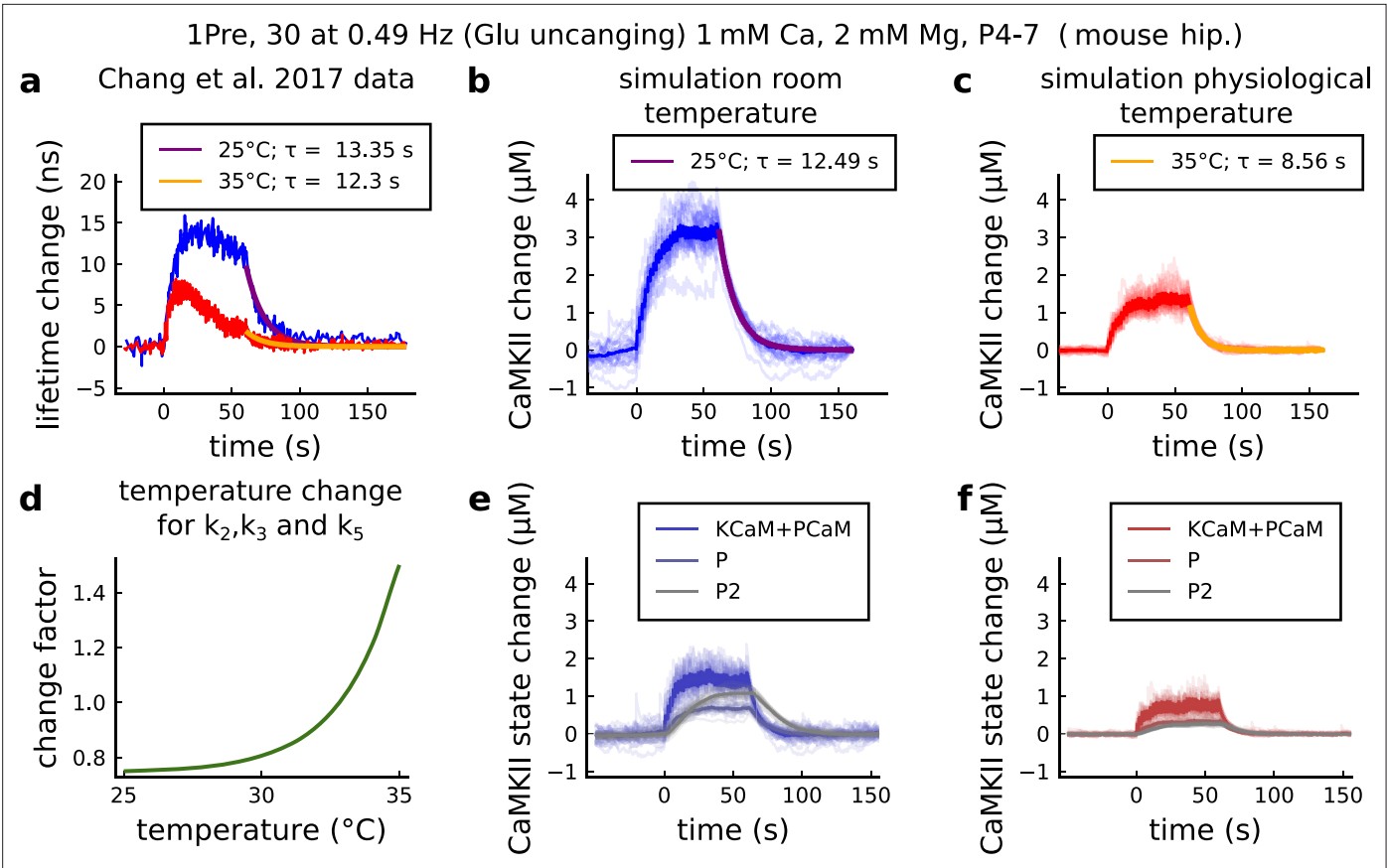

**Figure 19.** CaMKII temperature changes in the model caused by 1Pre, 30 at 0.49 Hz with glutamate uncaging (no failures allowed), 1 mM Ca, 2 mM Mg, P4-7 organotypic slices from mouse hippocampus. (**a**) CaMKII fluorescent probe lifetime change measured by *Chang et al., 2017* for 25°C (blue) and 35°C (red). The decay time ($\tau$) was estimated by fitting the decay after the stimulation (30 pulses at 0.49 Hz) using a single exponential decay, $y = a \cdot e^{-t \cdot b}$.(**b**) Simulation of the CaMKII concentration change (with respect to the baseline) at 25°C in response to same protocol applied in the panel **a**. The simulations on the panels **b, c, e, f** show the mean of 20 samples. (**c**) Same as in panel b but for 35°C. (**d**) Estimated temperature change factor for the dissociation rates $k_2$, $k_3$, and $k_5$ in the Markov chain in *Figure 18*. (**e**) Change in the concentration of the CaMKII states (25°C) which are summed to compose CaMKII change in the panel **b**. (**f**) Same as in panel e for 35°C with reference to the panel **c**.

(*Kakalis et al., 1995*). CaN decay time was modeled using experimental spine CaN activity dynamics measured in *Fujii et al., 2013*.

## The lack of reactions between CaN and CaMKII

The protein phosphatases responsible for CaMKII dephosphorylation have not been established unequivocally (*Lisman, 1989*). Our model of CaMKII is based directly on a quantitative model fit to FRET imaging data (*Chang et al., 2017*; *Chang et al., 2019*), which implicitly account for the effects of any 'hidden' phosphatases, absorbing their contribution into the decay rates of the CaMKII activity. As pointed out by *Otmakhov et al., 2015*, FRET sensor imaging of CaMKII activity unfortunately

**Table 11.** Concentration of each enzyme.

| Name | Value | Reference |
|---|---|---|
| Enzyme concentrations | | |
| Free CaM concentration (spine) | $CaM_{con} = 30\,\mu M$ | *Kakiuchi et al., 1982* |
| Free KCaM concentration (spine) | $mKCaM_{con} = 70\,\mu M$ | *Feng et al., 2011*; *Lee et al., 2009* |
| Free CaN spine concentration (spine) | $mCaN_{con} = 20\,\mu M$ | >10 μM (estimation from *Kuno et al., 1992*) |

**Table 12.** Parameters for the coarse-grained model published in **Pepke et al., 2010** and adapted by **Chang et al., 2019** and this work.

**Pepke et al., 2010** rate adaptation for the coarse-grained model $adapt(a, b, c, d, Ca) = \frac{a \cdot b}{c + d \cdot Ca}$. Refer to **Figure 18** for definition of variables.

| REACTIONS | Value | Reference |
|---|---|---|
| **Coarse-grained model, CaM-Ca reactions** | | |
| CaM0+2 Ca⇒ CaM2C<br>CaM2N+2 Ca⇒ CaM4 | $k_f^{2C} = adapt(k_{on}^{1C}, k_{on}^{2C}, k_{off}^{1C}, k_{on}^{2C}, Ca)$ | *Pepke et al., 2010* |
| CaM0+2 Ca⇒ CaM2N<br>CaM2C+2 Ca⇒ CaM4 | $k_f^{2N} = adapt(k_{on}^{1N}, k_{on}^{2N}, k_{off}^{1N}, k_{on}^{2N}, Ca)$ | *Pepke et al., 2010* |
| CaM2C⇒ CaM0+2 Ca<br>CaM4⇒ CaM2N+2 Ca | $k_b^{2C} = adapt(k_{off}^{1C}, k_{off}^{2C}, k_{off}^{1C}, k_{on}^{2C}, Ca)$ | *Pepke et al., 2010* |
| CaM2N⇒ CaM0+2 Ca<br>CaM4⇒ CaM2C+2 Ca | $k_b^{2N} = adapt(k_{off}^{1N}, k_{off}^{2N}, k_{off}^{1N}, k_{on}^{2N}, Ca)$ | *Pepke et al., 2010* |
| | $k_{on}^{1C} = 5\ \mu M^{-1}s^{-1}$ | 1.2 to 9.6 µM⁻¹s⁻¹ (*Pepke et al., 2010*) |
| | $k_{on}^{2C} = 10\ \mu M^{-1}s^{-1}$ | 5 to 35 µM⁻¹s⁻¹ (*Pepke et al., 2010*) |
| | $k_{on}^{1N} = 100\ \mu M^{-1}s^{-1}$ | 25 to 260 µM⁻¹s⁻¹ (*Pepke et al., 2010*) |
| | $k_{on}^{2N} = 200\ \mu M^{-1}s^{-1}$ | 50 to 300 µM⁻¹s⁻¹ (*Pepke et al., 2010*) |
| | $k_{off}^{1C} = 50\ s^{-1}$ | 10 to 70 s⁻¹ (*Pepke et al., 2010*) |
| | $k_{off}^{2C} = 10\ s^{-1}$ | 8.5 to 10 s⁻¹ (*Pepke et al., 2010*) |
| | $k_{off}^{1N} = 2000\ s^{-1}$ | $1 \cdot 10^3$ to $4 \cdot 10^3$ s⁻¹ (*Pepke et al., 2010*) |
| | $k_{off}^{2N} = 500\ s^{-1}$ | $0.5 \cdot 10^3$ to $> 1.10^3$ s⁻¹ (*Pepke et al., 2010*) |
| **Coarse-grained model, KCaM-Ca reactions** | | |
| KCaM0+2 Ca⇒ KCaM2C<br>KCaM2N+2 Ca⇒ KCaM4 | $k_f^{K2C} = adapt(k_{on}^{K1C}, k_{on}^{K2C}, k_{off}^{K1C}, k_{on}^{K2C}, Ca)$ | *Pepke et al., 2010* |
| KCaM0+2 Ca⇒ KCaM2N<br>KCaM2C+2 Ca⇒ KCaM4 | $k_f^{K2N} = adapt(k_{on}^{K1N}, k_{on}^{K2N}, k_{off}^{K1N}, k_{on}^{K2N}, Ca)$ | *Pepke et al., 2010* |
| KCaM2C⇒ KCaM0+2 Ca<br>KCaM4⇒ KCaM2N+2 Ca | $k_b^{K2C} = adapt(k_{off}^{K1C}, k_{off}^{K2C}, k_{off}^{K1C}, k_{on}^{K2C}, Ca)$ | *Pepke et al., 2010* |
| KCaM2N⇒ KCaM0+2 Ca<br>KCaM4⇒ KCaM2C+2 Ca | $k_b^{K2N} = adapt(k_{off}^{K1N}, k_{off}^{K2N}, k_{off}^{K1N}, k_{on}^{K2N}, Ca)$ | *Pepke et al., 2010* |
| | $k_{on}^{K1C} = 44\ \mu M^{-1}s^{-1}$ | *Pepke et al., 2010* |
| | $k_{on}^{K2C} = 44\ \mu M^{-1}s^{-1}$ | *Pepke et al., 2010* |
| | $k_{on}^{K1N} = 76\ \mu M^{-1}s^{-1}$ | *Pepke et al., 2010* |
| | $k_{on}^{K2N} = 76\ \mu M^{-1}s^{-1}$ | *Pepke et al., 2010* |
| | $k_{off}^{K1C} = 33\ s^{-1}$ | *Pepke et al., 2010* |
| | $k_{off}^{K2C} = 0.8\ s^{-1}$ | 0.49 to 4.9 s⁻¹ (*Pepke et al., 2010*) |
| | $k_{off}^{K1N} = 300\ s^{-1}$ | *Pepke et al., 2010* |

*Table 12 continued on next page*

*Table 12 continued*

| REACTIONS | Value | Reference |
|---|---|---|
| | $k_{off}^{K2N} = 20\ s^{-1}$ | 6 to 60 s$^{-1}$ (*Pepke et al., 2010*) |
| **Coarse-grained model, CaM-mKCaM reactions** | | |
| CaM0+mKCaM⇒ mKCaM0 | $k_f^{CaM0} = 0.0038\ \mu M^{-1}s^{-1}$ | *Pepke et al., 2010* |
| CaM2C+mKCaM⇒ mKCaM2C | $k_f^{CaM2C} = 0.92\ \mu M^{-1}s^{-1}$ | *Pepke et al., 2010* |
| CaM2N+mKCaM⇒ mKCaM2N | $k_f^{CaM2N} = 0.12\ \mu M^{-1}s^{-1}$ | *Pepke et al., 2010* |
| CaM4+mKCaM ⇒ mKCaM4 | $k_f^{CaM4} = 30\ \mu M^{-1}s^{-1}$ | 14 to 60 μM$^{-1}$s$^{-1}$ (*Pepke et al., 2010*) |
| mKCaM0⇒ CaM0+mKCaM | $k_b^{CaM0} = 5.5\ s^{-1}$ | *Pepke et al., 2010* |
| mKCaM2C⇒ CaM2C+mKCaM | $k_b^{CaM2C} = 6.8\ s^{-1}$ | *Pepke et al., 2010* |
| mKCaM2N⇒ CaM2N+mKCaM | $k_b^{CaM2N} = 1.7\ s^{-1}$ | *Pepke et al., 2010* |
| mKCaM4 ⇒ CaM0+mKCaM | $k_b^{CaM4} = 1.5\ s^{-1}$ | 1.1 to 2.3 s$^{-1}$ (*Pepke et al., 2010*) |
| **Coarse-grained model, self-phosphorylation reactions** | | |
| KCaM0⇒ PCaM0 KCaM2N⇒ PCaM2N KCaM2C⇒ PCaM2C KCaM4⇒ PCaM4 | $k_1 = 12.6\ s^{-1}$ | *Chang et al., 2019* |
| Fraction of activated CaMKII | $F = CaMKII/mKCaM_{con}$ | see *Equation 19* (*Chang et al., 2019*) |
| PCaM0⇒ *P*+CaM0 PCaM2N⇒ *P*+CaM2N PCaM2C⇒ *P*+CaM2C PCaM4⇒ *P*+CaM4 | $k_2 = 0.33\ s^{-1}$ | 0.33 $s^{-1}$; adapted from *Chang et al., 2019* |
| *P*⇒mKCaM | $k_3 = 4 \cdot 0.17 s^{-1}$ | 0.17$s^{-1}$ adapted from *Chang et al., 2019* |
| *P*⇒P2 | $k_4 = 4 \cdot 0.041 s^{-1}$ | 0.041$s^{-1}$ adapted from *Chang et al., 2019* |
| P2⇒P | $k_5 = 8 \cdot 0.017 s^{-1}$ | 0.017$s^{-1}$ adapted from *Chang et al., 2019* |
| **Calcineurin model, CaM-CaM4 reactions** | | |
| CaM4+mCaN⇒mCaNCaM4 | $k_f^{CaN} = 10.75\ \mu M^{-1}s^{-1}$ | 46 $\mu M^{-1}s^{-1}$ (*Quintana et al., 2005*) |
| mCaNCaM4⇒CaM4+mCaN | $k_b^{CaN} = 0.02\ s^{-1}$ | fit (*Fujii et al., 2013*) see *Figure 20* |

does not capture the identity of the phosphatases involved in the dephosphorylation of CaMKII. More specifically, *Otmakhov et al., 2015* observed no significant changes in the decay constant of their CaMKII FRET sensor when selectively inhibiting PP1 and PP2A. Given that these two phosphatases are widely used in models to determine plasticity, we believe that our model is more aligned with data of CaMKII activity in vivo.

Yet, our decision to include CaN in the model was determined by the evidence supporting CaN as the strongest candidate for calcium-sensitive protein phosphatase in the brain (*Baumgärtel and Mansuy, 2012*). Furthermore, the central role of CaN in synaptic plasticity has been demonstrated both pharmacologically and with genetic manipulation (*Onuma et al., 1998*; *Malleret et al., 2001*).

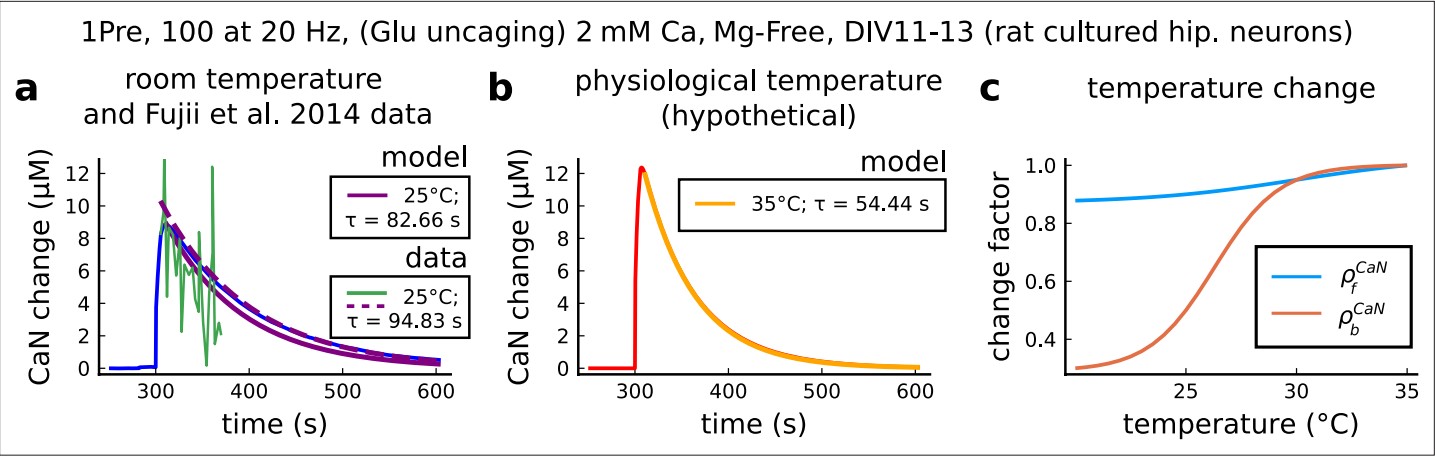

**Figure 20.** CaN temperature changes in our model caused by 1Pre, 100 at 20 Hz with glutamate uncaging (no failures allowed), 2 mM Ca, Mg-free, 11–13 days in vitro. (**a**), Simulated CaN change (blue solid line) in response to the same stimuli of the CaN measurement from **Fujii et al., 2013** RY-CaN fluorescent probe (green solid line). The decay time ($\tau$) estimated from data ($y = a \cdot e^{-t \cdot b}$) is 94.83 s (dashed purple line) and 82.66 s for our model (solid purple line). (**b**), Simulated CaN change for physiological temperature with decay time of 54.44 s. (**c**), Temperature change, $\rho_f^{CaN}$ and $\rho_b^{CaN}$, applied to CaN association and dissociation rates.

## Temperature effects on enzymatic activity

We included temperature factors in the coarse-grained model using Chang's data (**Chang et al., 2019**), as shown in **Figure 19**. For CaMKII, we fit the modified dissociation rates of the phosphorylation states $k_2$, $k_3$, and $k_5$ to match the data on relative amplitude and decay time using the following logistic function:

$$\rho_b^{CaMKII} = 162.171 - \frac{161.426}{1 + e^{0.511 \cdot (T - 45.475°C)}}.$$

For CaN, we fit the **Fujii et al., 2013** data at 25°C as seen in **Figure 20a**. However, since CaN-CaM dissociation rates at physiological temperatures were not reported, we set the temperature factor to CaN that fits the outcomes of the protocols that we aimed to reproduce. A reference value from the CaN-AKAP79 complex (**Li et al., 2012**) showed a $Q_{10} = 4.46 = (2.19\ s^{-1}/9.78\ s^{-1})$, which is nearly the temperature factor used in our model for CaN. Therefore, both the association and dissociation rates are modified using the following logistic functions:

$$
\begin{aligned}
\rho_f^{CaN} &= 2.503 - \frac{0.304}{1 + e^{1.048 \cdot (T - 30.668°C)}}\\
\rho_b^{CaN} &= 0.729 + \frac{3.225}{1 + e^{-0.330 \cdot (T - 36.279°C)}}.
\end{aligned}
$$

## Positioning of the plasticity regions

**Tigaret et al., 2016** LTP protocols were used to set the LTP region and as a first approximation of the LTD region. See **Figure 21 Top**. **Dudek and Bear, 1992**, **Dudek and Bear, 1993** and **Inglebert et al., 2020** were used to further define the LTD region. See **Figure 21 Middle and Bottom**. We highlight further a few points. For simplicity, we positioned the right border of the LTD region at the left border of the LTP region. The bottom part of the geometrical readout, under 4 μM of CaMKII, does not code for any dynamics. Note that some protocols may also enter and leave the plasticity regions multiple times, for example, TBS in **Figure 5b**, protocols in between LTD/LTP region in **Figure 3d** and age related LTD in **Figure 5f**. Because of this, we created an integrate and leak variable instead of using the time spent for predicting plasticity (see next section). This way, only after a certain time spent in a region would the synaptic weights start to change, as in Figure 23h. The coordinates of the plasticity regions are given in the last two rows of **Table 13**.

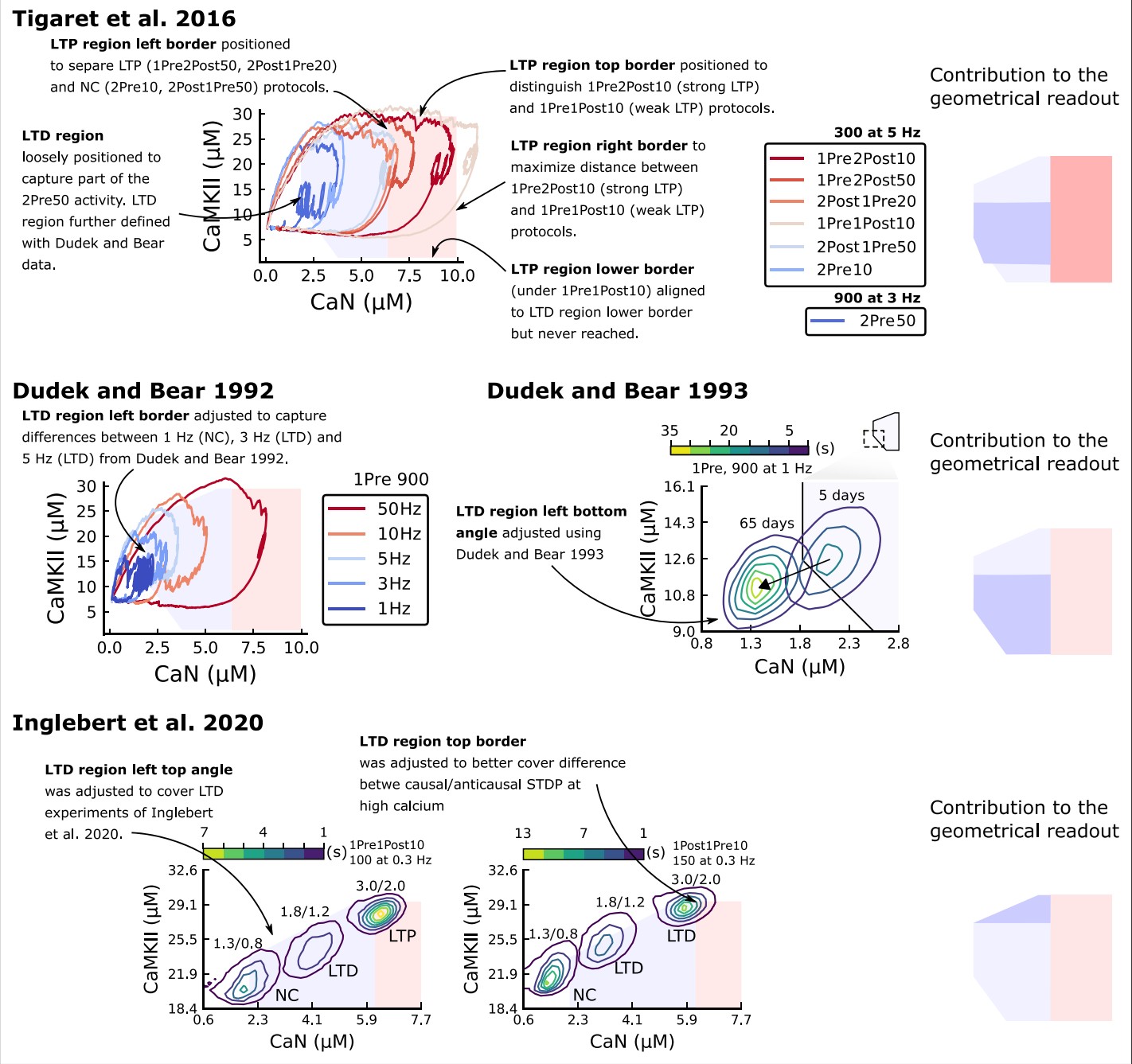

**Figure 21.** Positioning the plasticity regions. The figure shows how *Tigaret et al., 2016*, *Dudek and Bear, 1992*, *Dudek and Bear, 1993* and *Inglebert et al., 2020* contributes to define the plasticity regions. In summary, *Tigaret et al., 2016* data was used to define the LTP region, and *Dudek and Bear, 1992*, *Dudek and Bear, 1993*, *Inglebert et al., 2020* data were used to define the LTD region.

## Geometrical readout

We describe here the geometrical readout mechanism which allows for plasticity outcome assignment. First, we define the following variables which are representative of 'active CaMKII' and 'active CaN':

$$\text{LTD} \underset{D_{rate}(act_D)}{\overset{P_{rate}(act_P)}{\rightleftharpoons}} \text{NC} \underset{D_{rate}(act_D)}{\overset{P_{rate}(act_P)}{\rightleftharpoons}} \text{LTP}$$

**Figure 22.** Plasticity Markov Chain.

Active CaN

$$CaN = CaN4$$

Active CaMKII

$$
\begin{aligned}
KCaM &= KCaM0 + KCaM2C + KCaM2N + KCaM4 \\
PCaM &= PCaM0 + PCaM2C + PCaM2N + PCaM4 \\
CaMKII &= KCaM + PCaM + P + P2.
\end{aligned}
$$

(19)

The first two equations of (20) represent the total activation of K and P components of CaMKII associated with CaM as defined in the code that accompanies *Chang et al., 2019*. The last equation is from *Chang et al., 2019*.

Calcium entry in the spine initiates a cascade of events that ultimately leads to long term plasticity changes. Specific concentrations of CaMKII and CaN trigger activation functions $act_D$ and $act_P$ when they belong to one of the two polygonal regions (P and D), termed plasticity regions in the main text:

$$
\begin{aligned}
\frac{d}{dt} act_D &= a_D \cdot 1_D - b_D \cdot (1 - 1_D) \cdot act_D \\
\frac{d}{dt} act_P &= a_P \cdot 1_P - b_P \cdot (1 - 1_P) \cdot act_P.
\end{aligned}
$$

The variables $act_D$ and $act_P$ act as low pass filters of CaMKII and CaN activities with some memory of previous passages in the respective plasticity regions. To specify the LTP/LTD rates, termed $D_{rate}$ and $P_{rate}$, we use the activation functions, $act_D$ and $act_P$, as follows:

$$
\begin{aligned}
P_{rate}(act_P) &= t_P^{-1} \frac{act_P^2}{act_P^2 + K_P^2} \\
D_{rate}(act_D) &= t_D^{-1} \frac{act_D^2}{act_D^2 + K_D^2}.
\end{aligned}
$$

**Table 13.** Parameters of the plasticity readout.

The variables in this table were fitted as described in the section *Positioning of the plasticity regions.*

| Name | Value |
|---|---|
| Leaking variable (a.u.) | |
| Rise constant inside the LTD region | $a_D = 100 \ a.u. \cdot s^{-1}$ |
| Rise constant inside the LTP region | $a_P = 200 \ a.u. \cdot s^{-1}$ |
| Decay constant outside the LTD region | $b_D = 2 \cdot 10^{-2} \ a.u. \cdot s^{-1}$ |
| Decay constant outside the LTP region | $b_P = 0.1 \ a.u. \cdot s^{-1}$ |
| Plasticity Markov chain | |
| LTD rate time constant | $t_D = 18 \ s$ |
| LTP rate time constant | $t_P = 13 \ s$ |
| Half occupation LTP | $K_P = 1.3 \cdot 10^4 \ a.u.$ |
| Half occupation LTD | $K_D = 8 \cdot 10^4 \ a.u.$ |
| Plasticity regions (vertices determining the polygons) | |
| LTP region (CaN, CaMKII) | [6.35,1.4], [10,1.4], [6.35,29.5], [10,29.5] |
| LTD region (CaN, CaMKII) | [6.35,1.4], [6.35,23.25], [6.35,29.5], [1.85,11.32] [1.85,23.25], [3.76,1.4], [5.65,29.5] |

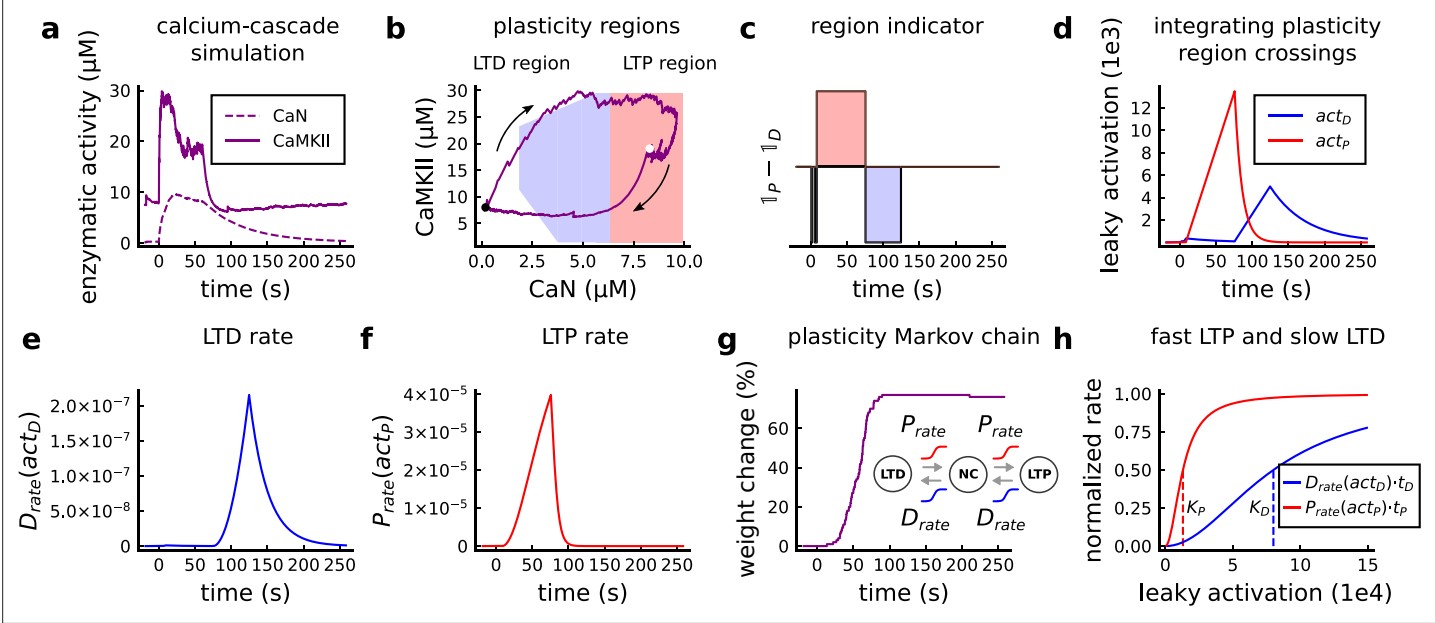

**Figure 23.** Plasticity readout for the protocol 1Pre2Post10, 300 at 5 Hz, from *Tigaret et al., 2016*. (**a**) CaMKII and CaN activity in response to protocol 1Pre2Post10. (**b**) Enzymatic joint activity in the 2D plane showing LTP and LTD's plasticity regions. The black point marks the beginning of the stimulation, and the white point shows the end of the stimulation after 60 s. (**c**) Region indicator illustrating how the joint activity crosses the LTP and the LTD regions. (**d**) The leaky activation functions are used as input to the LTP and LTD rates, respectively. The activation function has a constant rise when the joint-activity is inside the region, and exponential decay when it is out. (**e**) The LTD rate in response to the leaky activation function, $act_D$, in panel **d**. Note that this rate profile occurs after the stimulation is finished (60 s). The joint-activity is returning to the resting concentration in panel A. (**f**) The LTP rate in response to the leaky activation function, $act_P$, in panel **D**. (**g**) Outcome of the plasticity Markov chain in response to the LTD and LTP rates. The EPSP change (%) is estimated by the difference between the number of processes in the states LTP and LTD, $LTP - LTD$. (**h**) Normalized LTP and LTD rates (multiplied to their respective time constant, $t_D$, $t_P$) sigmoids. The dashed line represents the half-activation curve for the LTP and LTD rates. Note in panel **d** that the leaky activation function reaches the half-activation $K_p = 1.3e4$.

The Markov plasticity chain (see *Figure 22*) starts with initial conditions $NC = 100$, $LTD = 0$ and $LTP = 0$.

*Table 13* provides the parameters that define the boundaries of the plasticity regions (see *Figure 21* and *Figure 23b*).

*Figure 23* shows how the readout works to predict plasticity for a single orbit. *Figure 23a* shows the enzyme's activity alone which is combined to form an orbit as shown in *Figure 23b*. The region indicator of the respective orbit is shown in *Figure 23c*. Simultaneously, *Figure 23d* depicts the leaky activation $act_P$ and $act_D$, which will define the rate of plasticity induction in *Figure 23e and f*. The rates in the plasticity Markov chain will not reset to 0 if the orbit leaves the readout. The plasticity Markov chain is shown in *Figure 23g* with the prediction outcome represented as a weight change (%). *Figure 23h* shows the rate, $P_{rate}$ and $D_{rate}$, activation profile. The LTP activation rate is steep, meaning that orbits do not need to spend a long time inside the readout to promote LTP induction, while the LTD region requires five-fold longer activation times.

## Additional information

### Funding

| Funder | Grant reference number | Author |
|---|---|---|
| Medical Research Council | MR/V034111/1 | Cezar M Tigaret |
| University Côte d'Azur | ComputaBrain Idex UCA Jedi | Yuri Elias Rodrigues Hélène Marie Romain Veltz |

| Funder | Grant reference number | Author |
|---|---|---|
| Medical Research Council | MR/S026630/1 | Cian O'Donnell |
| Leverhulme Trust | RPG-2019-229 | Cian O'Donnell |

The funders had no role in study design, data collection and interpretation, or the decision to submit the work for publication.

## Author contributions

Yuri Elias Rodrigues, Conceptualization, Software, Formal analysis, Supervision, Investigation, Visualization, Writing – original draft, Project administration, Writing – review and editing; Cezar M Tigaret, Conceptualization, Software, Formal analysis, Validation, Investigation, Visualization, Writing – original draft, Writing – review and editing; Hélène Marie, Conceptualization, Supervision, Validation, Project administration, Writing – review and editing; Cian O'Donnell, Conceptualization, Software, Formal analysis, Supervision, Writing – original draft, Project administration, Writing – review and editing; Romain Veltz, Conceptualization, Software, Formal analysis, Supervision, Investigation, Writing – original draft, Project administration, Writing – review and editing

## Author ORCIDs

Yuri Elias Rodrigues ⓘ http://orcid.org/0000-0001-5730-4046
Cezar M Tigaret ⓘ http://orcid.org/0000-0001-5848-6697
Hélène Marie ⓘ http://orcid.org/0000-0003-2310-6097
Romain Veltz ⓘ http://orcid.org/0000-0003-4653-1475

## Decision letter and Author response

Decision letter https://doi.org/10.7554/eLife.80152.sa1
Author response https://doi.org/10.7554/eLife.80152.sa2

## Additional files

### Supplementary files

• MDAR checklist

### Data availability

The datasets at the basis of our model were obtained directly from the authors *Tigaret et al., 2016* or extracted from graphs in the references in *Appendix 1—table 1* using WebPlotDigitizer v 4.6 software (Rohatgi,A.)The dataset from *Tigaret et al., 2016* is freely available upon request. Concerning the dataset *Tigaret et al., 2016*, Jack Mellor the person to contact. One can contact Cezar M. Tigaret as well. The model is available on GitHub at https://github.com/rveltz/SynapseElife (copy archived at *Veltz, 2023*).

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

# Appendix 1

**Appendix 1—table 1.** Synaptic plasticity protocol parameters.

To fit the data from publications displaying a parameter interval (e.g. 70–100), we used a value within the provided limits. Otherwise, we depict in parentheses the value used to fit to the data. Further information is available in the github code and *Appendix 1—table 3*. Some of these experiments did not control AP generation following EPSP stimulation: *Mizuno et al., 2001*, *Dudek and Bear, 1992 Dudek and Bear, 1993*. We modeled this effect, described below. In addition, *Tigaret et al., 2016* used GABA(A)r blockers, which we modelled by setting the GABAr conductance to zero. Also, *Mizuno et al., 2001* LTD protocol used a partial NMDA blocker, which we modelled by reducing NMDA conductance by 97%.

| Experiment | Paper | Repetitions | Freq (Hz) | Age (days) | Temp. ($^{\circ}C$) | [Ca²⁺]o(mM) | [Mg²⁺]o(mM) |
|---|---|---|---|---|---|---|---|
| STDP | *Tigaret et al., 2016* | 300 | 5 | 56 | 35 | 2.5 | 1.3 |
| STDP | *Inglebert et al., 2020* | 100, positive delays | 0.3 | 21 | (30.45) | 1.3—3 | Ca/1.5 |
| STDP | *Inglebert et al., 2020* | 150, negative delays | 0.3 | 14 | 30 | 1.3—3 | Ca/1.5 |
| STDP | *Meredith et al., 2003* | 20 | 0.2 | 9—45 | 24—28 | 2 | 2 |
| STDP | *Wittenberg and Wang, 2006* | 70—100 | 5 | 14—21 | (22.5–23) | 2 | 1 |
| pre-burst | *Tigaret et al., 2016* | 300 and 900 | 3 and 5 | 56 | 35 | 2.5 | 1.3 |
| FDP | *Dudek and Bear, 1992* | 900 | 1—50 | 35 | 35 | 2.5 | 1.5 |
| FDP | *Dudek and Bear, 1993* | 900 | 1 | 7—35 | 35 | 2.5 | 1.5 |
| TBS | *Dudek and Bear, 1993* | 3—4 (5) epochs | 4Pre at 100 Hz, 10 x at 5 Hz | 6, 14 and 17 | 35 | 2.5 | 1.5 |
| LFS | *Mizuno et al., 2001* | 1—600 | 1 | 12—28 | (26.5–31) | 2.4 | 0 |

**Appendix 1—table 2.** Comparison of recent computational models for plasticity highlighting the experimental conditions implemented and the experiments in the hippocampus and cortex they reproduce. See *Appendix 1—table 3* for additional details on experimental conditions of experimental works.

| Model | *Graupner and Brunel, 2012* | *Ebner et al., 2019* | *Jędrzejewska-Szmek et al., 2017* | *Inglebert et al., 2020* | *Chindemi et al., 2022* | This paper |
|---|---|---|---|---|---|---|
| Model framework | Extension of *Shouval et al., 2002* | Extension of *Clopath et al., 2010* and modified from *Hay et al., 2011* | Modified from *Evans et al., 2013* | Extension of *Graupner and Brunel, 2012* | Extension of *Graupner and Brunel, 2012* | |
| **Parameter** | | | | | | |
| Temperature | Absent | Absent | Temperature corrected ion channels (but not receptors) | No temperature control needed (experiments covered are at 30° C) | Only in the GHK equation | Temperature is selectable on the dendritic spine level for ion channels, receptor and the calcium cascade |
| Development | Absent | Absent | Absent | Absent | Absent | Age is selectable and implemented by GABAr and NMDAr switch and BaP maturation |
| aCSF | Absent | Absent | Absent | Phenomenological changes in pre and post amplitudes to mimic extracellular calcium effects | In vivo or in vitro changes for release probability, calcium reversal potential on NMDAr-induced calcium influx | External Ca and Mg are selectable and affect release probability, reversal potential, NMDAr and VGCCs calcium current driving force |
| **Plasticity experiments (quant. comparisons only)** | | | | | | |
| *Sjöström et al., 2001* | X | X | | | X | |
| *Wittenberg and Wang, 2006* | X | | | | | X |
| *Wang et al., 2005* | X | | | | | |
| *Sjöström and Häusser, 2006* | X | X | | | X | |
| *Nevian and Sakmann, 2006* | | X | | | | |
| *Letzkus et al., 2006* | | X | | | | |
| *Weber et al., 2016* | | X | | | | |
| *Fino et al., 2010* | | | X | | | |
| *Pawlak and Kerr, 2008* | | | X | | | |
| *Shen et al., 2008* | | | X | | | |
| *Inglebert et al., 2020* | | | | X | | X |
| *Markram et al., 1997* | | | | | X | |
| *Rodríguez-Moreno and Paulsen, 2008* | | | | | X | |
| *Egger et al., 1999* | | | | | X | |
| *Tigaret et al., 2016* | | | | | | X |
| *Dudek and Bear, 1992* | | | | | | X |
| *Dudek and Bear, 1993* | | | | | | X |
| *Mizuno et al., 2001* | | | | | | X |
| *Meredith et al., 2003* | | | | | | X |
| *O'Connor et al., 2005* (not included due to space) | | | | | | X |
| *Bittner et al., 2017* (not included due to space) | | | | | | X |

**Appendix 1—table 3.** Comparison of the experimental conditions for the differentdatasets in Appendix 1—table 2 covering experiments from neocortex, hippocampus and striatum.

| Experimental work | Age (days) | $[Ca^{2+}]_o$ (Mm) | $[Mg^{2+}]_o$ (Mm) | Temperature (°C) |
|---|---|---|---|---|
| *Sjöström et al., 2001* | 12–21 | 2.5 | 1 | 32–34 |
| *Wittenberg and Wang, 2006* | 14–21 | 2 | 1 | 24–30 or 30–34 |
| *Wang et al., 2005* | embryonic day 17–18 | 3 | 2 | room |
| *Sjöström and Häusser, 2006* | 14–21 | 2 | 1 | 32–35 |
| *Nevian and Sakmann, 2006* | 13–15 | 2 | 1 | 32–35 |
| *Letzkus et al., 2006* | 21–42 | 2 | 1 | 34–35 |
| *Weber et al., 2016* | 49–77 | 1.25 | 1.3 or 0.1 | 32–35 |
| *Fino et al., 2010* | 15–21 | 2 | 1 | 34 |
| *Pawlak and Kerr, 2008* | 19–22 | 2.5 | 2 | 31–33 |
| *Shen et al., 2008* | 19–26 | 2 | 1 | room |
| *Inglebert et al., 2020* | 14–20 | 1.3–3.0 | Ca/1.5 | 30 |
| *Markram et al., 1997* | 14–16 | 2 | 1 | 32–34 |
| *Rodríguez-Moreno and Paulsen, 2008* | 9–14 | 2 | 2 | room |
| *Egger et al., 1999* | 12–14 | 2 | 1 | 34–36 |
| *Tigaret et al., 2016* | 50–55 | 2.5 | 1.3 | 35 |
| *Dudek and Bear, 1992* | 35 | 2.5 | 1.5 | 35 |
| *Dudek and Bear, 1993* | 7–35 | 2.5 | 1.5 | 35 |
| *Mizuno et al., 2001* | 12–28 | 2.4 | Mg-Free (most experiments) | 30 |
| *Meredith et al., 2003* | 9–45 | 2 | 2 | 24–28 |
| *O'Connor et al., 2005* | 14–21 | 2 | 1 | 27.5–32 |
| *Bittner et al., 2017* | 42–63 | 2 | 1 | 35 |

