## [Editor Report]

Synaptic plasticity is a ubiquitous but also highly complex phenomenon and developing a unifying description has been challenging. This study presents a realistic biophysical model of plasticity induction, with a novel read-out of CaMKII and Calcineurin. It is able to describe a wide range of experimental results and sets a new benchmark for realistic computational models.

---

## [Decision Letter]

**Decision letter after peer review:**

Thank you for submitting your article "A stochastic model of hippocampal synaptic plasticity with geometrical readout of enzyme dynamics" for consideration by *eLife*. Your article has been reviewed by 3 peer reviewers, including Mark van Rossum as the Reviewing Editor and Reviewer #1, and the evaluation has been overseen by Ronald Calabrese as the Senior Editor.

The reviewers have discussed their reviews with one another, and the Reviewing Editor has drafted this to help you prepare a revised submission. Overall, the reviewers saw substantial merit in the manuscript. Many of the revisions requested can be done without new simulations.

Essential revisions:

1) Regarding noise in the model.

a) Indicate the critical noise source in the model. Which of the various noise sources is most important?

b) The calcium-sensitive potassium (SK) channel has been modelled deterministically. Given that the same justification in terms of a small number of channels present in the small dendritic spine compartment applies to the SK channels as well as to the voltage-gated calcium channels and the AMPA and NMDA receptors, it is not clear why the authors have chosen a deterministic representation in the case of SK. The implications of these assumptions need to be investigated and discussed. It might be possible to rule out that SK noise matters from the first principles.

2) Many of the model parameters have been set to values previously estimated from synaptic physiology and biochemistry experiments, However, a significant number of important parameter values have been tuned to reproduce the plasticity experiments targeted in this study.

State which of the model parameters (eg Table 14) were considered to be free parameters adjusted to fit the LTP/LTD induction data, and which were assumed to be generic and therefore not adjusted when fitting experimental data.

Explain which of the plasticity outcomes have been reproduced because the parameters are chosen to do so. A clarification would have helped to substantiate the authors' conclusions. It could also form the basis for more predictions.

On a similar note, there is not much information given how the readout mechanism came about. How were the regions found?

How can experiments be used to more precisely define these regions?

3) Better explain the reason for the non-linear effects in Figure 4 and 5 and possible experimental predictions that follow from that.

Also, add experimental model predictions, for example, that firing variability can alter the rules of plasticity, in the sense that it is possible to add noise to cause LTP for protocols that did not otherwise induce plasticity.

4) Argue or show whether or not the detailed exocytosis model is needed (see below).

5) Discuss briefly why the multi-vesicular release is not considered in the presynaptic model.

6) Improve the writing (also see below). In particular, discuss the main hypotheses and assumptions underlying this work and clarify the main conclusions and goals of this modeling work. It is also important to tell the reader that 'geometric' does not imply a 'spatial' readout.

*Reviewer #1 (Recommendations for the authors):*

Regarding these points:

1. Noise. I like that the model produces realistic variability in plasticity. This is commendable. It is also interesting because the presence of noise neccessitates a more robust readout mechanism than a noise-free model.

What is the critical noise source in the model? It should be possible to determine this (I hope it is not in the readout mechanism, as that is to most unprincipled of all included noise sources).

Similarly, is the noise crucial for the model to work? Surely, as is shown the quantitative outcome of the model changes without noise, but could it have been re-adjusted? Or does the noise truly change it?

As a follow-up to the current study it might be of interest to study paired synapses as was done in Bartols and Sejnowski (*eLife*) or MOTTA and Helmstaedter (2019) and who quantified the correlation between them. This would be a further critical test of the model and could decide if the majority of the observed fluctuations in plasticity are due to heterogeneity or noise.

2. The readout. There is not much information given on how the readout mechanism came about. How were the regions found? How can experiments be used to more precisely define these regions? Related, what about the parameters in Table 14? It is fine if they were hand-tuned but just say so.

3. I thought the nonlinear effects in Figure 4 and 5 were pretty cool, but the reason behind them could be explained better, as that would help these predictions further in experiments.

While the paper is easy to read, I have quite a few -suggestions- for further improvement.

I thought the Introduction and Discussion were rather long. In particular, the Introduction is quite philosophical. After reading it, I was actually doubtful that the paper was going to show some novelty.

The J of Neurosc has 500 words for introduction and 1500 for Discussion. I think that works well.

On the other hand, some of the text is not explicit enough (the Methods, in particular, see below).

– It would help to replot the Tigaret data and the protocols.

– Some figures are better in the main text. I would suggest to move Figure 3S1 to the main text, as it is so crucial. Fig22 would also be good in the main text given the complexity of the dynamics; I was happy when I found it, but that was already pretty/too late.

– The coloring used in Fig3e is confusing.

– l372. 'pulse dependent amplification' Not quite sure what that means.

– Formatting in Table 1 is ambiguous, e.g. '100 positive delays' and '200 negative delays' belong to Inglebert , but this is hard to parse.

– l 758 onwards is somewhat typical for the confusion in the Methods. THe text does not mention the D→R transition, but it is in the table.

Then, the release is mentioned in l761, but then again in l764 ("In addition…").

The Methods section is full of such inconsistencies.

– Another example: l792 "Later we incorporate a higher number of.. parameters … with our.. model". That is poor English.

– Fig8 Caption "Our model also does not cover the aboslishing of release probability". Apart from poor English, it is confusing as it is unclear how this relates to the figure.

– Fig8. Why is there such a very sudden drop in the 50Hz curve? It looks odd to me given the dynamics.

– p24 Note that one also needs to specify the units of the alphas and betas. It is also somewhat in conflict with the common convention to call \tau_\infty m_\tau.

– Table4. The spine and dendrite have time constant of 1.5 s. That is very, very long. Why is this necessary?

– Table4. 10th line dendritic leak reversal should read differently.

– Equation1 uses square brackets to denote a concentration for outside Ca, but not for CaPre which is presumably also a concentration.

– Was spontaneous release included in the synapse?

*Reviewer #2 (Recommendations for the authors):*

The model proposed in this study is complex and multi scale combining elements from previous work and grounding the choice of many parameter values on experimental measurements and estimates. As such the tables summarising the modelling details in the Materials and methods section use different units:

1) For time – ms and s;

2) Electrical currents conductance – nS/um^2, nS and pS.

3) Concentration – μm and M.

I would suggest that the authors unify the presentation by using consistent units for all modelling components.

I have some further specific recommendation regarding the modelling description:

1) Please provide details regarding the threshold function in Equation (1).

2) Please clarify if Equation (7) has been fitted to data.

3) Please justify the calcium buffer and dye model assumptions that both bind one claim ion only. This is important as calcium indicators such as calmodulin are know to bind 4 calcium ions.

4) Could you please justify Equation (19).

*Reviewer #3 (Recommendations for the authors):*

1) Page 22, Equations. 1-2: Biologically, the effect of [Ca]_0 is certainly to increase the size of [ca^2+^] jump during an action potential. However, the size of the ca^2+^ jump in the model is normalized to 1, so the external [ca^2+^] is assumed to directly affect the dependence of release probability on [ca^2+^], without affecting the [ca^2+^] dynamics directly. This shows that the model is purely phenomenological and that the [ca^2+^]pre variable does not represent [ca^2+^] in any meaningful way. This makes it very hard to understand the model presented in Equations. 1-2, which is quite distracting, particularly since this bears a minimal impact on the main part of this work. Moreover, there are as many as 4 free parameters in the "half-activation" function h([Ca]). Including the value of cooperativity "s", the total number of parameters is equal to 5. This equals the number of data points in Figure 8e which is presumably used to calibrate the presynaptic model. The number of parameters seems very large, and the presynaptic model appears unnecessarily complicated. Even the most comprehensive, realistic biophysical models of Ca-triggered exocytosis from the calyx of Held literature do not have as many parameters. I highly suggest replacing the model with a simpler model, whose variables are biologically interpretable.

2) Given the large number of parameters in the model (with 14 Tables in the main part of the manuscript), it would be of value to clearly summarize which of the model parameters were considered to be free parameters that were adjusted to fit the LTP/LTD induction data, apart from the geometric read-out parameters, and which of the model parameters (e.g. the ionic conductances) were assumed to be generic and therefore not adjusted when fitting experimental data.

3) I suggest explaining in one sentence why the possibility of multi-vesicular release is not considered in the presynaptic model.

4) Throughout the manuscript, the verbs "predicted", "found", "hypothesized" are used very loosely when discussing the behavior of the model and its elements, rather than in describing experimentally verifiable predictions. Moreover, when used in the past tense, verbs "predicted" and "found" is suggestive of predictions that have been experimentally verified. This makes the manuscript hard to read since in some places it is hard to distinguish statements about model behavior from statements about past experimental results or verifiable predictions. Concrete comments related to this confusing phrasing follow below.

[Editors' note: further revisions were suggested prior to acceptance, as described below.]

Thank you for resubmitting your work entitled "A stochastic model of hippocampal synaptic plasticity with geometrical readout of enzyme dynamics" for further consideration by *eLife*. Your revised article has been evaluated by John Huguenard (Senior Editor) and a Reviewing Editor.

The manuscript has been improved but there are some remaining issues that need to be addressed, as outlined below:

*Reviewer #1 (Recommendations for the authors):*

The authors have addressed all my issues.

Regarding the contribution of the various noises, it would be nice to have the figure from the reply letter in the paper.

(Of course, it would have been a bit cleaner to switch the noises ON one by one, but this figure is already informative).

*Reviewer #2 (Recommendations for the authors):*

I do appreciate the amount of work done by the authors in addressing reviews comments and suggestions. The manuscript has been improved and most of my comments have been addressed satisfactorily except for the following points:

1) Regarding SK deterministic modelling and justification – it is simply not true that:

(1.1) There is "a lack of single-channel recordings of SK channels", please see

Hirschberg, B., Maylie, J., Adelman, J., and Marrion, N. (1998). Gating of recombinant small-conductance Ca-activated K^+^ channels by calcium. The Journal of General Physiology, 111(4), 565.

Hirschberg, B., Maylie, J., Adelman, J., and Marrion, N. (1999). Gating properties of single SK channels in hippocampal CA1 pyramidal neurons. Biophysical Journal, 77(4), 1905-1913.

(1.2) There is "a lack of published stochastic models of SK channel", please see

Stanley, D.A., Bardakjian, B.L., Spano, M.L. et al. Stochastic amplification of calcium-activated potassium currents in ca^2+^ microdomains. J Comput Neurosci 31, 647-666 (2011). https://doi.org/10.1007/s10827-011-0328-x

(available at https://link.springer.com/article/10.1007/s10827-011-0328-x#Equ6)

The above needs to be addressed.

2) The units for the various model parameters are still inconsistent, eg. Table 6. uses M for concentration and s for time while Table 7. – μm for concentration and ms for time. There are other instances along the lines of the above in the rest of the tables presenting model parameters throughout the manuscript.

I do know this is a matter of simple scaling but the parameter units used throughout should be consistent to avoid reproducibility issues that could be faced by other researchers and especially early career researchers studying this work in the future.

*Reviewer #3 (Recommendations for the authors):*

The authors have made a significant update to their manuscript in order to address the Reviewers' comments, adding a couple more simulation results to explain different features of the model. In general, I find this update sufficiently complete.

The only point I am still not fully satisfied with is the model describing the dependence of vesicle release probability on extracellular [ca^2+^] (Equations. 1-2, p. 23), retained from the original version of the manuscript since it appears both cumbersome and unphysiological. However, I completely agree with the Authors that correcting this would not affect the main conclusions while requiring additional but unnecessary simulations.

Therefore, with respect to the latter, I would be fully satisfied if the Authors added a single sentence on page 23, reiterating their response to this point, namely: (1) the presynaptic model for p_rel is purely phenomenological, (2) in principle, the [Ca] jump parameter δ_Ca should depend on [Ca]_0, but (3) replacing the model with a more physiological model would not affect the results, since the measured dependence of release probability on [Ca]_0 is already satisfied by this phenomenological model.

---

## [Author Response]

Essential revisions:1) Regarding noise in the model.a) Indicate the critical noise source in the model. Which of the various noise sources is most important?

To show the importance of the stochasticity of the different components of the model, we simulated three protocols of Dudek and Bear 1992 with deterministic equations. We added the results of these simulations as Figure 4 – Supplement 2. We show that, for the different protocols, if some of the channels are modelled with deterministic equations, the net effect on synapse weight differs from the expected outcome provided by the original model. We can conclude that all noise sources we introduced in our model are important.

b) The calcium-sensitive potassium (SK) channel has been modelled deterministically. Given that the same justification in terms of a small number of channels present in the small dendritic spine compartment applies to the SK channels as well as to the voltage-gated calcium channels and the AMPA and NMDA receptors, it is not clear why the authors have chosen a deterministic representation in the case of SK. The implications of these assumptions need to be investigated and discussed. It might be possible to rule out that SK noise matters from the first principles.

There are several stochastic models of AMPA and NMDA receptors based on single-channel recordings. Additionally, we had enough experimental data on single channel recordings to build a custom Markov chain model of VGCCs. For the SK channel, we could not find enough experimental data (age-dependence activity, temperature sensitivity, etc.) to custom-build a stochastic model. We thus decided to implement a deterministic model. Yet, we understand the reviewers’ comment that in theory, a stochastic model of SK channels could impact our results. We thus now provide a simulation with a stochastic model of SK, comparing it to the deterministic model implemented in the study.

We describe a minimal version of a stochastic model of SK compatible with the deterministic version. The deterministic model of SK channel fit at ~35C is described in the methods section and reproduced below:dmskdt=r(Ca)∙ρfSK−mskτSK/ρbSK

Because of the factor ρ *f^SK^* in the equation, which multiplies *r(Ca)* by ~2*,* the equation cannot be related to a 2-state Markov chain (MC). This could probably be possible with a 3-state MC but we used a different strategy. Noting that ρ ^*SK*^ ∼ 2 , we introduce a new equationdOskdt=r(Ca)−OskτSK/ρbSK,Osk=2msk

As 0 < r(Ca) < 1, it is straightforward to introduce a 2-state MC for which the above equation describes the probability of the open state. We then simulate two such independent (for a given Ca concentration) channels and approximate *m_sk_* as the sum (which belongs to [0,2Nsk]) of the open states for the 2 channels.

As the reviewer can see in Author response image 1, we do not find a major difference in the simulations of 3 protocols. Thus, we argue that adding a stochastic version of the SK channels in our current study would not fundamentally alter our main conclusions.

**Author response image 1. sa2fig1:** a comparison using Tigaret et al. 2016 1Pre2Post10 and 1Pre2Post50 protocols, and 900 at 50 Hz protocol from Dudek and Bear 1992 (100 repetitions) between the model with the deterministic SK channel (original model – blue), and the modified model including the stochastic SK channel (stochastic SK – red). Deterministic vs stochastic SK channel does not significantly modify the model’s behaviour.

To explain our rationale of using a deterministic version of SK channel, we provide this sentence in the Methods when describing SK channel model: “"Due to a lack of single-channel recordings of SK channels, and a lack of published stochastic models of SK channels, we modelled SK channels deterministically. In tests we found that this assumption had only a negligible impact on the outcomes of plasticity protocols (data not shown)" (page 40).

2) Many of the model parameters have been set to values previously estimated from synaptic physiology and biochemistry experiments, However, a significant number of important parameter values have been tuned to reproduce the plasticity experiments targeted in this study.State which of the model parameters (eg Table 14) were considered to be free parameters adjusted to fit the LTP/LTD induction data, and which were assumed to be generic and therefore not adjusted when fitting experimental data.Explain which of the plasticity outcomes have been reproduced because the parameters are chosen to do so. A clarification would have helped to substantiate the authors' conclusions. It could also form the basis for more predictions.

Most parameters were set with values previously defined by experimental work. We referred to these publications where necessary throughout the Methods and Tables in our original manuscript. For the few free parameters that were adjusted, we now provide additional information wherever necessary for the Tables concerned.

In the legend of Table 4 (neuron electrical properties), we explain which parameters are different from values obtained from the literature to fit experimental data (Golding et al. 2001; Buchanan et al. 2007).Parameters for the sodium and potassium conductance (Table 5) are labelled as generic since they are intentionally set to produce the BaP dynamics we have shown in the paper.Table 6 has no free parameters.Table 7 caption now includes a description saying ’Note that the buffer concentration, calcium diffusion coefficient, calcium diffusion time constant and calcium permeability were considered free parameters to adjust the calcium dynamics’.In Table 8 we had originally pointed out how we adapted the GluN2B rates from a published GluN2A model (Popescu et al. 2004; and Iacobucci and Popesco 2018). We now describe this adaptation in the Table 8 legend. In this Table, we now also better explain how we adjusted the NMDAr model to reflect the ratio between GluN2B and GluN2A, fitted from Sinclair et al. 2016; and the NMDAr conductance depending on calcium fitted from Maki and Popescu 2014.In Table 9 caption we now explain how the GABAr number and conductance were modified to fit GABAr currents as in Figures 15 b and e. The relevant parameters are indicated in the table.In Table 10 caption we now state the number of VGCCs per subtype that we used as a free parameter to reproduce the calcium dynamics (Figure 12).

On a similar note, there is not much information given how the readout mechanism came about. How were the regions found?

we reformatted the sections of the methods called ‘positioning of the plasticity regions’ and ‘Geometrical readout’ to extend the description of how the regions were found and used to develop the plasticity Markov chain (pages 47). We also added a new figure in the section ‘positioning of the plasticity regions’ to clearly illustrate the procedure (Figure 21). We hope that with this additional information the reviewers better understand the process by which we defined these regions.

How can experiments be used to more precisely define these regions?

In Figure 21, the parts of plasticity regions (both LTD and LTP) below a CaMKII concentration of 5 μM (never reached) were defined as continuation of the boundaries fitted with protocols. These lower boundaries could theoretically be more precisely defined if one would perform experiments with low levels of inhibition of CaMKII. We could also more precisely define these regions as function of age of animal or temperature with additional data points. Currently, we managed to fit the existing experimental variability due to age and temperature by modifying the inner workings of different components except the plasticity MC.

3) Better explain the reason for the non-linear effects in Figure 4 and 5 and possible experimental predictions that follow from that.

The origin of the non-linear effects underlying the results in Figures 4 and 5 is very intricate, with the added complication of noise. We agree with the reviewer that understanding such dynamics (non-linearities) would allow us to reach the very basic geometrical / dynamical operating regime of the model, possibly unlocking a stripped down version of the model with only phenomenological basis. This will be the subject of future work and, at this stage, we can only speculate based on the enzymatic activities in the 2D plane close to the plasticity regions. We conjecture that while non-linear effects in Figure 4 derive from various sources, they are primarily due to the interplay of presynaptic release dynamics, the NMDAr kinetics and the probability of EPSPs generating BaPs.

Regarding Figure 4 and frequency-dependent plasticity, we argue that the LTD protocols have to sustain enough vesicle recycling to stay long enough in the LTD region. In contrast, the LTP protocols have to release glutamate fast enough without depleting its reserves to reach the LTP region. To support our explanation, we have some results at high frequencies (40 Hz and above) from our simulations (Figure 4-e,f), but we have not shown these results. Indeed, we considered that long trains of 40Hz and above, including higher number of pairing repetitions, damage neurons and are thus non-physiological. However, we observed an attenuation tendency (far right part of Figure 4f graph) of the LTP strength at these higher frequencies. On this note, our model supports the view that such excessive depletion during high frequencies can be avoided by theta burst stimulation (TBS), which induces a pause in between 100 Hz bursts to allow for vesicle recovery (Figure 5b).

Also, Figure 5 expands on results obtained in Figure 4 by analyzing how varying the maturity of the synapse, by encoding age-related modifications in various model components, modifies the synapse plasticity outcomes in our model. In Figure 5 Supplement 1, we isolate which encoded age-dependent alterations are responsible for the observed nonlinearities in the developmental BCM. As depicted in Figure 5 supplement 1 d-i, BaP and NMDAr kinetics are more relevant than GABAr to influence the plasticity outcome dependent on age.

Below we discuss the experimental predictions that can be made from these interpretations of Figure 4 and 5.

In the original manuscript, in Figure 4 Supplemental 1, we provided several graphs that yield predictions, which can be tested following Dudek's experimental conditions. We now extracted some specific predictions from these graphs. We added these predictions at the end of the legend of this figure (page 54). We also better explained the content of this supplemental figure at the end of the main caption of Figure 4 (page 12).

From these simulations, we can provide several predictions. The external Mg controls the sliding threshold (panel a). By analysing how the concentration of external calcium modifies the BCM-like curve we suggest that stimulation patterns used by Dudek and Bear are unlikely to produce LTD at more physiological calcium concentrations (1-1.5mM). This also suggests that LTD is more likely to be driven by burst instead of slow firing (panel b). LTP and LTD generation are both affected by the distance of the synapses from the soma (panel c). A similar analysis was done by Ebner et al. 2019 describing a similar prediction. We show that LTD is more prone to variations of amplitude by temperature than LTP (panels d,e,f). We predict that stimuli of

Poissonian spike trains differently shape the plasticity outcome depending on the animal’s age (panels g,h,i).’

For age-dependence, we can extract predictions from Figure 5 supplemental 1, but that would only pertain to non-physiological alterations of components of the model. For instance, if BaP maturation doesn't occur, that is, BaP is always inefficient, then LTP cannot be induced by high frequencies. Also, if NMDAr are principally composed of the GluN2B subunit, meaning that the receptor does not go through the age-dependent maturation process, then LTD would remain pronounced at higher ages. We added these predictions at the bottom of caption of Figure 5 supplement 1 (page 55). We also better reference this supplemental figure at the bottom of the caption of the main Figure 5 (page 13).

Also, add experimental model predictions, for example, that firing variability can alter the rules of plasticity, in the sense that it is possible to add noise to cause LTP for protocols that did not otherwise induce plasticity.

We added model predictions for Figures 4 and 5 as described above. For experimental model predictions made from simulations in Figure 7, we added a few sentences with specific examples pertaining to the reviewer’s comment at the end of the description of this figure in the main text (page 18, line 872).

4) Argue or show whether or not the detailed exocytosis model is needed (see below).

The detailed exocytosis model we implemented was motivated by the following minimal requirements:

1. Taking into account synaptic failures, which impact STDP protocols. This is best modeled by counting the released vesicles, hence our choice of a stochastic model.

2. Vesicle depletion, which strongly shapes the plasticity outcome of frequency/pulse number repetition dependent protocols.

3. Influence of external calcium on release probability, a component necessary to model Iglebert et al. 2020.

In our original manuscript, we had shown a direct comparison of our model (discrete presynaptic release) with a deterministic version of release (averaged presynaptic release), Figure 3 Supplement 1 (page 51). As there are no failures in the deterministic version, the enzymes are over-activated and the trajectory of the combined enzyme dynamics go way beyond the LTP region.

Point 1 above is fundamental to us for the construction of our model because, in future studies, we would like to investigate how plasticity is modulated by synapse size. Indeed in the current model we mimic a relatively big synapse (corresponding to 100 AMPA receptors) with a large presynaptic domain (20-30 docked vesicles). In this case, one could probably work with an average release model at the cost of redesigning the plasticity regions. However this deterministic description would fall short when we will model a small synapse with only a few AMPA receptors (eg. 10) and a small presynaptic domain (5-10 docked vesicles). There, the failure and stochastic nature of our presynapse will be fundamental phenomena to take into account. To argue our choice of modeling for the presynapse, we added a couple of sentences in the Methods (page 22, lines 748).

5) Discuss briefly why the multi-vesicular release is not considered in the presynaptic model.

We acknowledge that multi-vesicular release (MVR) at SC-CA1 synapses was shown to be prominent after manipulations that increase release probability, e.g. during the facilitation seen with paired-pulse stimulations [Tong and Jahr 1994; Christie and Jahr 2006, Oertner et al., 2002]. However, we considered that our presynaptic model is relatively complex (as pointed out by Reviewer 3). Also, we do not hold enough information on how MVR participates in plasticity outcomes of the different protocols we used in this study. Therefore, we chose not to incorporate this mechanism in this work. We now added two sentences at the end of the description of the presynaptic model in the Methods section acknowledging this modeling choice (page 24, line 794).

6) Improve the writing (also see below). In particular, discuss the main hypotheses and assumptions underlying this work and clarify the main conclusions and goals of this modeling work. It is also important to tell the reader that 'geometric' does not imply a 'spatial' readout.

We have modified the text according to the different points raised by the reviewers.

– We corrected the writing as proposed by Reviewer 3 in paragraphs: see answers to weaknesses point 2; recommendations point 4; and minor comments points

4-16.

– We corrected inconsistencies in the methods section as proposed by Reviewer 1 – multiple points in Recommendations section.

– We modified a sentence explaining clearly that the geometric readout does not imply a spatial readout (page 7, line 232).

Reviewer #1 (Recommendations for the authors):Regarding these points:1. Noise. I like that the model produces realistic variability in plasticity. This is commendable. It is also interesting because the presence of noise neccessitates a more robust readout mechanism than a noise-free model.What is the critical noise source in the model? It should be possible to determine this (I hope it is not in the readout mechanism, as that is to most unprincipled of all included noise sources).Similarly, is the noise crucial for the model to work? Surely, as is shown the quantitative outcome of the model changes without noise, but could it have been re-adjusted? Or does the noise truly change it?

We have answered this point in the essential revision (point 1a).

As a follow-up to the current study it might be of interest to study paired synapses as was done in Bartols and Sejnowski (eLife) or MOTTA and Helmstaedter (2019) and who quantified the correlation between them. This would be a further critical test of the model and could decide if the majority of the observed fluctuations in plasticity are due to heterogeneity or noise.

We thank the reviewer for pointing out these additional studies referring to the nanoconnectomics of synaptic circuits. Beyond the current study, it could indeed be of interest to take this information into account to incorporate further constraints (ex. competition for resources, heterogeneity of the components) in the model.

2. The readout. There is not much information given on how the readout mechanism came about. How were the regions found? How can experiments be used to more precisely define these regions? Related, what about the parameters in Table 14? It is fine if they were hand-tuned but just say so.

We have answered this point in the essential revision (point 2).

3. I thought the nonlinear effects in Figure 4 and 5 were pretty cool, but the reason behind them could be explained better, as that would help these predictions further in experiments.While the paper is easy to read, I have quite a few -suggestions- for further improvement.I thought the Introduction and Discussion were rather long. In particular, the Introduction is quite philosophical. After reading it, I was actually doubtful that the paper was going to show some novelty.The J of Neurosc has 500 words for introduction and 1500 for Discussion. I think that works well.On the other hand, some of the text is not explicit enough (the Methods, in particular, see below).

We tried to be very exhaustive in the description of previous types of models attempting to describe long-term synaptic plasticity in the introduction. We now removed a few sentences that we could consider rather ‘philosophical’ and reformatted some sentences, but we added a sentence suggested by reviewer 3 to clarify our work. Omitting the citations, we are now around 600 words (modifications not highlighted in revised manuscript) and we hope that this will be satisfactory. For the discussion, we reduced it by about 300 words, leaving at about 1750 words (modifications not highlighted in revised manuscript). We hope that this will be satisfactory. Finally, we significantly reworded the methods to make the text more explicit and remove some inconsistencies (see red writing).

– It would help to replot the Tigaret data and the protocols.

We now added schematic diagrams of Tigaret STDP protocols in Figure 2a. The Tigaret data were already replotted in Figure 3g. We added captions accordingly.

– Some figures are better in the main text. I would suggest to move Figure 3S1 to the main text, as it is so crucial. Fig22 would also be good in the main text given the complexity of the dynamics; I was happy when I found it, but that was already pretty/too late.

We now added the old Figure 3S1 as new Figure 2b-d. We decided to leave the original Figure 22 (now Figure 23) at its original position in the section ‘Positioning the plasticity regions’, but we refer to it earlier in Figure 3 caption in relation to Figure 3c.

– The coloring used in Fig3e is confusing.

Unfortunately we are not sure what the reviewer refers to here, as the color coding of Figure 3e matches the color coding of Figure 3d,f,g. Maybe the reviewer meant to harmonize Figure 3i. We now modified it to better match the colors in this figure.

– l372. 'pulse dependent amplification' Not quite sure what that means.

This refers to the effect shown in Figure 4f, that the magnitude of plasticity increases with increasing number of stimulation repetitions. We agree that the phrase “pulse dependent amplification” is potentially confusing. We changed the sentence to “This general effect, that increasing pulse number tends to increase the magnitude of plasticity, was also observed in simulations of Tigaret (2016) (see Figure 3h)”.

– Formatting in Table 1 is ambiguous, e.g. '100 positive delays' and '200 negative delays' belong to Inglebert , but this is hard to parse.

we modified Table 1 to add rows.

– l 758 onwards is somewhat typical for the confusion in the Methods. THe text does not mention the D→R transition, but it is in the table.Then, the release is mentioned in l761, but then again in l764 ("In addition…").The Methods section is full of such inconsistencies.

We have corrected the methods to avoid such inconsistencies (see writing highlighted in red throughout the Methods section).

– Another example: l792 "Later we incorporate a higher number of.. parameters … with our.. model". That is poor English.

We corrected this sentence (now page 23, line 789).

– Fig8 Caption "Our model also does not cover the aboslishing of release probability". Apart from poor English, it is confusing as it is unclear how this relates to the figure.

We removed this sentence.

– Fig8. Why is there such a very sudden drop in the 50Hz curve? It looks odd to me given the dynamics.

By a back-of-the-envelope estimate, when looking at Figure 8d, every pre-synaptic spike induces a release D -> D-1 with probability 0.75 at the beginning of the stimulation. At high frequency, replenishment of vesicles does not have time to occur. Hence D0/0.75 ~ 25/0.75 ~ 33 pulses before the pool is empty which is about the spike number at which the drop occurs.

– p24 Note that one also needs to specify the units of the alphas and betas. It is also somewhat in conflict with the common convention to call \tau_\infty m_\tau.

We agree with the reviewer about unorthodox notations and loose units definition of α and β. However, as we used a model by Migliore et al. 1999, we chose to stick to their notations. We added a sentence on page 26 to help the reader understand the units.

– Table4. The spine and dendrite have time constant of 1.5 s. That is very, very long. Why is this necessary?

We agree that this *apparent* time constant of 1.5 s is much slower than the typical timescales of voltage dynamics in spines and dendrites. However for this model, these particular values are irrelevant. The reason is related to the fact that the classic interpretation of membrane time constant comes from point neuron models where all current flows across the membrane. In contrast, in spatially extended neuron models, current can also flow axially. The rate of voltage change in any given compartment depends on both axial and membrane current flow. In dendritic spines, the membrane leak conductance has been previously estimated to be so small (< 1 nS) relative to the spine neck conductance (~10 nS) that only a negligible current flows across the spine membrane (Koch, Biophysics of Computation, 1998), making its apparent time constant negligible. As for the dendrite, although we might indeed expect more membrane leak current flow than in the spine case, in our model the soma is quite a large current sink, so again most of the current escape from the dendrite occurs axially rather than across the membrane. We designed the model like this to make the voltage dynamics reproduce Goldings et al. 2001 using a passive dendrite (Figure 9 a,b). To demonstrate these current flow patterns, we plot in Author response image 2 the voltage and current dynamics for one pairing of the Tigaret et al. (2016) 1Pre1Post10 protocol. Left panel: voltage equations 3, 4, and 5 of the manuscript (note that the spine trace completely overlaps with the dendrite trace and is thus diff). Middle panel: leak current for the spine and dendrite, respectively g_L_spine * (E_rev-V_sp) and g_L_dend * (E_rev-V_dend) showing that they are nearly zero, in comparison to the soma leak. Right panel: g_L_soma * (E_rev – V_soma) showing that most of the leak current scape across the soma.

Note that in the left panel (1) dendrite and spine voltages nearly match; (2) in the middle panel, leak currents from dendrite and spine are nearly zero; (3) in the left panel, the soma current dictates the voltage leak in the system; (4) the voltage in all compartments decays at a timescale of <10 ms, much faster than the apparent membrane time constants of 1.5 s (correctly) calculated by the reviewer.

In order to further demonstrate that the dendrite membrane leak current is negligible in this particular model, we also ran additional simulations where the dendrite membrane leak conductance was set to zero. The plot in Author response image 3 shows the probability distributions for the weight changes from the seven plasticity protocols in Tigaret et al. (2016) where the left side of each shape is from simulations without dendrite leak, and the right side is from simulations with dendrite leak (as in the paper). The fact that the two distributions appear visually identical indicates that the dendrite leak conductance has only a negligible effect on the plasticity outcomes.

**Author response image 3. sa2fig3:** 

– Table4. 10th line dendritic leak reversal should read differently.

We corrected to ‘dendritic leak conductance’.

– Equation1 uses square brackets to denote a concentration for outside Ca, but not for CaPre which is presumably also a concentration.

We corrected the notations as suggested.

– Was spontaneous release included in the synapse?

Spontaneous release was not included in our model. As pointed out by reviewer 3, the presynaptic model is already quite complex and therefore we did not want to add additional complexity.

Reviewer #2 (Recommendations for the authors):The model proposed in this study is complex and multi scale combining elements from previous work and grounding the choice of many parameter values on experimental measurements and estimates. As such the tables summarising the modelling details in the Materials and methods section use different units:1) For time – ms and s;2) Electrical currents conductance – nS/um^2, nS and pS.3) Concentration – μm and M.I would suggest that the authors unify the presentation by using consistent units for all modelling components.

We corrected these inconsistencies using the same units for the modelling components within the same tables.

I have some further specific recommendation regarding the modelling description:1) Please provide details regarding the threshold function in Equation (1).

The function p_rel was fitted in Figure 8e against data from Tigaret et al. 2016 and Hardingham et al. 2006. We added a sentence to clarify this on page 23, line 766.

2) Please clarify if Equation (7) has been fitted to data.

Equation (7) was fitted using data from Buchanan and Mellor 2007, see Figure 9b. We better explained this on page 28, line 833.

3) Please justify the calcium buffer and dye model assumptions that both bind one claim ion only. This is important as calcium indicators such as calmodulin are know to bind 4 calcium ions.

While it is true that prominent genetically-encoded calcium indicators such as GCaMP are based on Calmodulin and therefore inherit its co-operative, 4-site calcium binding properties, in the Tigaret et al. 2016 experiments which we are replicating, they used the synthetic dye Fluo-5f which is well-modelled with a single ca^2+^-dye binding reaction (Maravall et al., 2000; Bartol et al., 2015). We note that we are explicitly modelling Calmodulin and including its co-operative nonlinear calcium binding properties. As for the remaining endogenous calcium buffers (other than Calmodulin), it is true that they may potentially have more complicated calcium binding properties, but they are unfortunately poorly quantified experimentally. Instead we include a parsimonious generic buffer model that represents an aggregate of these largely unknown endogenous buffers. Future iterations of the model could include more detailed versions of these endogenous buffers, e.g. calbindin (Bartol et al., 2015). We have added new text to this effect in the Methods section (lines page 32, 894) to further justify our modelling choices for the dye and buffers.

4) Could you please justify Equation (19).

The first two equations of (19) represent the total activation of K (resp. P) associated with CaM as defined in the code (https://github.com/ryoheiyasuda/CaMKII_Simulation/blob/master/CaMKII_SimRev.py) that accompanies Chang et al. 2019. The last equation is from Chang et al. 2019. We wrote a sentence to clarify on page 48, line 1096.

Reviewer #3 (Recommendations for the authors):1) Page 22, Equations. 1-2: Biologically, the effect of [Ca]_0 is certainly to increase the size of [ca^2+^] jump during an action potential. However, the size of the ca^2+^ jump in the model is normalized to 1, so the external [ca^2+^] is assumed to directly affect the dependence of release probability on [ca^2+^], without affecting the [ca^2+^] dynamics directly. This shows that the model is purely phenomenological and that the [ca^2+^]pre variable does not represent [ca^2+^] in any meaningful way. This makes it very hard to understand the model presented in Equations. 1-2, which is quite distracting, particularly since this bears a minimal impact on the main part of this work. Moreover, there are as many as 4 free parameters in the "half-activation" function h([Ca]). Including the value of cooperativity "s", the total number of parameters is equal to 5. This equals the number of data points in Figure 8e which is presumably used to calibrate the presynaptic model. The number of parameters seems very large, and the presynaptic model appears unnecessarily complicated. Even the most comprehensive, realistic biophysical models of Ca-triggered exocytosis from the calyx of Held literature do not have as many parameters. I highly suggest replacing the model with a simpler model, whose variables are biologically interpretable.

We agree that this part concerning the dependency of the model on [Ca]_0 is phenomenological and it could be therefore improved. However, given that we are only interested in p_rel(Ca_0), which we fit to experimental data, doing so with a phenomenological model (as we did in Figure 8e) or with a more biophysical model would not change the p_rel function. Additionally making this change would require resimulation of all the figures with little impact on the results of the paper.

2) Given the large number of parameters in the model (with 14 Tables in the main part of the manuscript), it would be of value to clearly summarize which of the model parameters were considered to be free parameters that were adjusted to fit the LTP/LTD induction data, apart from the geometric read-out parameters, and which of the model parameters (e.g. the ionic conductances) were assumed to be generic and therefore not adjusted when fitting experimental data.

We have answered this point in the essential revision (point 2).

3) I suggest explaining in one sentence why the possibility of multi-vesicular release is not considered in the presynaptic model.

We have answered this point in the essential revision (point 5).

4) Throughout the manuscript, the verbs "predicted", "found", "hypothesized" are used very loosely when discussing the behavior of the model and its elements, rather than in describing experimentally verifiable predictions. Moreover, when used in the past tense, verbs "predicted" and "found" is suggestive of predictions that have been experimentally verified. This makes the manuscript hard to read since in some places it is hard to distinguish statements about model behavior from statements about past experimental results or verifiable predictions. Concrete comments related to this confusing phrasing follow below.

We have amended the text accordingly using the comments made by review.

[Editors' note: further revisions were suggested prior to acceptance, as described below.]

The manuscript has been improved but there are some remaining issues that need to be addressed, as outlined below:Reviewer #1 (Recommendations for the authors):The authors have addressed all my issues.Regarding the contribution of the various noises, it would be nice to have the figure from the reply letter in the paper.(Of course, it would have been a bit cleaner to switch the noises ON one by one, but this figure is already informative).

We thank the reviewer for the positive comment on our revised manuscript. We now added the figure regarding the importance of noise as ‘Figure 4 – Supplement 2’ and added a paragraph describing this figure on page 12.

Reviewer #2 (Recommendations for the authors):I do appreciate the amount of work done by the authors in addressing reviews comments and suggestions. The manuscript has been improved and most of my comments have been addressed satisfactorily except for the following points:(1) Regarding SK deterministic modelling and justification – it is simply not true that:(1.1) There is "a lack of single-channel recordings of SK channels", please seeHirschberg, B., Maylie, J., Adelman, J., and Marrion, N. (1998). Gating of recombinant small-conductance Ca-activated K^+^ channels by calcium. The Journal of General Physiology, 111(4), 565.Hirschberg, B., Maylie, J., Adelman, J., and Marrion, N. (1999). Gating properties of single SK channels in hippocampal CA1 pyramidal neurons. Biophysical Journal, 77(4), 1905-1913.(1.2) There is "a lack of published stochastic models of SK channel", please seeStanley, D.A., Bardakjian, B.L., Spano, M.L. et al. Stochastic amplification of calcium-activated potassium currents in ca^2+^ microdomains. J Comput Neurosci 31, 647-666 (2011). https://doi.org/10.1007/s10827-011-0328-x(available at https://link.springer.com/article/10.1007/s10827-011-0328-x#Equ6)The above needs to be addressed.

We thank the reviewer for the positive comment on the revised manuscript. We are sorry that we missed these references regarding single channel recordings and stochastic models of SK channels and we thank the reviewer for pointing them out. We modified the erroneous sentence on page 40 to mention these publications.

2) The units for the various model parameters are still inconsistent, eg. Table 6. uses M for concentration and s for time while Table 7. – μm for concentration and ms for time. There are other instances along the lines of the above in the rest of the tables presenting model parameters throughout the manuscript.I do know this is a matter of simple scaling but the parameter units used throughout should be consistent to avoid reproducibility issues that could be faced by other researchers and especially early career researchers studying this work in the future.

We corrected the remaining inconsistencies throughout the tables.

Reviewer #3 (Recommendations for the authors):The authors have made a significant update to their manuscript in order to address the Reviewers' comments, adding a couple more simulation results to explain different features of the model. In general, I find this update sufficiently complete.The only point I am still not fully satisfied with is the model describing the dependence of vesicle release probability on extracellular [ca^2+^] (Equations. 1-2, p. 23), retained from the original version of the manuscript since it appears both cumbersome and unphysiological. However, I completely agree with the Authors that correcting this would not affect the main conclusions while requiring additional but unnecessary simulations.Therefore, with respect to the latter, I would be fully satisfied if the Authors added a single sentence on page 23, reiterating their response to this point, namely: (1) the presynaptic model for p_rel is purely phenomenological, (2) in principle, the [Ca] jump parameter δ_Ca should depend on [Ca]_0, but (3) replacing the model with a more physiological model would not affect the results, since the measured dependence of release probability on [Ca]_0 is already satisfied by this phenomenological model.

We thank the reviewer for the positive comments on the revised manuscript. We now added the suggested sentence on page 24 as proposed.